# Ontogeny and transcriptional regulation of Thetis cells

Yoselin A. Paucar Iza[1,2,3], Tyler Park[4], Eliyambuya Baker[3], Gayathri Shibu[2,3], Tilman Hoelting[3], Greyson Feather[3], Anushka Yadav[3], Yollanda Franco Parisotto[1,3], Zihan Zhao[3], Blossom Akagbosu[3], Marc Elosua Bayes[3,10], Logan Fisher[2,3], Lucas M. James[5,6,7], Jianping Ma[8], Benjamin D. Philpot[5,6,7], Behdad Afzali[8], Christina Leslie[4] & Chrysothemis C. Brown[1,2,3,9 ✉]

Thetis cells (TCs) are a recently identified lineage of RORγt⁺ antigen-presenting cells comprising four subsets, TC I to TC IV, including a tolerogenic subset (TC IV) that instructs tolerance to gut microbiota and food antigens[1–6]. A developmental wave of TCs during early life creates a crucial window of opportunity for establishing intestinal tolerance[1,5]. The ontogeny of TCs and the cues that shape their abundance and heterogeneity remain unknown, however, limiting efforts to harness their therapeutic potential. Here we identify a population of RORγt⁺ progenitors, termed Thetis–lymphoid tissue inducer progenitors (TLPs), that give rise to the immediate TC progenitor (TCP) and the lymphoid tissue inducer (LTi) progenitor (LTiP), and identify PU.1 as the transcription factor that governs TC fate. Despite transcriptional similarity to myeloid-derived conventional dendritic cells, we show that TCs descend from the common lymphoid progenitor. Deletion of the plasmacytoid dendritic cell (pDC) lineage-determining transcription factor TCF4 expands TLPs and TCs, indicating a shared developmental branch with pDCs. TLPs are enriched in fetal liver, but, unlike LTi cells, TCs emerge postnatally, indicating that developmentally timed environmental cues promote TCP differentiation. We identify one such cue, RANKL provision by lymphoid tissue organizer cells, that is essential for TC I differentiation. Together, these findings define the ontogeny of TCs and the transcription factors that promote TC differentiation and heterogeneity, facilitating future investigations of these enigmatic cells and their therapeutic potential for tolerance induction in food allergy and autoimmunity.

Antigen-presenting cells (APCs) determine the nature of the immune response, initiating diverse inflammatory T cell responses to pathogens or tolerance to self and innocuous foreign antigens. This functional versatility is achieved by distinct subsets of specialized APCs. Conventional dendritic cells (cDCs) dominate current frameworks of APC-directed T cell differentiation[7,8], but several cell types express major histocompatibility complex class II (MHCII), including B cells, monocytes, macrophages, pDCs and RORγt⁺ LTi cells, with RORγt⁺ LTi cells having uncertain roles in T cell priming and being known mainly for their central role in lymph-node organogenesis[9]. The distinct developmental origins of these APC lineages, spanning both myeloid and lymphoid progenitors, enable precise spatiotemporal regulation of the APC landscape across tissues and developmental stages. Adding to this repertoire, studies have identified a group of RORγt⁺ APCs, referred to here as TCs, that encompasses four subsets (TC I–IV) that are transcriptionally distinct from RORγt⁺MHCII⁺ LTi cells[1–5,7,10–12]. Although the

exact roles of TC I–III remain largely unknown, TC IV has an essential role in establishing tolerance to gut microbiota and food antigens by promoting peripherally induced regulatory T (pT_reg) cells[1–6], highlighting the therapeutic potential of TCs in food allergy, autoimmunity and transplantation.

TCs emerge in a developmental wave, most prominently in gut lymph nodes at the time of weaning[1,2,5,11,13], ensuring enriched representation of tolerogenic APCs during the first encounters with microbiota and dietary antigens and indicating that TC abundance could determine the window of opportunity for enhanced intestinal pT_reg cell differentiation[5,14–16]. Understanding the pathways and environmental cues that drive TC differentiation could provide insights into how intestinal tolerance is regulated and reveal avenues to increase TC abundance and harness their immunoregulatory function in adults. At present, the ontogeny of TCs and their developmental pathways are not known. Named for their unusual hybrid phenotype, TCs display

[1]Howard Hughes Medical Institute, New York, NY, USA. [2]Immunology and Microbial Pathogenesis Program, Weill Cornell Medicine Graduate School of Medical Sciences, New York, NY, USA. [3]Immuno-Oncology Program, Memorial Sloan Kettering Cancer Center, New York, NY, USA. [4]Computational and Systems Biology Program, Memorial Sloan Kettering Cancer Center, New York, NY, USA. [5]Department of Cell Biology and Physiology, University of North Carolina at Chapel Hill, Chapel Hill, NC, USA. [6]Neuroscience Center, University of North Carolina at Chapel Hill, Chapel Hill, NC, USA. [7]Carolina Institute for Developmental Disabilities, University of North Carolina at Chapel Hill, Chapel Hill, NC, USA. [8]Immunoregulation Section, Kidney Diseases Branch, National Institute of Diabetes and Digestive and Kidney Diseases, NIH, Bethesda, MD, USA. [9]Department of Pediatrics, Memorial Sloan Kettering Cancer Center, New York, NY, USA. [10]Present address: Boston Children's Hospital, Harvard Medical School, Boston, MA, USA. ✉e-mail: brownc10@mskcc.org

transcriptional features of both cDCs and AIRE-expressing medullary thymic epithelial cells (mTECs), including AIRE expression in TC I and TC III[1,7]. Recent studies have proposed reclassifying TCs as dendritic cells, termed RORγt[+] DCs[2] or tolerizing DCs[3], while others have suggested that TCs belong to the family of RORγt[+] type 3 innate lymphoid cells (ILC3s), along with LTi cells[17]. However, ontogenetic analyses have established that TCs are derived from neither the common dendritic cell progenitor (CDP)[1,2,5,6] nor the ILC progenitor (ILCP)[1], underscoring the need to define their developmental origin and lineage relationships.

## TC progenitors are enriched in fetal liver

Given their transcriptional overlap with AIRE[+] mTECs, we first sought to establish whether all TC subsets are haematopoietic in origin. To address this, we generated bone marrow (BM) chimeras with CD45.2 and CD45.1 mice (Fig. 1a). Analysis of mesenteric lymph nodes (mLNs) 4 weeks after irradiation and transfer of CD45.2 donor cells confirmed the presence of all TC subsets among CD45.2 cells (Extended Data Fig. 1a,b). Conversely, host CD45.1[+] TC subsets were almost completely absent, despite efficient generation of TCs in reciprocal chimeras with donor CD45.1 BM (Extended Data Fig. 1c), demonstrating that TCs are radiosensitive haematopoietic cells. The developmental window for TCs in early life prompted us to examine whether fetal liver (FL) might also be a source of TC progenitors. Indeed, analysis of FL chimeras 4 weeks after reconstitution demonstrated the presence of all of the TC subsets among CD45.2 cells in mLN (Extended Data Fig. 1d,e). Strikingly, TCs, as well as LTi cells, were more abundant in mice reconstituted with FL than BM cells (Fig. 1b and Extended Data Fig. 1f), and this increase in cell number was not observed for other innate cell types, such as pDCs and cDCs (Extended Data Fig. 1f), indicating that FL is enriched in a progenitor with TC differentiation potential. To determine when TCs first arose, we examined *Rorc*[Venus-creERT2] (hereafter *Rorc*[Venus]) mice and found that TCs were first detectable within the mLN at postnatal day 4 (P4) (Fig. 1c).

The classical model of haematopoiesis describes the progressive differentiation of haematopoietic stem cells (HSCs) into lymphoid- or myeloid-restricted progenitors. Whereas cDCs are descended from CX3CR1[+] myeloid progenitors[18,19], pDCs arise from distinct CX3CR1[+] pre-pDC progenitors, which are reportedly descended from the common lymphoid progenitor (CLP)[20–22], although this is debated[23,24]. To determine the ontogeny of TCs, we first analysed *Il7r*[cre]*R26*[lsl-tdTomato] mice in which IL7R[+] progenitors, including CLPs and pre-pDCs, and their descendants, are labelled[25]. Given that FL CLPs retain myeloid potential and thus contribute to cDC differentiation during early life[26,27], we analysed both young (2 weeks old) and adult (9 weeks old) mice. Around 75% of cDCs were tdTomato[+] in 2-week-old mice, declining to 20–40% in adult mice (Fig. 1d). By contrast, despite only around 16% of TCs expressing IL7R (Extended Data Fig. 2a), we observed almost universal expression of tdTomato by IL7R[−] TCs in both age groups (Fig. 1d), with uniform labelling across TC I–IV subsets (Extended Data Fig. 2b), indicating that TCs are derived exclusively from an IL7R[+] progenitor, regardless of developmental stage. Unlike early-life CLP-derived cDCs, which still passed through an intermediate CX3CR1[+] myeloid progenitor, as shown by more than 90% labelling in P14 *Cx3cr1*[cre] *R26*[lsl-tdTomato] mice (Extended Data Fig. 2c), TCs exhibited similar labelling to lymphoid cells such as T cells and ILCs (about 60%). Taken together, these data indicate that TCs arise from an IL7R[+] progenitor that is enriched in FL.

## Identification of the TC progenitor

To determine the identity of the immediate TC progenitor and delineate the developmental hierarchy of 'TC-poiesis' (Fig. 1e), we performed single-cell RNA-sequencing (scRNA-seq) on Lin[−] IL7R fate-mapped and Lin[−]RORγt(Venus)[+] cells from *Il7r*[cre]*R26*[lsl-tdTomato] and *Rorc*[Venus] mice, respectively. Cells were isolated from FL, BM and mesenteric anlagen at embryonic days 16.5 to 18.5 (E16.5–18.5), and from liver, BM and mLN at P7, to identify all of the potential sites and stages of TC development (Fig. 1f, Supplementary Fig. 1 and Supplementary Table 1). Notably, BM Lin[−]RORγt(Venus)[+] cells were almost undetectable and were not captured. Cells were barcoded and fixed, enabling simultaneous RNA profiling and post-hoc identification of cells according to age and tissue of origin (Extended Data Fig. 3a). We retained 101,192 cells following quality control. Unsupervised clustering revealed the full repertoire of haematopoietic progenitors, defined by canonical genes (Fig. 1g and Extended Data Fig. 3b), Among these, *Il7r* expression was observed in CLPs and a fraction of lymphoid myeloid-primed progenitors (LMPPs; Extended Data Fig. 3c), consistent with data from adult BM progenitors[25]. Flow cytometry of P10 *Il7r*[cre]*R26*[lsl-tdTomato] spleen demonstrated labelling across granulocyte and myeloid lineages (Extended Data Fig. 3d), confirming the multipotent nature of early-life CLPs.

To obtain a more granular picture of TC development, we subclustered FL LMPPs, CLPs, monocyte dendritic progenitors (MDPs), CDPs and ILCPs alongside the earliest ILC and TC clusters in mesenteric anlagen and mLN, respectively (Fig. 1h,i and Extended Data Fig. 3e,f), and applied CellRank 2 for reconstruction of cell-fate trajectories[28]. This revealed distinct developmental trajectories for ILCs, LTi cells, cDCs and TCs (Extended Data Fig. 3g). ILCPs expressing the canonical genes *Zbtb16*, *Pdcd1*, *Tox* and *Tcf7* emanated from a CLP/LMPP cluster, whereas preDC1s and preDC2s arose from MDP/CDP cells. Two clusters expressing the signature pre-pDC genes *Siglech*, *Bst2* and *Tcf4* emanated from LMPP/CLP cells. Although this signature is also characteristic of ontogenically related transitional DCs (tDCs), which give rise to cDC2s by a non-canonical developmental pathway[21,22,29–31] (Fig. 1e), these clusters lacked expression of the tDC marker *Cd300lg*[22] (Fig. 1i).

As well as these known progenitors, we identified a heterogeneous RORγt[+] cluster that included FL cells with an increased probability of assuming TC fate (Fig. 1j,k). To gain finer resolution, we subclustered these cells into three clusters (Fig. 1l). Two clusters largely derived from mLN (Fig. 1m). Of these, one exhibited low TC fate probability and aligned transcriptionally with ILCPs (Fig. 1j and Extended Data Fig. 3e), including expression of *Zbtb16* and *Pdcd1* (Fig. 1m), indicating that these cells represent the previously described RORγt[+] LTiPs[32–34]. The second mLN-derived cluster aligned transcriptionally with TCs and exhibited high TC fate probability (Fig. 1j and Extended Data Fig. 3e), indicating that these cells represent the elusive immediate TCP. The remaining cluster, comprising FL cells, aligned transcriptionally with TCs; however, pseudotime analysis predicted that these cells represented the precursors to both LTiPs and TCPs (Fig. 1j,n), so we named these cells Thetis–lymphoid tissue inducer progenitors (TLPs). Consistent with previous reports of separate developmental origins for LTiPs and ILCPs[32,35–38], the predicted differentiation pathway of CLP → TLP → LTiP → LTi cells was distinct from the trajectory of CLP → ILCP → ILC subsets (Fig. 1n). TLPs did not express *Thy1* (CD90) or *Cxcr6* (Extended Data Fig. 3h), which are markers previously associated with FL LTiPs. Instead, these genes were expressed by more terminally differentiated FL ILC3 and LTi cells (Extended Data Fig. 3h). TLPs expressed *Il1r2*, *Itgam*, *Sirpa* and *Csf1r*, as well as stem-like (*Cd34* and *Igfbp4*) and cell proliferation (*Stmn1*) genes, and low levels of MHCII (*H2ab1*) (Fig. 1m,o and Extended Data Fig. 3e), consistent with their progenitor identity. Flow cytometry of Lin[−]CXCR6[−] cells in the FL of *Rorc*[Venus] mice confirmed the presence of RORγt(Venus)[+] cells expressing TLP markers (Fig. 1p).

Distinguishing genes for the immediate TCP included *Il1r2*, *Gria3*, *Tmem176a/b* and *Tnfsf11* (Fig. 1m), closely resembling the signature of a cluster of 'early' TCs, distinct from TC I–IV, identified in our previous scRNA-seq on TCs (CXCR6[−]RORγt[+]MHCII[+]) from different lymph

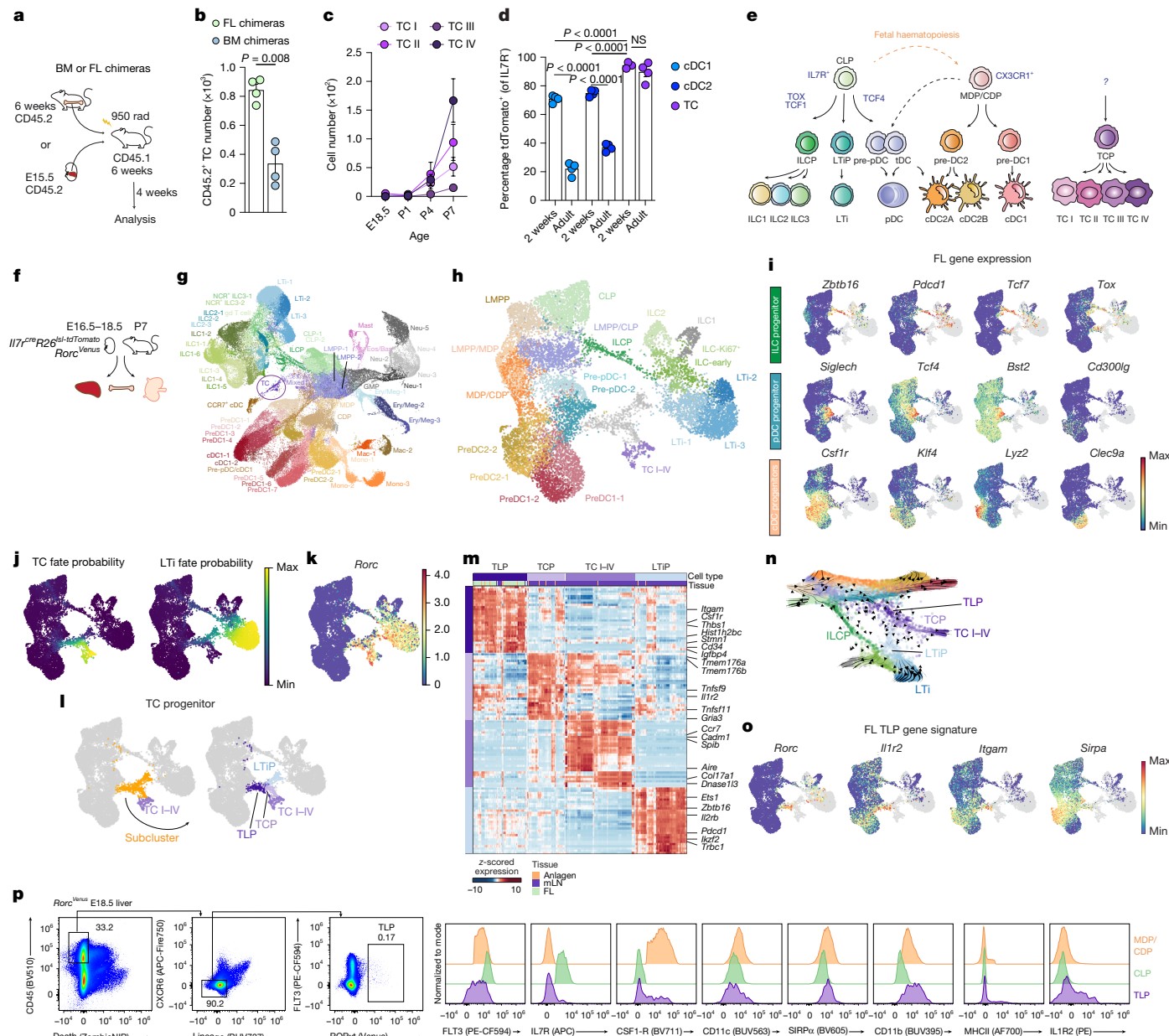

**Fig. 1 | Identification of the TC progenitor. a**, Schema for BM and FL chimeras. **b**, Number of CD45.2⁺ TCs in FL and BM chimeras, as in **a** (*n* = 4 mice per group). **c**, Number of TC I–IV in mesenteric anlagen or mLNs at the indicated ages (*n* = 5 mice for E18.5 and *n* = 6 mice for all other time points). **d**, Frequency of tdTomato-labelled cDCs or TCs isolated from mLN of 2- or 9-week-old *Il7r^cre^R26^lsl-tdTomato^* mice (*n* = 4 per group). Each symbol represents an individual mouse. **e**, Schematic representation of the classical model of haematopoiesis. **f**, Schema for scRNA-seq on IL7R fate mapped and RORγt⁺ cells during FL and early postnatal haematopoiesis. **g**, Uniform manifold approximation and projection (UMAP) of 101,192 cells, as in **f**, coloured by cluster annotation. The TC cluster is circled in purple. NCR⁺ ILC3, natural cytotoxicity receptor-positive ILC3; Mono, monocyte; Mac, macrophage; Neu, neutrophil/granulocyte progenitor; Eos/Bas, eosinophil/basophil progenitor; Ery/Meg, erythroid/megakaryocyte progenitor; GMP, granulocyte-monocyte progenitor. **h**, UMAP embedding of 14,581 FL progenitors, TCs and ILCs subsetted from **f** and coloured by cluster annotation. **i**, UMAP with FL-derived cells coloured by unimputed expression of the indicated progenitor genes. Non-FL-derived cells are coloured in grey. **j**, UMAP coloured by TC fate probability or LTi fate probability. **k**, UMAP coloured by unimputed expression of *Rorc* (right). **l**, Subclustering of cells with high TC fate probability, shown in orange. **m**, Heatmap showing scaled, imputed expression of the top 20 differentially expressed genes (one versus the rest, fold change (FC) > 1.5, adjusted *P* < 0.01) for TLP, TCP, LTiP and TC I–IV clusters. **n**, Force-directed layout of cells as in **h**, coloured by cluster annotation. Arrows and lines represent the direction of expected cell-state transitions inferred from the CellRank transition matrix, showing progression or progenitor cells towards terminal states. **o**, UMAP coloured by unimputed expression of TLP signature genes. **p**, Flow-cytometry identification of TLP and expression of TLP markers by indicated progenitors in E18.5 FL from *Rorc^Venus^* mice. Representative of *n* = 4 mice. Data in **a**–**c** are representative of 2–3 independent experiments; data in **p** are representative of more than 10 independent experiments. Each symbol represents an individual mouse. Error bars: mean ± s.e.m.; two-tailed unpaired *t*-test (**b**) or two-way analysis of variance (ANOVA; **d**).

nodes of P14 *Rorc^Venus^* mice[5] (Extended Data Fig. 4a). Indeed, projection of the early-TC signature onto our progenitor UMAP demonstrated enrichment in the TCP cluster (Extended Data Fig. 4b), establishing the presence of TCPs across all lymph nodes.

Overall, these findings identified previously uncharacterized RORγt⁺ TC progenitors at distinct stages of differentiation in FL and lymph nodes, and indicate that there is a common developmental pathway for TCs and LTi cells that is distinct from the ILC pathway.

## Establishing precursor–progeny relationships

To establish a strategy for prospective TCP isolation, we performed Smart-seq3 (SS3) on CXCR6−RORγt(Venus)+MHCII+ cells from mLNs of 2-week-old *Rorc*[Venus] mice (Supplementary Fig. 2). Unsupervised clustering and analysis of gene signatures confirmed the presence of TC I–IV, Ki67+ TC, TCP, LTiP and LTi clusters (Fig. 2a,b and Extended Data Fig. 4c). Index-sorting analysis validated our gating strategy for TC I–IV subsets (Extended Data Fig. 4d), used throughout this study, and revealed that both TCPs and LTiPs could be distinguished from other TC subsets by the lack of NCAM1 and EpCAM expression (Fig. 2c and Extended Data Fig. 4e). Among NCAM1−EpCAM− cells, around 40% represented TCPs or TCPs at early stages of differentiation (early TCs; Extended Data Fig. 4f). Distinguishing genes for LTiPs versus TCPs included *Pdcd1* and *Il1r2*, respectively (Fig. 2b). Flow cytometry of RORγt+CXCR6−NCAM1− EpCAM− cells confirmed mutually exclusive expression of IL1R2 and PD1 (Fig. 2d and Extended Data Fig. 4g), and showed that about 40% of these cells were Ki67+ (Extended Data Fig. 4h).

We next sorted RORγt(Venus)+CXCR6−NCAM1−EpCAM−PD1− cells, encompassing IL1R2+ TCPs and IL1R2− early TCs, from mLNs of P11– 12 *Rorc*[Venus] mice (Extended Data Fig. 4i), and cultured them on mLN slices from 2–3-week-old CD45.1 mice, to recapitulate the physiological environment for TC differentiation (Extended Data Fig. 4j). After 24 h, most of the CD45.2+ cells were RORγt(Venus)+ MHCII+CXCR6− and had acquired TC II–IV features, including EpCAM and CD11c expression (Fig. 2e). Crucially, TCP cultures failed to generate CXCR6+ LTi cells or CD11c+MHCII+ cDC subsets, demonstrating restricted TC differentiation potential. Together, these data identify the transcriptional and molecular features of TCPs and establish the TCP as the immediate common progenitor for TCs.

We next tested whether the putative TLP population represented the precursors to TCPs and LTiPs. Differential gene expression analysis of the TLP, TCP and LTiP clusters (Fig. 1l) identified *Siglecf* expression in TLPs and TCPs, but not in LTiPs or TC I–IV (Fig. 2f and Supplementary Table 2), a finding confirmed by orthogonal analysis of SS3-derived TCP and LTiP clusters (Fig. 2f) and flow cytometry of FL TLPs (Extended Data Fig. 4k). Apart from granulocyte progenitors and a small proportion of pre-DCs, *Siglecf* was not expressed by candidate TC and LTi precursor cell types in the FL and BM (Fig. 2f and Extended Data Fig. 4l), indicating the utility of *Siglecf*[cre] for tracing the progeny of TLPs and TCPs. Notably, we found that around 40–50% of LTi cells and TC I–IV subsets were tdTomato+ in the mLN of P11 *Siglecf*[cre]*R26*[lsl-tdTomato]*Rorc*[Venus] mice (Fig. 2g and Extended Data Fig. 4m), compared with around 10–20% of ILC subsets.

Overall, these results indicate that TCs and LTi cells are descended from FL RORγt+ TLPs, which give rise to progenitors in the lymph nodes with restricted TC or LTi differentiation potential.

## TCs arise from the CLP

Because TLPs and TCPs were mainly IL7R− (Fig. 1p and Extended Data Fig. 5a), IL7R fate mapping in TCs reflected descendancy from an upstream IL7R+ progenitor. Analysis of FL progenitors (Fig. 1h) revealed *Il7r* expression by CLPs, a mixed LMMP/CLP cluster and pre-pDCs (Extended Data Fig. 5a), the last of which potentially included tDCs, owing to their transcriptional similarity. The established CLP origin of LTi cells[26,39,40] indicated that TLPs were CLP-derived. However, because TCs have been described as DCs, based on transcriptomic similarity[2,3,11,41], we sought to determine whether TC development proceeds through a CLP → TLP → TCP pathway or through cDC progenitors, either canonical myeloid CDPs or lymphoid-derived tDCs (Fig. 1e). We therefore established stringent gating strategies for FACS isolation of FL CLPs (Lin−FLT3+IL7R+CSF1R−Ly6D−CD27+), pre-pDC/tDCs (Lin−FLT3+ CSF1R−Siglec-H+) and MDP/CDPs (Lin−FLT3+IL7R−CSF1R+MHCII−/lo) from E17.5–18.5 *Rorc*[Venus] mice, informed by our analysis of FL progenitors

(Fig. 1i and Extended Data Fig. 5b,c) and previously refined CLP definitions[42]. The CD45.2+ progenitors were transferred into non-irradiated CD45.1 P2 mice and analysed 6 days later (Fig. 3a). Only CLP transfers generated TCs in spleen (Fig. 3b,c), whereas MDP/CDPs yielded cDC1, cDC2 and monocytes, and pre-pDC/tDC progenitors produced pDCs and cDCs (Fig. 3b,c). In mLN, CLP similarly generated TCs, encompassing all four subsets (Extended Data Fig. 5d), but insufficient recovery precluded analysis of pre-pDC/tDC or MDP/CDP progeny. Given the low recovery of tDC/pre-pDC progeny, to definitively exclude a precursor– progeny relationship between pre-pDC/tDCs and TCs, we examined *Tcf4*[STOP] mice lacking expression of TCF4 (E2-2), the crucial transcription factor required for pDC and tDC development[30,43]. Unexpectedly, TLP numbers were increased in FL of E18.5 *Tcf4*[STOP/STOP] mice (Fig. 3d), indicating that TCs are not derived from either pre-pDCs or tDCs, and instead indicating a role for TCF4 in restraining TLP cell fate.

To examine a cell-intrinsic role for TCF4 in suppressing TC development, we generated competitive mixed CD45.1/*Tcf4*[STOP/STOP] FL chimeras (Fig. 3e), circumventing postnatal lethality in TCF4-null mice[44]. In line with our previous findings, TC subsets were preferentially derived from TCF4-deficient FL (Fig. 3e,f and Extended Data Fig. 5e), suggesting that TLPs and pre-pDCs arise from a shared progenitor, with TCF4 expression determining the bifurcation in cell fate. Notably, our earlier CellRank 2 analysis of cell-fate trajectories predicted that CLP-derived IL7R+SiglecH[lo/int] cells are precursors of both TLPs and pre-pDCs (Fig. 3g). To test this idea, we used *Siglech*[cre]*R26*[lsl-Tdtomato] mice and found an almost identical frequency of labelled cells (around 20%) among LTi cells and TC subsets in P14 mLN, compared with about 5% labelling in ILCs (Fig. 3h), resolving the enigmatic origins of both TC and LTi cell precursors (Fig. 3i).

As well as transcriptional similarity with DCs, impaired differentiation of TCs in *Flt3l*[−/−] mice could be interpreted as potential DC identity, despite similar defects being found in lymphoid lineages including ILCs and LTi cells[11,45–47]. Notably, although around 40% of TLPs express FLT3 (Extended Data Fig. 6a), this was not maintained in mLN TCP or TC subsets (Extended Data Fig. 6b). Inducible FLT3 activation in neonatal or adult R26:FlpoERT2 *Flt3*[Frt-ITD] mice increased TC numbers by around 2–4-fold in mLNs and skin-draining peripheral LNs (pLNs), but did not affect splenic TCs, despite robust expansion of cDCs and pDCs (Extended Data Fig. 6c). Thus, FLT3 regulation of TCs is LN dependent and modest relative to its role in DCs.

Overall, although rare IL7R+ myeloid-primed pathways cannot be excluded, CLP transfers yielding TCs, alongside exclusion of tDC or CDP origin, indicate that TCs are a lymphoid-restricted lineage, developmentally distinct from cDCs.

## RORγt is dispensable for TCP specification

We next sought to identify the transcription factors that instruct TC fate, first focusing on RORγt, the defining marker of TCs and their immediate progenitors. As well as the established RORγt dependency of LTi cells[9], previous studies that used mice deficient for the +7-kilobase *Rorc* enhancer or *Cd11c*[cre]*Rorgt*[fl/fl] mice revealed impaired differentiation of TC II–IV (termed tolDC or RORγt+ DC II–IV), but TC I remained unaffected[2,3]. Preservation of TC I in these models could reflect alternative *Rorc* enhancer usage (Extended Data Fig. 7a) and variable expression of CD11c (Fig. 2c), leaving the broader role of RORγt in TC lineage specification unresolved. Analysis of RORγt protein in TC progenitors and subsets from E18.5 FL and P18 mLN of C57Bl/6 J mice revealed lowest expression in FL TLPs (Fig. 4a) with mLN TCPs spanning RORγt[lo] to RORγt[hi] cells, indicating progressive RORγt upregulation during TC development. Of the TC subsets, TC I expressed the lowest levels of RORγt, comparable to TLP, prompting closer examination of RORγt expression in TC I. Of the Lin−MHCII+ cells, TC I was distinguishable from cDCs by NCAM1 expression. Surprisingly, we found that around 40–50% of Lin−MHCII[hi]NCAM1+ cells in mLN

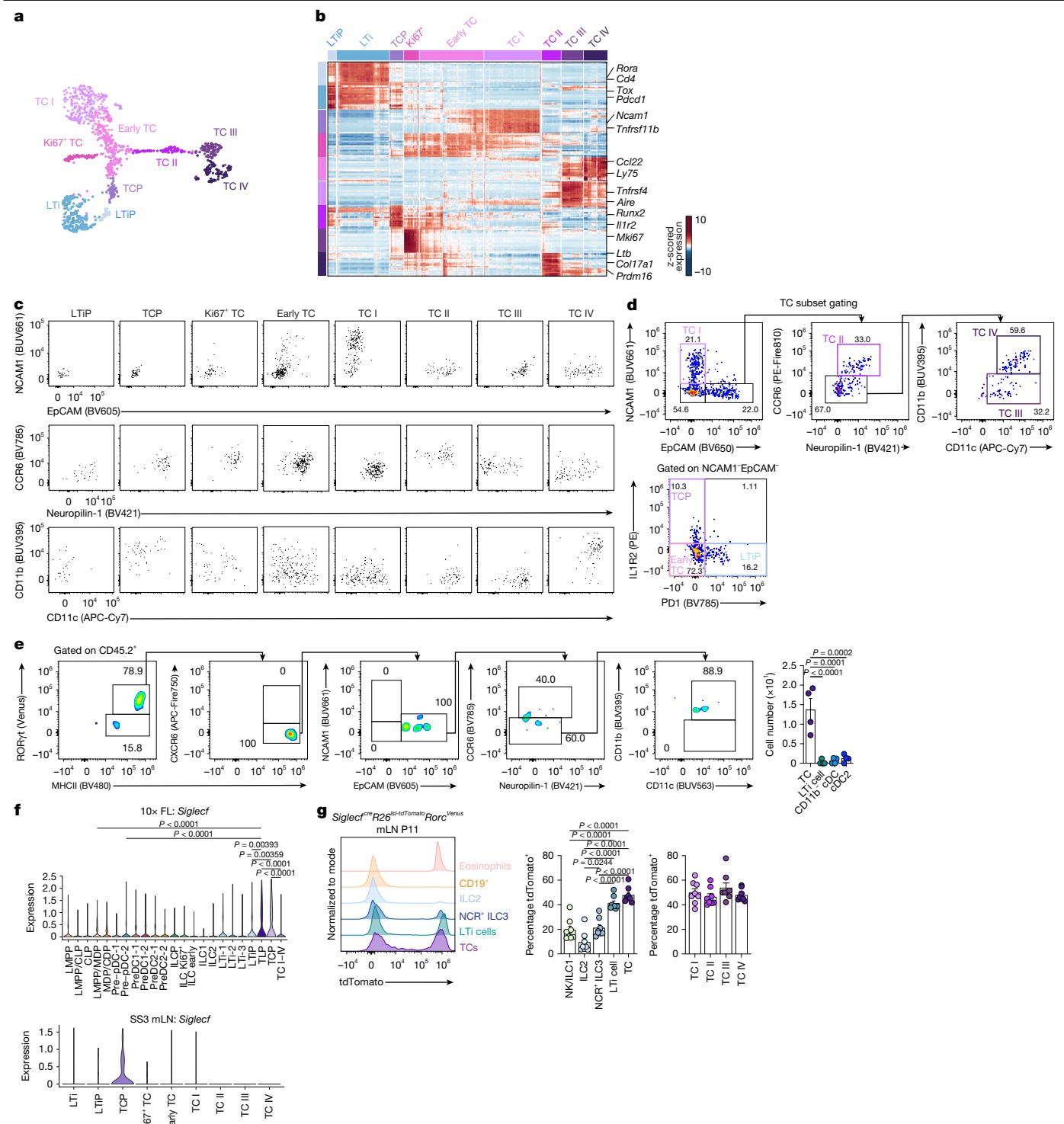

**Fig. 2 | Establishing precursor–progeny relationships for TLP and TCP.**
**a**, UMAP of 1,138 Lin⁻CXCR6⁻RORγt⁺MHCII⁺ cells from the mLN of 2-week-old
*Rorc^Venus* mice profiled by SS3. **b**, Heatmap reporting scaled, imputed expression
of the top 20 differentially expressed genes for each cluster (one versus the
rest, FC > 1.5, adjusted *P* < 0.01). **c**, Index sorting flow cytometry of cells in TC
clusters identified in **a**. **d**, Flow plots for identification of TCP and TC I–IV
subsets. Representative of *n* = 3 mice. **e**, TCPs and early TCs from the mLN of
P11–12 *Rorc^Venus* mice were cultured ex vivo on mLN slices from 2–3-week-old
CD45.1 mice. Representative flow cytometry of CD45.2⁺ progeny at 24 h and
summary bar graph of cell numbers recovered for indicated cell types are

shown (*n* = 4 mice). **f**, Expression of *Siglecf* in FL and TC progenitor clusters
identified in Fig. 1h or mLN clusters in **a**; SS3. **g**, Expression of tdTomato in
indicated cell types in mLN of P11 *Siglecf^cre R26^lsl-tdTomato Rorc^Venus* mice (*n* = 8)
and summary graph of frequency of tdTomato⁺ cells among the indicated
cell types (NCR⁺ ILC3, natural cytotoxicity receptor⁺ ILC3). Each symbol
represents an individual mouse. Data in **d** are representative of three
independent experiments; data in **e** are pooled from two independent
experiments; data in **g** are representative of two independent experiments.
Each symbol represents an individual mouse. Error bars: mean ± s.e.m.;
one-way ANOVA (**e**–**g**). All *P* values are indicated on the corresponding graphs.

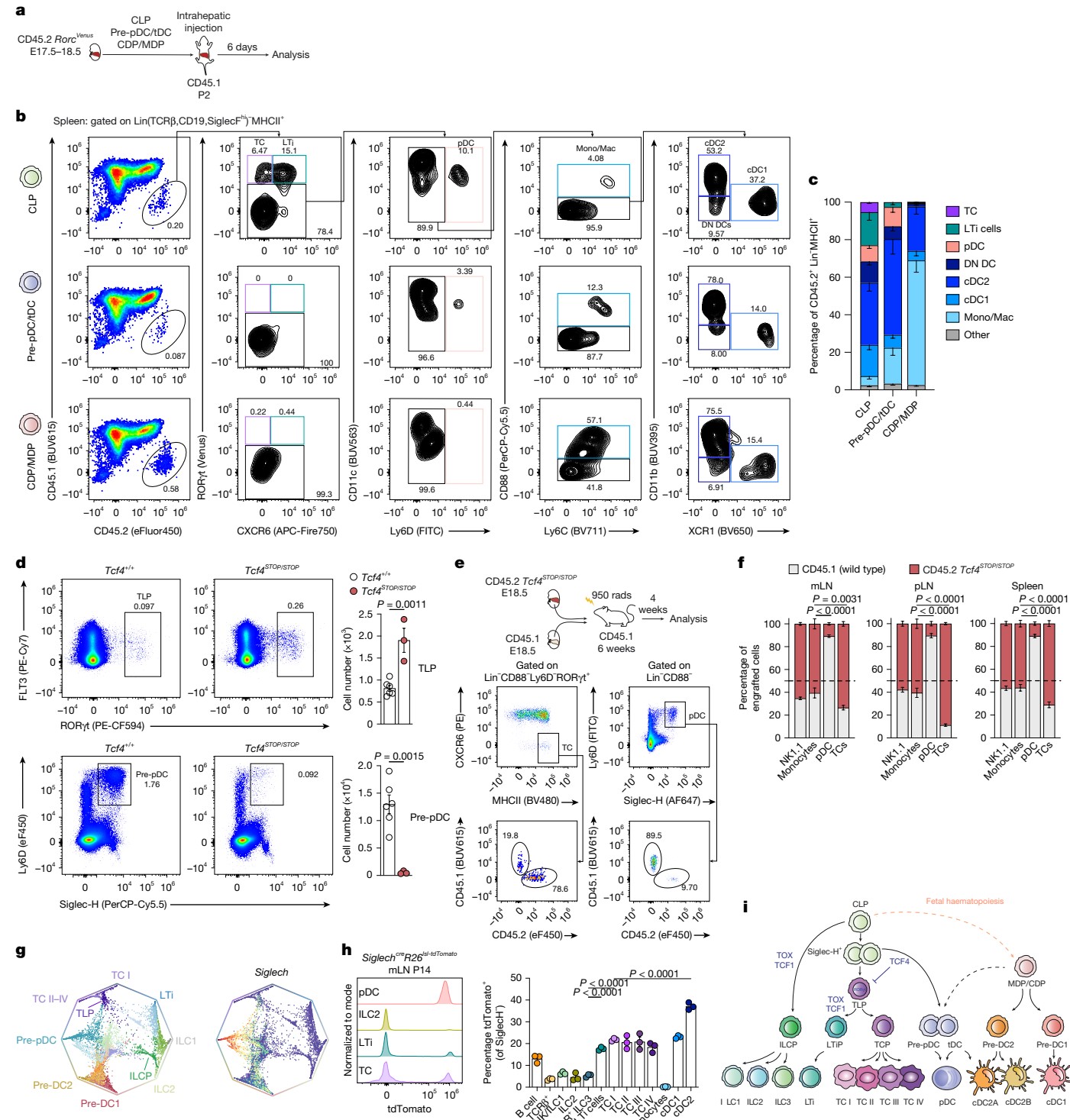

**Fig. 3 | TCs are derived from the CLP. a**, Schema for adoptive transfer of FL progenitors into neonatal recipients. **b**, Flow cytometry of splenic Lin⁻CD45.2⁺ APCs, 6 days after transfer of CLPs (top), pre-pDC/tDCs (middle) or CDP/MDPs (bottom) into CD45.1 P2 recipients. **c**, Frequency of APC subsets among splenic Lin(CD19, TCRβ, Siglec-F^hi)⁻CD45.2⁺MHCII⁺ cells, 6 days after transfer (*n* = 6 CLP recipients, *n* = 6 pre-pDC/tDC recipients and *n* = 3 MDP/CDP recipients). **d**, Representative flow plots and summary graphs for the absolute number of pre-pDCs and TLPs in E18.5 FL of *Tcf4*^STOP/STOP^ (*n* = 3) and littermate wild-type (*n* = 6) mice. **e**, Competitive chimeras were generated with a 1:1 mix of CD45.2⁺ *Tcf4*^STOP/STOP^ and CD45.1 wild-type E18.5 FL transplanted into lethally irradiated CD45.1 recipients and analysed 4 weeks later (*n* = 5 recipient mice). Representative flow plots of pDCs and TCs in mLN. **f**, Proportion of CD45.2 and CD45.1 cells among indicated cell types in mLNs, skin-draining pLNs and spleen (*n* = 5 mice). **g**, Polygon plot demonstrating cell-fate probabilities for the cells in **h**. Each dot is a cell and its position indicates the probability of reaching the terminal states on the vertices. Cells are coloured by cluster annotation (left) or expression of *Siglech* (right). **h**, Frequency of tdTomato⁺ cells among the indicated cell types in mLNs of P14 *Siglech*^cre^*R26*^lsl-tdTomato^ mice (*n* = 3). **i**, Schema delineating haematopoiesis and TC ontogeny based on our findings. Data in **c** are pooled from six independent experiments; data in **d**–**f** and **h** are representative of 2–3 independent experiments. Each symbol represents an individual mouse. Error bars: mean ± s.e.m.; two-way ANOVA (**d** and **f**) or one-way ANOVA (**h**). All *P* values are indicated on the corresponding graphs.

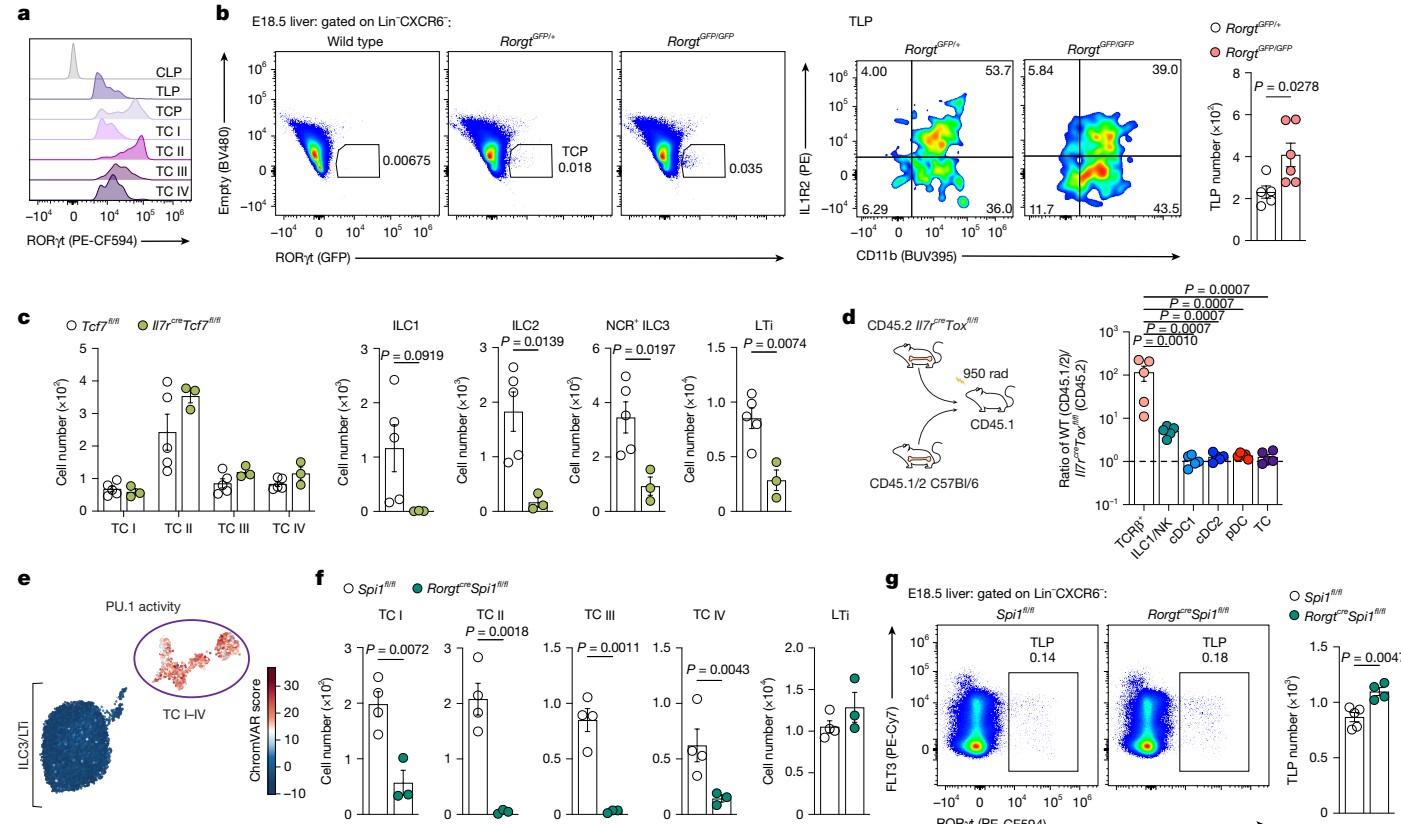

**Fig. 4 | PU.1 is required for TC I–IV differentiation. a**, Flow cytometry of RORγt expression among TC subsets from mLNs of P18 or TLPs from FL of E18.5 C57Bl/6 mice. Representative of *n* = 4 mice. **b**, Flow cytometry of GFP⁺ (RORγt 'wannabe') TLPs in FL of E18.5 *Rorgt^GFP/GFP* mice (*n* = 5) or littermate *Rorgt^GFP/+* controls (*n* = 6), and summary graph of TLP cell numbers. **c**, Number of TCs and ILCs in mLNs of 2-week-old *Il7r^cre^Tcf7^fl/fl* mice (*n* = 5) or littermate controls (*n* = 3). **d**, Mixed BM chimeras were generated with a 1:1 mix of CD45.2⁺ *Il7r^cre^Tox^fl/fl* and CD45.1/2 wild-type BM transplanted into lethally irradiated CD45.1 recipients and analysed 4 weeks later. The ratio of CD45.1.2 wild-type cells to CD45.2 Tox-deficient cells among indicated cell types is shown. **e**, UMAP of RORγt⁺MHCII⁺

cells profiled by scRNA/ATAC-seq[1] coloured by the chromVAR deviation score for the PU.1 motif. **f**, Number of TCs and LTi cells in mLNs of P14 *Rorgt^cre^Spi1^fl/fl* mice (*n* = 4) or littermate controls (*n* = 3). **g**, Representative flow cytometry of TLPs in FL of E18.5 *Rorgt^cre^Spi1^fl/fl* mice (*n* = 4) or littermate controls (*n* = 5), and summary graph of TLP numbers. Data in **a**, **c** and **g** are representative of two independent experiments; data in **d** and **f** are representative of three independent experiments; data in **b** are pooled from two independent experiments. Error bars: mean ± s.e.m.; two-tailed unpaired *t*-test (**b**, **c**, **e** and **g**) or one-way ANOVA (**d**). All *P* values are indicated on the corresponding graphs.

of P14 *Rorc^Venus^Aire^GreenLantern* mice lacked expression of RORγt(Venus) (Extended Data Fig. 7b). SS3-seq on RORγt(Venus)⁻NCAM1⁺MHCII⁺ cells confirmed their identity as bona fide TC I (Extended Data Fig. 7c). Analysis of *Rorc^Venus^Rorgt^cre^R26^lsl-tdTomato* fate-mapping mice demonstrated that most RORγt⁻ TC I lacked tdTomato expression (Extended Data Fig. 7d), indicating that the level or duration of RORγt expression in TLPs and TCPs is not sufficient to induce labelling of their RORγt⁻ TC I progeny. Given the low levels of RORγt in TC progenitors and TC I, we wondered whether RORγt was required for their differentiation. Analysis of E18.5 *Rorgt^GFP/GFP* (RORγt-null) mice revealed a modest increase in GFP⁺ TLP numbers relative to *Rorgt^GFP/+* mice (Fig. 4b), indicating that RORγt is dispensable for TLP specification. To circumvent the LN deficiency in *Rorgt^GFP/GFP* mice and assess TCP differentiation, we generated competitive CD45.2 *Rorgt^GFP/GFP* and CD45.1/2 *Rorc^Venus* BM chimeras, which yielded rare CD45.2⁺GFP^lo TCs, encompassing NCAM1⁻EpCAM⁻ TCPs or NCAM1⁺ TC I (Extended Data Fig. 7e,f). These findings demonstrate a graded requirement for RORγt in TC development, with progressive upregulation and dependence from RORγt^lo TLPs, TCPs and TC I to RORγt^hi TCPs and TC II–IV cells.

## PU.1 governs TC fate

We next wanted to understand the transcription factors that govern LTiP and TCP specification. LTi cell development is dependent on TCF1

(*Tcf7*) and TOX[48,49], transcription factors that are also expressed by the TLP (Extended Data Fig. 7g). However, TC subsets were unaffected in *Il7r^cre^Tcf7^fl/fl* mice, despite profound loss of ILC1–3 and LTi cells (Fig. 4c). Tox-deficient mice lack lymph nodes, so, to determine the role of TOX in TC development, we generated competitive BM chimeras with CD45.2 *Il7r^cre^Tox^fl/fl* and CD45.1/2 wild-type BM. Despite an almost complete absence of radiosensitive ILC/natural killer (NK) cells from CD45.2 BM, TC differentiation remained intact (Fig. 4d and Extended Data Fig. 7h). To identify the transcription factors that instruct TC fate, we used our published scRNA/ATAC-seq data on TCs and LTi cells[1]. ChromVAR analysis identified PU.1 (encoded by *Spi1*) as a candidate TC-specific transcriptional regulator (Fig. 4e). Analysis of *Spi1* expression in our scRNA-seq FL progenitor dataset (Fig. 1h) revealed high expression in TLPs, TCPs and TCs, with negligible expression in LTiPs (Extended Data Fig. 7i). Ablation of PU.1 in RORγt⁺ cells with *Rorgt^cre^Spi1^fl/fl* mice resulted in an almost complete absence of TC I–IV in P14 mLN, but no impairment in LTi cells (Fig. 4f). FL TLP numbers were not impaired in these mice (Fig. 4g), indicating that PU.1 is required for the later stages of TC differentiation. In contrast to its role in DCs, in which PU.1 is required for FLT3 expression[50], PU.1-deficient TLPs maintained FLT3 (Extended Data. Fig. 7j), indicating alternative mechanisms by which PU.1 promotes TC differentiation. Together, these findings identify PU.1 as a key regulator of TC I–IV differentiation.

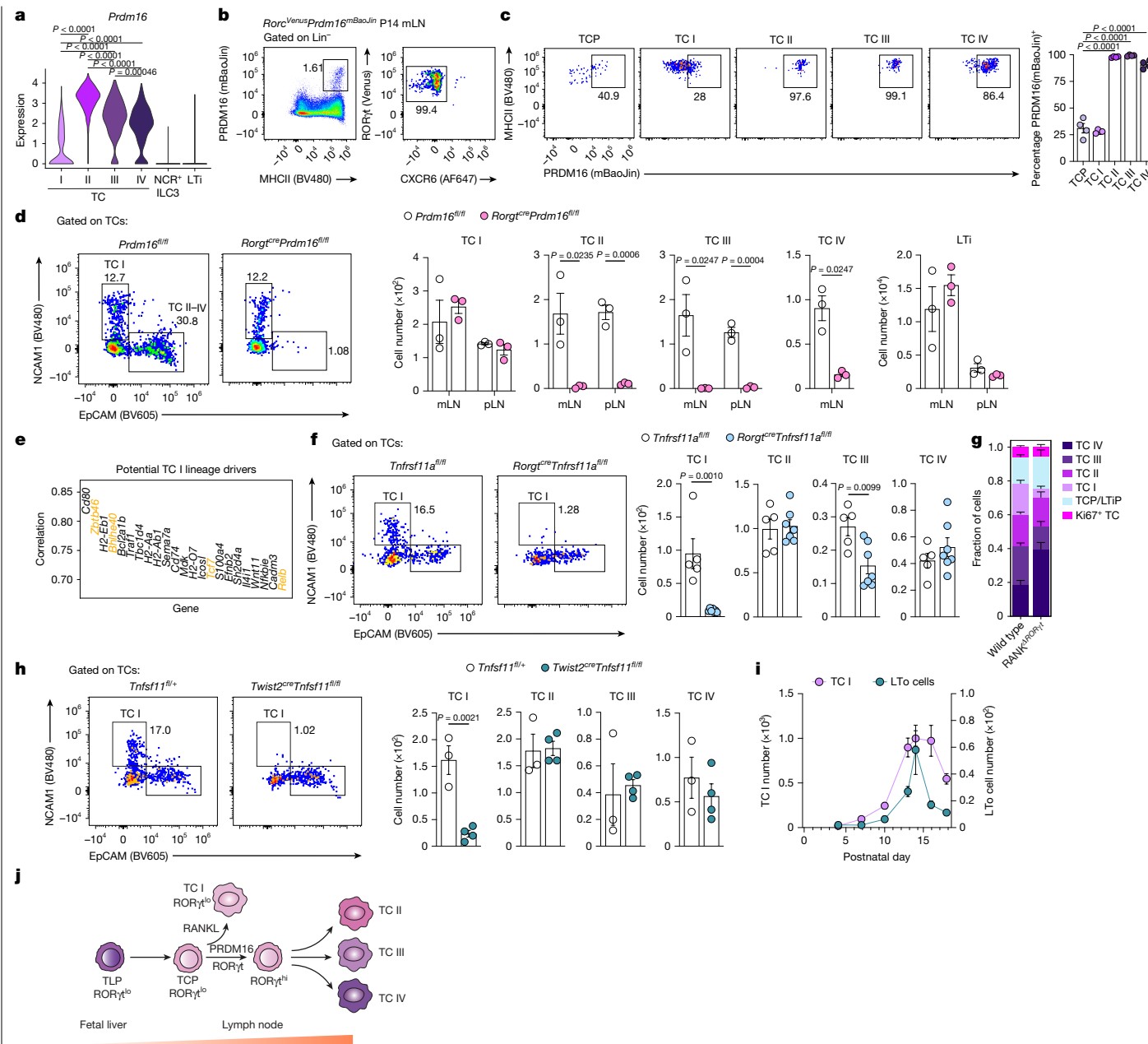

**Fig. 5 | RANK/NF-κB and PRDM16 drive TC heterogeneity. a**, Violin plot of *Prdm16* expression by the indicated RORγt⁺MHCII⁺ subsets profiled by scRNA/ATAC-seq[1]. **b**, Representative flow cytometry of PRDM16(mBaoJin)-expressing APCs in mLNs of P14 *Rorc^Venus^Prdm16^creERT2-mBaoJin^* mice (*n* = 4), gated on Lin(TCRβ, CD19, B220, Siglec-F^hi^, NK1.1)⁻. **c**, Representative flow cytometry of PRDM16(mBaoJin) expression by TC subsets in mLNs of P14 *Rorc^Venus^ Prdm16^creERT2-mBaoJin^* mice and summary graph (*n* = 4). **d**, Number of TCs in mLNs and pLNs of P14 *Rorgt^cre^Prdm16^fl/fl^* mice and littermate controls (*n* = 3 per group). **e**, CellRank 2-predicted driver genes of TC I cell fate. **f**, Representative flow cytometry of TCs in mLNs of P16 *Rorgt^cre^Tnfrsf11a^fl/fl^* mice (*n* = 7) or littermate controls (*n* = 5) and summary graph of TC subset numbers. **g**, Fraction of each TC subset among TCs isolated from mLNs of *Rorc^Venus^Rorgt^cre^Tnfrsf11a^fl/fl^* mice

(*n* = 6) or littermate *Rorc^Venus^Tnfrsf11a^fl/fl^* mice (wild type; *n* = 4) profiled by scRNA-seq. **h**, Representative flow cytometry of TCs in mLNs of P14 *Twist2^cre^ Tnfsf11^fl/fl^* mice (*n* = 4) or littermate controls (*n* = 3) and summary graph of TC subset numbers. Each symbol represents an individual mouse. **i**, Summary graph of TC I and CD45⁻CD31⁻gp38⁺VCAM1⁺ICAM1⁺ LTo cell numbers in mLNs of P4 to P18 *Rorc^Venus^* mice (*n* = 3, P4; *n* = 10, P7 and P16; *n* = 9, P10; *n* = 7, P13; *n* = 8, P14; *n* = 6, P18). **j**, Schematic delineating TC differentiation in LNs. Data in **b**–**d** are representative of three independent experiments; data in **f** and **h** are representative of two independent experiments; data in **i** are pooled from two independent experiments. Error bars: mean ± s.e.m.; one-way ANOVA (**a** and **c**) or two-tailed unpaired *t*-test (**d**, **f** and **h**). All *P* values are indicated on the corresponding graphs.

## Transcriptional regulation of TC heterogeneity

The wave of TC differentiation in early life indicates that fetal or early postnatal haematopoiesis favours TCP differentiation. Consistent with the dominant role of FL haematopoiesis in LTi cell differentiation[32], TLPs were enriched in E17.5 FL relative to adult (8-week-old)

BM from *Rorc^Venus^* mice (Extended Data Fig. 8a,b). However, our earlier data showed that differentiated TCs are not present until a few days after birth, indicating that postnatal cues promote the diversification of TCPs into four distinct subsets. To identify such cues, we focused on transcription factors expressed by TC subsets but not by TCPs. Our earlier analyses had revealed one such factor, the transcriptional

regulator PRDM16 (Fig. 2b and Extended Data Fig. 4a), which has been shown to regulate non-ILC3 RORγt[+] APC differentiation[3]. Our published scRNA-seq analysis of RORγt[+] APCs[1] demonstrated PRDM16 expression in TC II–IV, but low or negligible expression in TC I (Fig. 5a). To examine this further, we inserted a sequence encoding creERT2-T2A-mBaoJin in the 3′ untranslated region (UTR) of *Prdm16* (Extended Data Fig. 8c). Analysis of P14 *Prdm16*[mBaoJin]*Rorc*[Venus] dual-reporter mice revealed that PRDM16[+]MHCII[+] cells represented TCs across all the lymph nodes and tissues examined (Fig. 5b and Extended Data Fig. 8d,e). Among TC subsets, PRDM16(mBaoJin) was highly expressed by TC II–IV but was low or absent in TCP or TC I (Fig. 5c and Extended Fig. 8f). In line with this pattern of expression, TC II–IV were almost completely absent in the mLNs and pLNs of *Rorgt*[cre]*Prdm16*[fl/fl] mice, with no impairment in TC I or LTi cells (Fig. 5d). Consistent with the loss of TC IV, these mice exhibited impaired intestinal pT$_{reg}$ differentiation and dysregulated T helper 17 (T$_H$17) cell responses, confirming the crucial role for TC IV in intestinal tolerance (Extended Data Fig. 8g).

The bifurcation in TC I compared with TC II–IV fate commitment indicated that distinct cues promote TC I fate. Orthogonal CellRank 2 analysis of scRNA-seq data (Fig. 1e) and chromVAR analysis of our published scATAC-seq dataset[1] identified the NF-κB transcription factor RELB as a candidate regulator of TC I differentiation (Fig. 5e and Extended Data Fig. 9a). TC I exhibit transcriptional overlap with AIRE[+] mTECs; thus, RELB was intriguing, given that AIRE[+] mTECs are dependent on RANKL-induced non-canonical NF-κB signalling[51–53], suggesting that there are parallel pathways for AIRE[+] mTEC and TC I differentiation. Analysis of RANK (*Tnfrsf11a*)–GFP expression by TCs in mLNs of P16 *Tnfrsf11a*[GFP] mice revealed the highest levels in TC I (Extended Data Fig. 9b). Genetic ablation of RANK signalling on TCs in *Rorgt*[cre] *Tnfrsf11a*[fl/fl] mice led to a complete absence of TC I and a modest reduction in TC III (Fig. 5f). Using scRNA-seq on TCs from mLNs of 2-week-old *Rorc*[Venus]*Rorgt*[cre]*Tnfrsf11a*[fl/fl] or littermate *Rorc*[Venus]*Tnfrsf11a*[fl/fl] mice confirmed the loss of TC I, as well as a reduction in the proportion of TC III, but no change in TCP proportions (Fig. 5g and Extended Data Fig. 9c,d). Competitive chimeras generated with a 1:1 mix of CD45.2 *Rorgt*[cre] *Tnfrsf11a*[fl/fl] and CD45.1/2 wild-type BM showed preferential derivation of TC I from wild-type BM (Extended Data Fig. 9e), demonstrating a cell-intrinsic requirement for RANK signalling.

To determine the cellular source of RANKL that promotes TC I differentiation, we used an scRNA-seq atlas of the human gut across development[54]. This revealed two candidate RANKL-expressing cell types in developing lymph nodes: LTi cells and lymphoid tissue organizer (LTo) cells (Extended Data Fig. 9f), a key stromal cell type that acts in concert with LTi cells to promote the development of lymph nodes. Ablation of RANKL in LTi cells using *Rorgt*[cre]*Tnfsf11*[fl/fl] mice did not affect TC numbers (Extended Data Fig. 9g). By contrast, ablation of RANKL expression in LTo cells using *Twist2*[cre]*Tnfsf11*[fl/fl] mice led to an almost complete absence of TC I (Fig. 5h). Parallel analysis of LTo cell and TC I abundance in mLN of *Rorc*[Venus] mice during the first three weeks of life revealed similar developmental kinetics, indicating that developmentally restricted LTo cells and their associated signals could determine the window for TC I differentiation (Fig. 5i). Together, these findings identify spatiotemporally restricted cues and downstream transcription factors that drive the differentiation of TCPs into distinct subsets (Fig. 5j).

## TC subsets are terminally differentiated cells

Finally, we investigated whether TC subsets represent terminally differentiated or intermediate cell states. Using *Prdm16*[creERT2-mBaoJin]*R26*[lsl-tdTomato] mice to temporally label TC II–IV, we analysed tdTomato expression in APCs after alternate-day 4-OHT treatment from P7–14. TC II–IV were the only cell types labelled with tdTomato (Extended Data Fig. 10a), and nearly all tdTomato[+] cells retained RORγt expression (Extended Data Fig. 10b), indicating that TC II–IV do not convert to non-RORγt[+] APCs or TC I. To assess TC plasticity further, we generated

an *Aire*[GreenLantern-creERT2] allele in which GreenLantern-creERT2 is inserted into the endogenous 3′ UTR (Extended Data Fig. 10c). Analysis of dual-reporter *Rorc*[Venus]*Aire*[GreenLantern-creERT2] mice confirmed that there was restricted expression of GreenLantern in TC I and III subsets (Extended Data Fig. 10d). Fate-mapping of AIRE[+] TC subsets in *Aire*[GreenLantern-creERT2] *R26*[lsl-tdTomato] mice treated with 4-OHT from P7–18 resulted in labelled AIRE[+] TC I and TC III, albeit with reduced efficiency in TC III (Extended Data Fig. 10e). Importantly, AIRE[−] TC II and IV and cDCs were not labelled. Collectively, these findings indicate that TC subsets represent stable cell states.

## Discussion

Tolerogenic APCs, proposed more than 20 years ago[55], have enormous therapeutic promise for food allergies, autoimmunity and transplantation. The discovery of TC IV, a dedicated APC for pT$_{reg}$ cell generation, offers a tangible route towards this long-sought goal. However, the sharp post-weaning decline in TC numbers, if it is mirrored in humans, poses challenges for clinical translation. Identification of TC precursors, their developmental pathway and the cues that promote their differentiation creates opportunities to generate TC subsets in vivo and ex vivo and to explore the functions of the less-understood TC subsets.

Our findings refine the current models of haematopoiesis by charting TC development through two previously uncharacterized RORγt[+] progenitors, TLPs and TCPs, revealing unanticipated lineage relationships. Since the discovery of TCs, several studies have reported transcriptionally overlapping RORγt[+] APC populations and proposed reclassifying TCs as DCs on the basis of shared transcriptional features[2,3,7,11,41]. However, because cell morphology and gene expression are not fixed, consensus guidelines prioritize ontogeny for DC classification, defining DCs by CDP origin[19]. Accordingly, pDC nomenclature has been widely debated following reports of a predominantly lymphoid origin[8,23,56]. Despite transcriptional overlap with cDCs, we show that TCs represent a distinct APC lineage that shares developmental origins with lymphoid-derived LTi cells. Expansion of TLPs in the absence of TCF4 indicates a shared early developmental branch with pDC-primed progenitors, although further studies are required to address a cell-intrinsic role of TCF4 in TLP versus pre-pDC fate bifurcation.

Our results indicate that the early-life window for TC development is determined in part by developmentally restricted progenitors, providing insight into the temporal regulation of intestinal tolerance. Although the bipotent nature of TLPs at the single-cell level remains to be determined, a possible common progenitor for LTi cells and TCs is intriguing, given their divergent phenotypes and functions. In contrast to TCs, LTi cells have limited roles in T cell priming but are essential for lymphoid organogenesis. A shared TC–LTi cell developmental program therefore provides an elegant mechanism linking lymph-node development to the emergence of tolerogenic TCs, ensuring tolerance to the first-encountered intestinal antigens.

Overall, these studies delineate the developmental pathway for TCs, defining molecular signatures and transcriptional regulators from early specification in the FL and BM to lineage commitment in lymph nodes. Given the central role of TCs in tolerance to food and microbiota antigens, these findings open new avenues to explore their immune-regulatory functions and therapeutic potential in food allergies, autoimmunity and transplantation.

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

## Methods

### Mice

$Aire^{GreenLantern\text{-}T2A\text{-}creERT2}$ mice were generated by inserting a targeting construct into the $Aire$ 3′ UTR by homologous recombination in embryonic stem cells on an $F_1$ 129:C57Bl/6J background. The IRES-GreenLantern-T2A-creER-frt-NeoR-frt cassette targeting construct was created by cloning. Homologous arms were retrieved from BAC clone RP24-209K20. To facilitate embryonic stem cell targeting, a CRISPR–Cas9 system was used. The gRNA was in vitro transcribed using a MEGA shortscript T7 kit (Life Tech, AM1354) using recombineering techniques. The targeting vector, Cas9 protein (Fisher Scientific A36498 Truecut Cas9 Protein v2) and gRNA were co-electroporated into G1 embryonic stem cells derived from an $F_1$ hybrid blastocyst of 129S6 × C57BL/6J. The resulting chimeras were bred with FLPeR mice to excise the NEO cassette, backcrossed to C57Bl/6J for more than six generations and confirmed by SNP analysis to be more than 99.9% C57Bl/6J.

Prdm16-P2A-iCreERT2-T2A-mBaoJin KI mice (referred to as $Prdm16^{creERT2\text{-}mBaoJin}$) were generated by Biocytogen using CRISPR–Cas9-based extreme genome editing technology. A single sgRNA was designed to target exon 16 of the $Prdm16$ gene. A targeting vector was constructed in the TV-4G vector that contained the 5′ homologous arm (about 1.2 kb), the exon 16-P2A-iCreERT2-T2A-mBaoJin and the 3′ homologous arm (about 1.2 kb). The targeting vector DNA, Cas9 mRNA and sgRNA were microinjected into zygotes derived from C57BL/6J mice and then the embryos were transferred to surrogate mothers. The P2A-iCreERT2-T2A-mBaoJin cassette was inserted into exon 16, upstream of the $Prdm16$ 3′ UTR. $F_0$ positive founders were bred with wild-type C57BL/6J mice to obtain $F_1$ pups. The $F_1$ germline-transmission heterozygous mice were verified by PCR, DNA sequencing and Southern blot with 5′ and 3′ ends to exclude random insertions.

$Rorc^{Venus\text{-}creERT2}$ (also referred to as $Rorc^{Venus}$), $Il7r^{cre}$, $Spi1^{fl/fl}$, $Tox^{fl/fl}$, $Tcf7^{fl/fl}$, $Tnfrsf11a^{cre\text{-}GFP}$, $Tnfrsf11a^{fl/fl}$, $Tnfsf11^{fl/fl}$, $Twist2^{cre}$, $Prdm16^{fl/fl}$, R26:FlpoERT2 $Flt3^{Frt\text{-}ITD}$, $Siglech^{cre}$ and $Tcf4^{STOP}$ mice have been described previously[1,25,57–66]. $Rorgt^{cre}$, $Rorgt^{GFP}$, $R26^{lsl\text{-}tdTomato}$ (Ai14), $Cx3cr1^{cre}$, C57BL/6 (CD45.2) and CD45.1 (PtprcK302E) mice were purchased from Jackson Laboratories. $Siglecf^{cre}$ mice (NM-KI-231128) were purchased from Shanghai Model Organisms. Generation and treatments of mice were performed under protocol 21-05-007, approved by the Sloan Kettering Institute Institutional Animal Care and Use Committee. All mouse strains were maintained in the Sloan Kettering Institute animal facility in specific pathogen-free conditions in accordance with institutional guidelines and ethical regulations. Both male and female mice were included in the study, and we did not observe any sex-dependent effects. All mice analysed were age- and litter-matched, unless otherwise specified. The age of mice at the time of analysis is included in the figures, legends and main text for individual experiments. No experimental interventions requiring randomization to control or experimental groups were performed in this study. For steady-state immune phenotyping, fate-mapping experiments or analysis of transcription factor-deficient mice, all the mice in a litter were analysed and no blinding was done. No statistical calculations were done to determine sample size. The sample size for bone-marrow or fetal-liver chimera experiments was decided based on preliminary data and previous publications with similar experiments[1,5,49].

### Tissue processing

Mice were killed by $CO_2$ inhalation. Organs were collected and processed as follows. Lymphoid organs, embryonic and postnatal liver were digested in collagenase in RPMI1640 medium supplemented with 5% fetal bovine serum, 1% L-glutamine, 1% penicillin–streptomycin, 10 mM HEPES, 1 mg ml$^{-1}$ collagenase A (Sigma, 11088793001) and 1 U ml$^{-1}$ DNase I (Sigma, 10104159001) for 45 min (lymph nodes) or 30 min (liver) at 37 °C, 250 r.p.m. The large intestine was removed, flushed with PBS and incubated in PBS supplemented with 5% fetal bovine serum, 1% L-glutamine, 1% penicillin–streptomycin, 10 mM HEPES, 1 mM dithiothreitol and 1 mM EDTA for 15 min to remove the epithelial layer. Samples were washed and incubated in digest solution for 30 min. Then 0.635-cm ceramic beads (MP Biomedicals, 116540034) were added to large-intestine samples (three per sample) to help with tissue dissociation. Digested samples were filtered through 100-μm strainers and centrifuged to remove collagenase solution. Postnatal liver single-cell suspensions underwent another density-gradient centrifugation using a 40% Percoll gradient. For flow-cytometry analysis of fetal liver progenitors, single-cell suspensions of digested liver were depleted of lineage-positive (CD19, NK1.1, TCRβ, TCRγδ, Ter-119, Ly-6G, CD3e, FceR1a, CD90.2 and CD88) cells by staining with biotinylated antibodies followed by magnetic bead (Dynabeads M-280 streptavidin; Invitrogen) negative selection as per the manufacturer's instructions. Bone-marrow cells were isolated by centrifugation, depleted of lineage-positive (CD19, NK1.1, TCRβ, TCRγδ, Ter-119, Ly-6G, CD3e, FceR1a, CD90.2 and CD88) cells by staining with biotinylated antibodies followed by magnetic bead (Dynabeads M-280 streptavidin; Invitrogen) negative selection as per the manufacturer's instructions. For parallel flow-cytometry analysis of TC and LTo cells, lymph nodes were digested in HBSS without calcium and magnesium, supplemented with 2% fetal bovine serum, 2 mM calcium chloride (Sigma, C1016), 10 mM HEPES, 0.2 mg ml$^{-1}$ collagenase A (Sigma, 11088793001), 50 μg DNase I (Sigma, 10104159001) and 0.8 mg ml$^{-1}$ Dispase II (Sigma, D4693) for 20 min at 37 °C, followed by mechanical dissociation of any remaining tissue. Digested samples were quenched with quenching buffer (PBS, 5% fetal bovine serum and 5 mM EDTA) and then filtered through 100-μm strainers and centrifuged to remove collagenase solution.

### Flow cytometry

For flow-cytometric analysis, dead cells were excluded by staining with LIVE/DEAD Fixable Zombie NIR in PBS for 10 min at 4 °C with inclusion of anti-CD16/32 to block binding to Fc receptors before cell-surface staining. Extracellular antigens were stained for 30 min at room temperature in staining buffer (2% FBS, 0.1% Na azide, in PBS), diluted 1:1 with Brilliant Violet (BD Biosciences) staining buffer. For intracellular protein analysis, cells were fixed and permeabilized with Cytofix (BD Biosciences) and/or an Ebioscience Foxp3 kit, as per the manufacturer's instructions. Intracellular antigens were stained for 30 min or overnight at 4 °C in the respective 1× Perm/Wash buffer. The antibodies used for flow cytometry and FACS are listed in Supplementary Table 3. Unless otherwise stated, we used the following gatings: TCs, Lin (Siglec-F$^{hi}$, TCRβ, TCRγδ, CD19, B220, NK1.1, CD88, Ly6C)$^-$RORγt$^+$CXCR6$^-$MHCII$^+$; LTis, Lin$^-$RORγt$^+$ CXCR6$^+$CCR6$^+$MHCII$^+$; cDC2s, Lin$^-$RORγt$^-$CD11c$^+$ MHCII$^+$CD11b$^+$XCR1$^-$; cDC1s, Lin$^-$CD88$^-$Ly6C$^-$RORγt$^-$CD11c$^+$MHCII$^+$ CD11b$^-$XCR1$^+$; and FL TLPs, Lin (TCRβ, TCRγδ, CD19, B220, NK1.1, CD88, Ter-119, Ly6G, CD3e, FceR1a, CD90.2)$^-$CXCR6$^-$RORγt$^+$. Example flow plots for the TC subsets and TLP gating are shown in Figs. 2d,e and 1p, respectively. Samples were acquired on a Cytek Aurora. FACS isolation was done using a Cytek Aurora Cell Sorter or BD Aria.

### BM chimera mice

BM cells were isolated from the indicated donor mice and depleted of CD90.2$^+$ and TER-119$^+$ cells using magnetic bead-based depletion. BM cells were resuspended in PBS and a total of $2 × 10^6$ to $5 × 10^6$ cells were injected into 6-week-old recipient mice that had been irradiated with 950 rad per mouse one day earlier. Mice were analysed four weeks after reconstitution unless otherwise stated.

### FL chimera mice

FL cells were isolated as described above from the indicated donor mouse strains and ages. The FL cells were resuspended in PBS and a total of $2 × 10^6$ to $5 × 10^6$ cells were injected into 6-week-old recipient mice that had been irradiated with 950 rad per mouse one day earlier. Mice were analysed four weeks after reconstitution unless otherwise

stated. For competitive FL and BM chimeras, equivalent numbers of cells from each donor were injected into recipient mice.

## 10x Genomics Flex scRNA-sequencing

The mLN and liver from P7 $Rorc^{Venus}$ mice ($n = 9$), mesenteric anlagen, FL and fetal BM from E17.5–18.5 $Rorc^{Venus\text{-}creERT2}$ mice ($n = 25$), mLN, liver and BM from P7 $Il7r^{cre}R26^{lsl\text{-}tdTomato}$ mice ($n = 4$) and mesenteric anlagen, FL and fetal BM from E17.5–18.5 $Il7r^{cre}R26^{lsl\text{-}tdTomato}$ mice ($n = 9$) were processed for Flex scRNA-seq analysis of progenitors (Supplementary Table 1). FL, liver, fetal BM and BM were depleted of lineage (TCRβ, TCRγδ, CD19, B220, NK1.1, Ter-119, Ly-6G, CD3e)$^+$ cells by staining with biotinylated antibodies followed by magnetic bead (Dynabeads M-280 streptavidin; Invitrogen) negative selection. Cells were incubated with anti-CD16/32, and extracellular antigens were stained for 30 min at room temperature in sorting buffer (2% FBS, 2 mM EDTA, in PBS) and labelled with BioLegend TotalSeq-C Hashtag antibodies. Cells were washed and resuspended in cRPMI with SYTOX blue (Invitrogen) to exclude dead cells. Cells were sorted as per Supplementary Fig. 1 into 500 µl of fixation buffer (4% formaldehyde, 1× Conc. Fix and Perm Buffer; 10x Genomics, PN2000517). Cells were fixed for 16–20 h at 4 °C. To stop the fixation, cells were spun down at 850$g$ for 5 min at room temperature and quenched with 500 µl of Quenching Buffer (1× Conc. Quench Buffer; 10x Genomics, PN-2000516). Preparations were then processed by adding 0.1 volumes of Enhancer (10x Genomics, PN2000482) and 10% glycerol for storage at −80 °C. Samples were then thawed at room temperature, centrifuged at 850$g$ for 5 min and resuspended in 1 ml 0.5× PBS, 0.02% BSA. Cell concentration and viability were assessed by 0.2% (w/v) Trypan Blue staining (Countess II). Cells were processed per hybridization according to the 10× protocol. Hybridizations were set up in 80 µl of hybridization mix with 20 µl of Mouse WTA probes (10x Genomics, PN-2000718) and 4 µl of Antibody Multiplexing Barcode (10x Genomics, PN-2000917- 2000932). Hybridizations were done at 42 °C for 16–24 h. After hybridization, samples were diluted in Post-Hyb wash buffer and measured by 0.2% (w/v) Trypan Blue staining (Countess II). For each experiment, we pooled an equal number of cells from each hybridization to have an equal contribution per sample. Pooled cells were then washed three times in Post-Hyb wash buffer for 10 min at 42 °C, resuspended in Post-Hyb resuspension buffer, filtered through a Miltenyi Biotec 30 µm filter and measured with the cell counter to determine the amount required for the Chromium X run. GEM encapsulation was done according to the 10x Genomics protocol. After loading the Chip Q and running it on the Chromium X, GEMs were recovered and processed as indicated by 10x Genomics. After processing, the product was preamplified and indexed to construct the sequencing library. All libraries were sequenced on an Illumina Novaseq X with standard dual indexing and demultiplexing.

## Plate-based Smart-seq3 sequencing

RORγt$^+$CXCR6$^-$MHCII$^+$ cells and RORγt$^-$MHCII$^+$NCAM1$^+$ cells were FACS-isolated from mLN of 2-week-old $Rorc^{Venus}$ mice ($n = 14$). Lymph nodes were processed as outlined above. Cells were depleted of lineage (TCRβ, TCRγδ, CD19, B220, NK1.1, Ly6G)$^+$ cells by staining with biotinylated antibodies followed by magnetic bead (Dynabeads M-280 streptavidin; Invitrogen) negative selection. Live Lin(TCRβ, TCRγδ CD19, B220, NK1.1, Ly6G, SiglecF$^{hi}$)$^-$CD88$^-$Ly6C$^-$MHCII$^+$RORγt(Venus)$^+$ CXCR6$^-$ or Lin$^-$CD88$^-$Ly6C$^-$MHCII$^+$RORγt(Venus)$^-$NCAM1$^+$ cells were then sorted into single wells. Cells were also stained for EpCAM, CD11c, CD11b, CCR6 and NRP1 for acquiring index sorting information on cell-surface expression. Cells were sorted into 3 µl Lysis Master Mix consisting of PEG 8000, Triton X-100, dNTPs, Oligo dT30VN and RNase inhibitor in 384-well plates. First-strand cDNA synthesis was done using Maxima H Minus Reverse Transcriptase (ThermoFisher, EP0751) according to the manufacturer's protocol, using a custom template-switch oligonucleotide in a 2 µM final concentration. Then cDNA was amplified for 23 cycles using KAPA HiFi HotStart ReadyMix (Kapa Biosystems, KK2601)

and cleaned up using aMPure XP beads (Beckman Coulter, A63882) at a 0.6× ratio. Full-length cDNA was tagmented using Tn5 (Illumina, 20034198) for 10 min at 55 °C and libraries were prepared with the Nextera XT DNA Library Preparation Kit (Illumina, FC-131-1024) in a total volume of 6.25 µl with 12 cycles of PCR. Indexed libraries were pooled by volume, according to plate quadrant, and cleaned by aMPure XP beads (Beckman Coulter, A63882) at a 0.8× ratio. Pools were sequenced on a NovaSeq X in a PE100 run using a NovaSeq X 10B Reagent Kit (Illumina). An average of 551,000 paired reads were generated per cell.

## 10x 3′ scRNA-sequencing v.3.1

RORγt$^+$CXCR6$^-$MHCII$^+$ cells were FACS-isolated from mLN of 2-week-old (P16) $Rorc^{Venus}Tnfrsf11a^{fl/fl}$ ($n = 4$ biological replicates) and $Rorc^{Venus}$ $Rorgt^{cre}Tnfrsf11a^{fl/fl}$ mice ($n = 6$ biological replicates). Lymph nodes were processed as outlined above. Each biological replicate single-cell suspension was hashtagged with BioLegend TotalSeq-B Hashtag antibodies. Live Lin(TCRβ, TCRγδ, CD19, NK1.1, SiglecF$^{hi}$)$^-$CD64$^-$Ly6C$^-$ CD90.2 + MHCII$^+$RORγt(Venus)$^+$CXCR6$^-$ cells were then sort purified. Cells were sorted into cRPMI-20% FBS before being pelleted and resuspended in cRPMI-2% FBS. Then scRNA-seq of FACS-sorted cell suspensions was done on a Chromium instrument (10x Genomics) following the user guide manual for 3′ v.3.1. In brief, cells were washed once with PBS containing 1% BSA and resuspended in PBS containing 2% BSA to a final concentration of 700–1,300 cells per microlitre. Cell viability was confirmed to be greater than 80% by 0.2% (w/v) Trypan Blue staining (Countess II). Cells were captured in droplets. After reverse transcription and cell barcoding in droplets, emulsions were broken, and cDNA was purified using Dynabeads MyOne SILANE followed by PCR amplification, as per the manufacturer's instructions. Around 10,000 cells were targeted. Samples were multiplexed on one lane of 10x Chromium using HashTag oligonucleotides (HTO) following a previously published protocol[67]. Libraries were sequenced on an Illumina Novaseq X with standard dual indexing and demultiplexing.

## Single-cell RNA-seq and single-cell ATAC-seq computational analysis

**Preprocessing of the 10x Flex scRNA-seq.** The scRNA-seq and HTO FASTQ files were aligned to mm10 (Cell Ranger mouse reference genome mm10-2020-A) and HTO barcodes, and were counted and demultiplexed using Cell Ranger v.8.0.0 multi to generate RNA and HTO count matrices for each sample. Each sample was further demultiplexed based on HTO counts using HTODemux function in Seurat v.4.4.0. Two samples (Flex_15 and Flex_16) formed a separate TC analysis and were not included in the downstream analysis. RNA count matrices for individual samples were first merged into a single count matrix. Next, barcodes classified as singlets from HTO demultiplexing were further filtered, based on the number of RNA-seq transcripts (more than 1,000 and fewer than 60,000), the number of detected genes (more than 500 and fewer than 8,000) and the fraction of mitochondrial transcripts (less than 10%). Finally, any genes detected in fewer than two cells in the scRNA-seq data were discarded. After clustering (described in 'Dimensionality reduction, cell clustering and visualization') and analysis of quality control metrics and differentially expressed genes, we identified 10 minor contaminant or low-quality control clusters (fibroblast, cluster 54; hepatocyte, cluster 62; osteoclast, cluster 56; mixed/undefined, 42, 49 and 59; low quality, clusters 25, 44, 57 and 61), which were excluded from downstream analyses. In total, 101,192 cells remained, with a median scRNA-seq library-size of 10,314 from 17,330 genes. Cluster labels were manually assigned and curated based on expressed genes identified previously[1,22,68,69], as well as cell-type annotation assigned by SCimilarity[70] (v.0.3.0) and CellTypist[71] (v.1.5.3).

**Preprocessing of the 10×3′ scRNA-seq.** Samples were demultiplexed as per the method above. Barcodes classified as singlets from HTO demultiplexing were further filtered, based on the number of RNA-seq

transcripts (more than 1,800), the number of detected genes (more than 600) and the fraction of mitochondrial transcripts (less than 8%). Finally, any genes detected in fewer than two cells in the scRNA-seq data were discarded. After clustering (described in 'Dimensionality reduction, cell clustering and visualization') and analysis of quality-control metrics and differentially expressed genes, we identified 7 minor contaminant or low-quality control clusters (B cells, cluster 0; T cells, clusters 3 and 9; CCR7+ DCs, cluster 11; cDC2s, cluster 14; low quality, clusters 13 and 15), which were excluded from downstream analyses. In total, 1,505 cells remained, with a median scRNA-seq library size of 13,768 from 20,103 genes.

**Preprocessing of the Smart-seq3 dataset.** Smart-seq3 FASTQ files from individual plate wells were first combined into single FASTQ files. The combined FASTQ files were aligned to the mouse reference genome (GRCm38.p6 from GENCODE release M25) and counted using zUMIs v2.9.7e with STAR v.2.7.11a. Plates 5 and 6 were not included in downstream analysis owing to low-quality control metrics. A further 45 cells isolated from the large intestine in plates 1–4 were excluded. The remaining barcodes were filtered on the basis of the number of RNA-seq transcripts (more than 1,500), the number of detected genes (more than 1,000) and the fraction of mitochondrial transcripts (less than 5%). Any genes detected in fewer than two cells were discarded. In total, 1,260 cells remained, with a median library size of 16,606 from 27,543 genes.

**Dimensionality reduction, cell clustering and visualization.** For each scRNA-seq dataset, the filtered count matrix was library-size normalized, log-transformed (log-normalized expression values) and then centred and scaled (scaled expression values) using Seurat v.4.4.0 (Flex). Principal component analysis was done on the scaled data (npcs = 50). A nearest-neighbour graph was constructed using the first 30 principal components with 30 nearest neighbours. Clustering was performed using the Louvain algorithm (resolutions 2 for Flex) on the shared nearest-neighbour graph. For the 3' scRNA-seq data, analysis was done using Scanpy as per the steps above, with the first 50 principal components and 25 nearest neighbours. Cell clustering was visualized using UMAP computed from the same nearest-neighbour graph as that used for clustering. For the Smart-seq3 data, batch correction of cells from plates 1–4 was done using Seurat's anchor-based canonical correlation analysis integration method. Filtered count matrices from each plate were first library-size normalized and log-transformed. Integration anchors were then identified using the FindIntegrationAnchors function with 5,000 anchor features, followed by data integration using the IntegrateData function. The integrated expression matrix was scaled, used for principal component analysis, clustered and visualized using UMAP as described above, with the first 50 principal components, 30 nearest neighbours and a clustering resolution of 0.6.

### Differential gene-expression tests

Differentially expressed genes (DEGs) between groups of cells were identified by MAST[72], performed using Seurat functions. MAST was run on the log-normalized expression values. In all tests, genes were considered only if they were detected in at least 1% of the cells in at least one of the two groups compared (min.pct = 0.01, logfc.threshold = 0). In one-versus-rest DE tests comparing multiple groups, each group was compared with all the cells from other groups. Specific DE comparisons are described in the results. DEGs were reported according to their log-transformed fold change (greater than 1.5) and adjusted *P*-value (less than 0.01). Ribosomal and mitochondrial genes were removed from the final list of genes reported and visualized. Where stated, the top DEG markers were subsequently selected for each group, based on fold change.

**Gene signature scores.** The TC signature scores were computed using the AddModuleScore function with default parameters from Seurat.

DEGs with a log-transformed fold change greater than 1.5 and adjusted *P* < 0.01 reported previously[5] (GSE294005) were used as signature genes for TC I–IV subsets, or genes reported in Fig. 1m were used as signature genes for TCP and LTiP.

**Data imputation for scRNA-seq data.** MAGIC imputation[73] was applied to the log-normalized expression values to further de-noise and recover missing values. Imputed gene-expression values were used for data visualization on heatmaps, where stated.

**Cell fate analysis.** Cell fate analysis was done using CellRank 2 v.2.0.6 (ref. 28). First, pseudotime was computed using Palantir v.1.3.6 with a boundary cell in the LMPP cluster (barcode: flex_5_CGACACAAGTGTTTGTACAGACCT-1) as start point. Cell–cell transition matrix was computed using the PseudotimeKernel based on Palantir pseudotime. Macrostates were computed using the GPCCA estimator (n_states = 20). Macrostate in the LMPP cluster was selected as the initial state for the estimator. From the identified macrostates, cells annotated as TC I, TC II, pDC, cDC1, cDC2, ILC1, ILC2 and LTi cells were selected as terminal states, based on a priori knowledge of haematopoietic cell differentiation. Cell-fate probabilities for each lineage were computed using the compute_fate_probabilities function (tol = $1 \times 10^{-8}$). Gene-expression trends were computed by fitting generalized additive models to MAGIC-imputed expression data using GAM function (distribution=gaussian, link=identity). Finally, driver genes of each lineage were identified using the compute_lineage_drivers function.

**scRNA/ATAC-seq analysis.** The scRNA/ATAC-seq data were obtained from ref. 1 (GSE205065) and processed as previously described[1]. Genome tracks were visualized using the plotBrowserTrack function in ArchR v.1.0.3 with peaks from the TC and ILC3/LTi clusters. Motif enrichment was done using chromVAR v.1.14.0 as previously described[1]. The 'top motif' for each transcription factor was selected by correlating its log-normalized gene-expression values (from multiome scRNA-seq) with the deviation *z*-scores of its motifs, in the same cells, and picking the motif with the highest Pearson correlation coefficient.

**Adoptive transfer of FL progenitors.** Progenitors were pre-enriched from fetal (E17.5–18.5) liver from *Rorc*[Venus] mice by depletion of lineage-positive (CD19, NK1.1, TCRβ, TCRγδ, Ter-119, Ly-6G, CD3e, FceR1a, CD90.2 and CD88) cells as described above. Single-cell suspensions of enriched FL were stained for lineage markers (Streptavidin and Siglec-F) alongside XCR1, CSF1-R, Ly6D, IL7R, FLT3, CD27, Siglec-H, MHCII and CXCR6, and indicated that progenitors were FACS-isolated, as per Extended Data Fig. 5b. Approximately $20 \times 10^3$ CLP cells, pre-pDC/tDC or MDP/CDP cells were transferred by intrahepatic injection into P2 CD45.1 recipients. Spleen and mLN were collected 6 days later and analysed.

**In vitro culture of lymph-node progenitors.** The mLNs from 2–3-week-old CD45.1 mice were individually embedded in 6% w/v low-melting-point agarose in PBS. The mLNs were oriented to allow for slicing across the largest cross-section. Once they had hardened, a 15.5-mm specimen tube (Precisionary Instruments) was used to extract a section of agarose containing the mLN, and the section was then glued onto the specimen tube stage with Krazy Super Glue. Immediately afterwards, the specimen and specimen tube were submerged in a Compresstome VF-510-0Z buffer tray containing ice-cold cRPMI. Then, sections 300 μm thick were generated using a Compresstome VF-510-0Z vibratome, with a speed of 0.16 mm s$^{-1}$ and a frequency of 27 Hz. Slices were immediately placed in a 6-well plate containing 0.4-μm cell-culture inserts and 3 ml per well cRPMI, equilibrated at 37 °C, 5% CO$_2$. Stainless steel washers (7/16 inch) were placed over the slices. Slices were then rested at 37 °C and 5% CO$_2$ for 2 h. TCPs and early TCs from mLN of P11–12 *Rorc*[Venus] mice were sort-purified as live, Lin(TCRβ, B220, CD19,

NK1.1, SiglecF$^{hi}$)$^-$CD88$^-$Ly6C$^-$RORγt(Venus)$^+$MHCII$^+$CXCR6$^-$NCAM1$^-$EpCAM$^-$PD1$^-$ cells. Next, 50 cells were added to the mLN slices. After 24 h, agarose was removed and the mLN slices were digested in RPMI 1640 medium supplemented with 5% fetal bovine serum, 1% L-glutamine, 1% penicillin–streptomycin, 10 mM HEPES, 1 mg ml$^{-1}$ collagenase A (Sigma, 11088793001) and 1 U ml$^{-1}$ DNase I (Sigma, 10104159001) for 30 min at 37 °C, 250 r.p.m. Digested samples were filtered through 100-µm strainers and centrifuged to remove the collagenase solution. Single-cell suspensions were stained and analysed by flow cytometry, as outlined above.

**Neonatal 4-OH tamoxifen administration.** For neonatal labelling in strains harbouring creERT2 alleles, pups were injected intraperitoneally with 4-OHT every 48 h. To induce recombination in $Aire^{GreenLantern\text{-}creERT2}R26^{lsl\text{-}tdTomato}$ mice, pups were administered 75 µg 4-OHT at P3, progressively increasing to a final dose of 150 µg on P18. To induce activation of FLT3 in R26:FlpoERT2 $Flt3^{Frt\text{-}ITD}$ mice, pups received 25 µg 4-OHT at P3, progressively increasing to a final dose of 75 µg on P11.

**Tamoxifen gavage.** For tamoxifen administration in adult mice, 40 mg tamoxifen was dissolved in 100 µl ethanol and subsequently in 900 µl sunflower oil (Sigma-Aldrich) and sonicated for 30–60 min. Mice were orally gavaged with 100 µl tamoxifen.

### Statistics and reproducibility

Analysis of all data was done using unpaired two-tailed $t$-tests, one- or two-way ANOVA with a 95% confidence interval or model-based analysis of single-cell transcriptomics[73], as specified in the text or figure legends. Details of the number of replicates, sample size, significance tests and value and meaning of $n$ for each experiment are included in the Methods or figure legends. Statistical tests were done with GraphPad Prism v.9 and v.10. $P$ values of less than 0.05 were considered to indicate statistical significance, adjusted for multiple comparisons. All experiments were repeated at least twice as successful, independent experiments.

### Reporting summary

Further information on research design is available in the Nature Portfolio Reporting Summary linked to this article.

### Data availability

The scRNA-seq datasets are available in the NCBI Gene Expression Omnibus under accession number GSE316677. This manuscript uses publicly available data from the Human Cell Atlas, available at https://www.gutcellatlas.org, and previously published scRNA/ATAC-seq data from GSE174405 and GSE294005. Source data are provided with this paper.

### Code availability

No new software or code was generated in this study. The scripts used to analyse scRNA-seq data can be accessed at Github (https://github.com/pty0111/TC-progenitor-2026).

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

**Acknowledgements** We thank J. van der Veeken and C. Campbell for discussions; staff at the Single-cell Analysis and Innovation Lab and Integrated Genomics Operation Core at MSK for single-cell genomics assays; A. Bhandoola for the $Tcf7^{fl/fl}$ mice; S. Vardhana for the $Tox^{fl/fl}$ mice; P. Cohen for the $Prdm16^{fl/fl}$ mice; R. Levine for the R26:FlpoERT2 $Flt3^{Frt\text{-}ITD}$ mice; Y. Kobayashi for the $Tnfrsf11a^{cre\text{-}GFP}$ mice; F. Geissmann for the $Spi1^{fl/fl}$ mice; M. Colonna for the $Tnfsf11^{fl/fl}$ and $Tnfrsf11a^{fl/fl}$ mice; H. Takayanagi for the $Twist2^{cre}$ mice; M. Dalod and E. Tomasello for the $Siglech^{cre}$ mice; and H. Rodewald for the $Il7r^{cre}$ mice. This work was supported by NIH/National Institute of Allergy and Infectious Diseases DP2 award DP2AI171116 (to C.C.B.), the G. Harold and Leila Y. Mathers Foundation, the V Foundation, a Pew Biomedical Scholar Award (C.C.B.), and NCI Cancer Center Support Grant P30 CA08748, NIH/NHGRI HG012103 (C.L.). C.C.B. is a Freeman Hrabowski Scholar with the Howard Hughes Medical Institute (HHMI). Y.A.P.I. is supported by an HHMI Gilliam Fellowship. T.P. is supported by a Cancer Research Institute Immuno-Informatics Postdoctoral Fellowship (CRI14462). G.S. is supported by a Kravis-WiSE fellowship. T.H. is supported by a Walter Benjamin Fellowship of the German Research Foundation. We acknowledge the use of the Integrated Genomics Operation Core, funded by an NCI Cancer Center Support Grant (CCSG, P30 CA08748), Cycle for Survival and the Marie-Josée and Henry R. Kravis Center for Molecular Oncology.

**Author contributions** Y.A.P.I. and C.C.B. conceived the study, designed experiments, analysed and interpreted data, and wrote the manuscript. Y.A.P.I., E.B., G.S. and T.H. designed and performed experiments and analysed data; G.F., Y.F.P., B. Akagbosu, Z.Z. and L.F. performed experiments; T.P. designed and performed computational analyses for SS3 and 10x Flex data; A.Y. designed and performed computational analysis for 10x scRNA-seq on RANK-deficient TCs; M.E.B. designed and performed computational analysis of publicly available human gut cell atlas data; L.M.J., J.M., B. Afzali and B.D.P. provided mouse strains, tissues and intellectual expertise; C.L. supervised computational analyses of SS3 and 10x Flex data; C.C.B. supervised the research; and all authors read and approved the manuscript.

**Competing interests** The authors declare no competing interests.

**Additional information**
**Correspondence and requests for materials** should be addressed to Chrysothemis C. Brown.

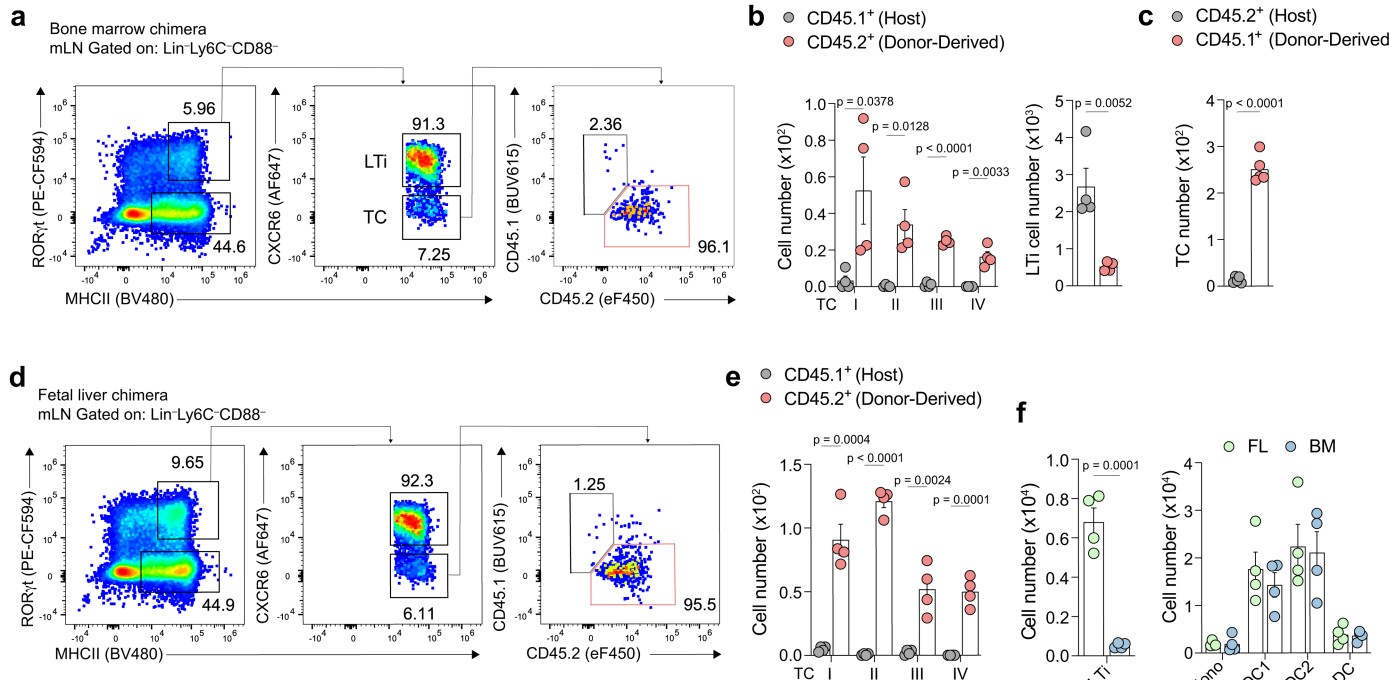

**Extended Data Fig. 1 | Thetis cell progenitors are enriched in fetal liver.**
**a-b**, Flow cytometry of TCs in mesenteric lymph node (mLN) of BM chimeras as in (Fig. 1a) and summary graphs of CD45.2 and CD45.1 TC and LTi subset numbers. **c**, Numbers of CD45.1 TCs in reciprocal BM chimeras. **d**–**e**, Flow cytometry of TCs in mLN of E15.5 FL chimeras as in (Fig. 1a) and summary graph of TC subset numbers. **f**, Number of CD45.2+ cells for indicated cell types in BM and FL chimeras as in Fig. 1a. Data in **a**–**f** are representative of four independent experiments. Error bars: means ± s.e.m. Two-tailed unpaired t-test (b, c, e, f). All *P* values are indicated on the corresponding graphs.

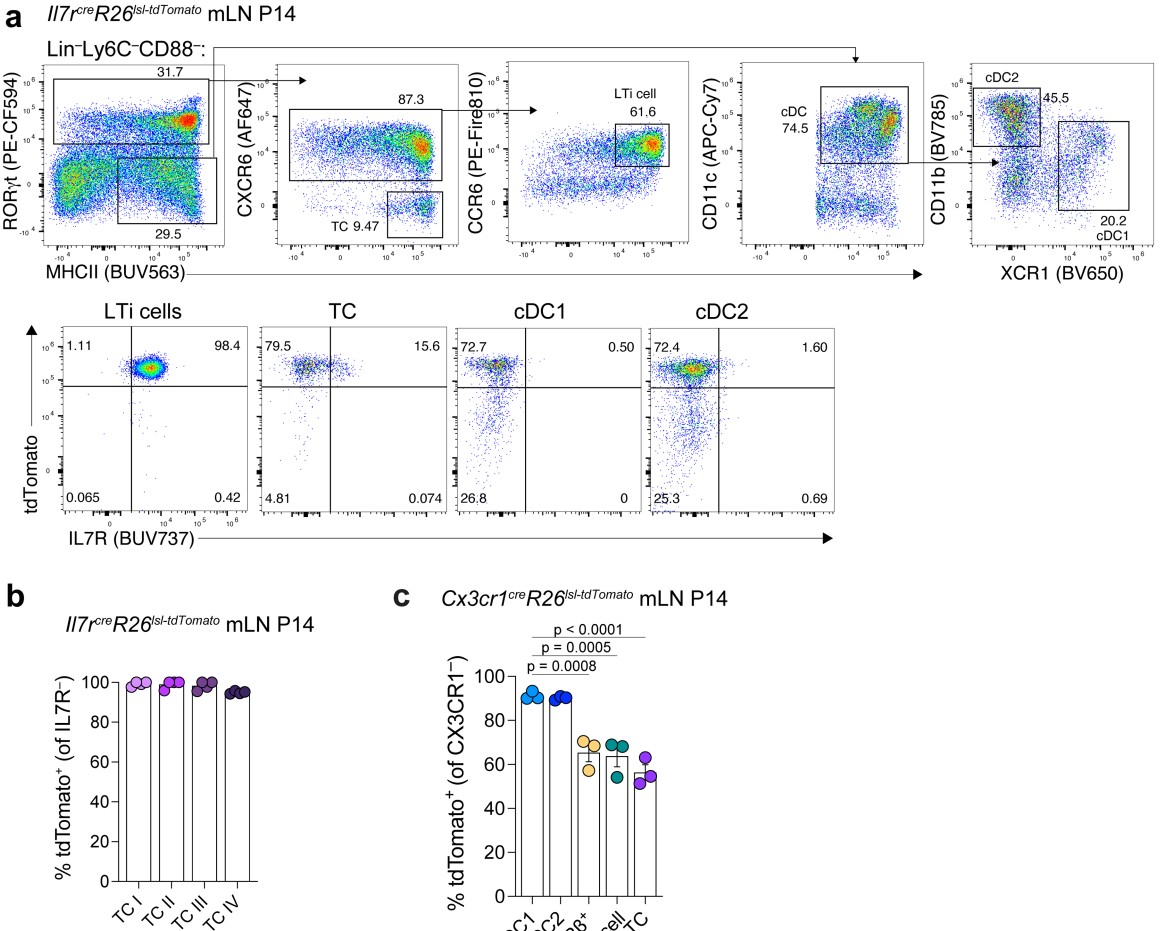

**a** *Il7r^cre R26^lsl-tdTomato* mLN P14

**b** *Il7r^cre R26^lsl-tdTomato* mLN P14

**c** *Cx3cr1^cre R26^lsl-tdTomato* mLN P14

**Extended Data Fig. 2 | Thetis cells are descended from an IL7R⁺ progenitor.**
**a**, Representative flow cytometry analysis of tdTomato expression in LTi cells, TCs and cDCs isolated from mLN of *Il7r^cre R26^lsl-tdTomato* fate-mapped mice at P14 (*n* = 4). **b**, Frequency of tdTomato⁺ cells among individual TC subsets in mLN of P14 *Il7r^cre R26^lsl-tdTomato* fate-mapped mice (*n* = 3). **c**, Frequency of tdTomato⁺ cells among indicated cell types in mLN of P14 *Cx3cr1^cre R26^lsl-tdTomato* fate-mapped mice (*n* = 3). Error bars: means ± s.e.m. Data in **b** and **c** representative of three independent experiments. One-way ANOVA (b). All *P* values are indicated on the corresponding graph.

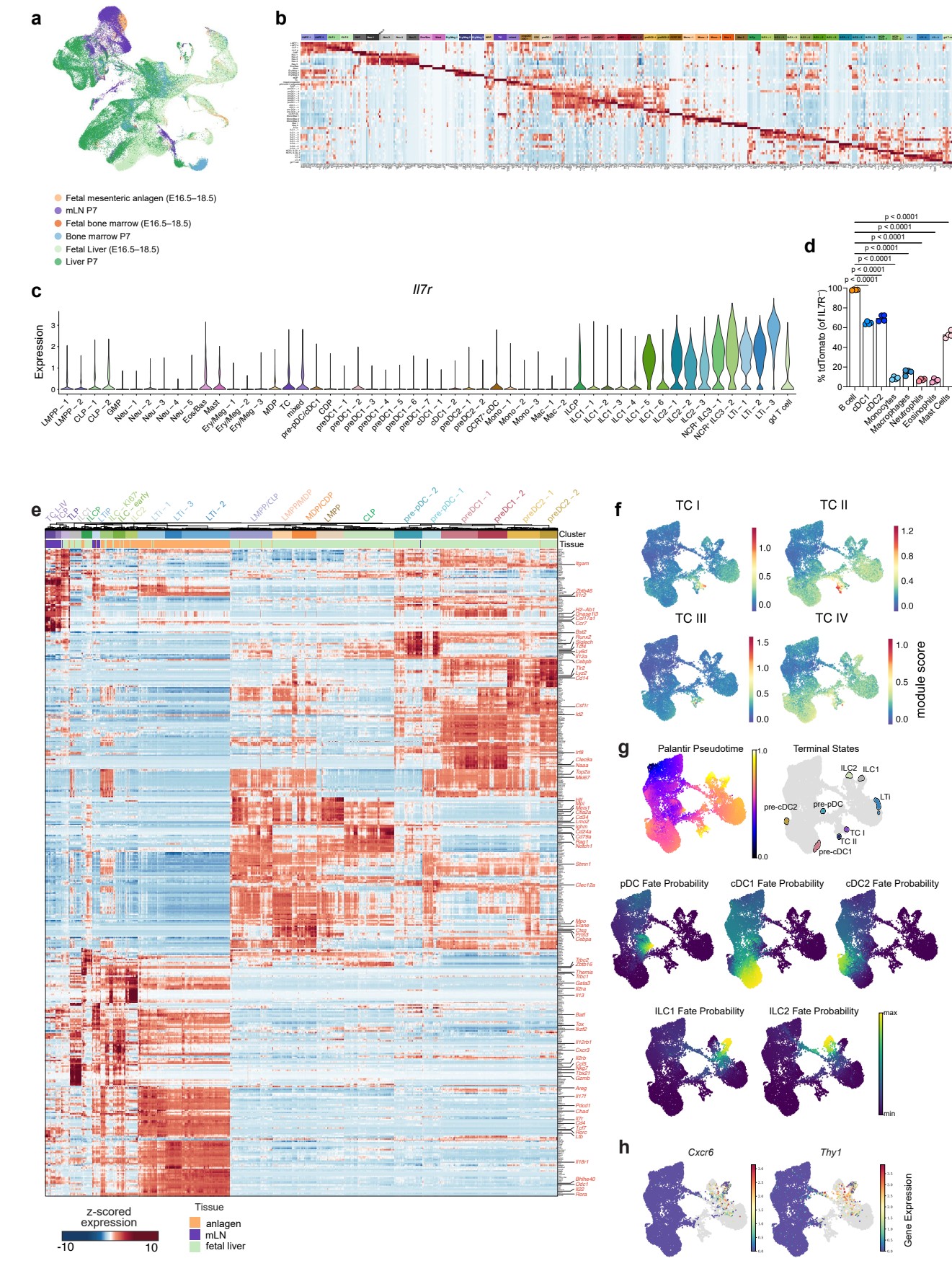

**Extended Data Fig. 3 |** See next page for caption.

**Extended Data Fig. 3 | scRNA-seq analysis of fetal and postnatal hematopoiesis. a**, Uniform manifold approximation and projection (UMAP) of fetal and postnatal hematopoietic cells as in Fig. 1g, colored by tissue and developmental stage. **b**, Heatmap showing scaled, imputed expression of top 5 differentially expressed genes (one vs the rest, FC > 1.5, adj. $P$ < 0.01) for each pseudo-bulked cluster identified in Fig. 1g. **c**, Expression of *Il7r*. **d**, Frequency of tdTomato$^+$ cells among indicated cell types in spleen of P10 *Il7r$^{cre}$R26$^{lsl-tdTomato}$* mice ($n$ = 4). **e**, Heatmap showing scaled, imputed expression of top 20 differentially expressed genes (one vs the rest, FC > 1.5, adj. $P$ < 0.01) for each cluster of cells shown in Fig. 1h and k. Annotated clusters ordered by hierarchical clustering of differentially expressed genes. **f**, UMAP of 14,581 FL progenitors, TCs and ILCs colored by TC I-IV gene module scores[4]. **g**, UMAP colored by pseudotime (top left), select, inferred terminal states using CellRank 2 (top right), and fate probabilities towards the indicated terminal states (bottom). **h**, UMAP with FL-derived cells colored by unimputed expression of CXCR6 and Thy1 (CD90). Non-FL cells colored in grey. Each symbol represents an individual mouse. Error bars: means ± s.e.m. Data in **d** representative of two independent experiments. One-way ANOVA (d). All $P$ values are indicated on the corresponding graph.

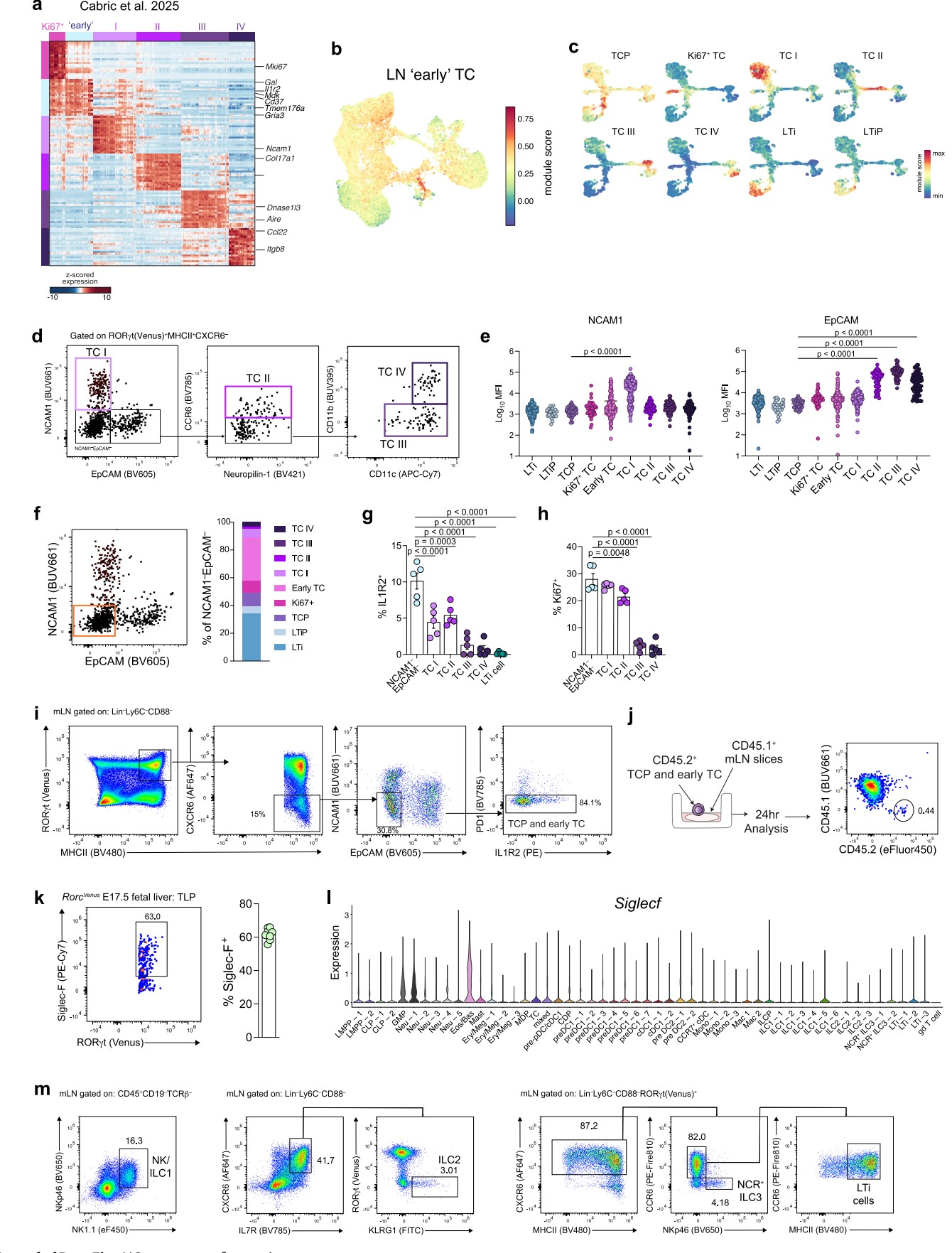

**Extended Data Fig. 4** | See next page for caption.

**Extended Data Fig. 4 | TCP represents the immediate TC progenitor.**
**a**, Heatmap showing scaled, imputed expression of top 20 differentially expressed genes (one vs the rest, FC > 1.5, adj. $P$ < 0.01) for each cluster of TCs previously identified by Cabric et al.[5], TCP signature genes highlighted in red. **b**, UMAP of FL progenitors, TCs and ILCs from Fig. 1h, colored by expression of 'early TC' module score. **c**, UMAP as in Fig. 2a colored by TC I-IV or LTi gene module scores[5] or TCP or LTiP gene module scores defined in this study. **d**, Gating strategy for identification of TC subsets based on index-sorting analysis Fig. 2c. **e**, Summary graphs for NCAM1 and EpCAM cell surface protein expression (MFI, median fluorescence intensity). Each symbol represents an individual cell. **f**, Proportion of each cluster among index-sorted CXCR6⁻RORγt (Venus)⁺MHCII⁺NCAM1⁻EpCAM⁻ cells. **g**, Frequency of IL1R2⁺ cells among indicated TC subsets in mLN of P14 *Rorc*^Venus^ ($n$ = 4) mice. **h**, Frequency of Ki67⁺ cells across TC subsets from mLN of P14 C57Bl/6 mice ($n$ = 4). **i**, Flow sorting for isolation of the TCP and early TCs from mLN of P11–12 *Rorc*^Venus^ mice. **j**, Schema for ex vivo cultures of TCP and early TCs from with mLN slices from 2–3-week-old CD45.1 mice. **k**, Flow cytometry of Siglec-F expression on FL TLPs from E17.5 *Rorc*^Venus^ mice and summary graph ($n$ = 8 mice). **l**, Expression of *Siglecf* in clusters defined in Fig. 1g. **m**, Representative flow cytometry for identification of ILC subsets in mLN of P11 *Siglecf*^Fre^*R26*^lsl-tdTomato^*Rorc*^Venus^ mice. Each symbol represents an individual mouse. Error bars: means ± s.e.m. Data in **g** and **h** representative of three independent experiments. Data in **i**, **j** and **m** representative of two independent experiments. Data in **k** representative of three independent experiments. One-way ANOVA (e, g, h). All $P$ values are indicated on the corresponding graphs.

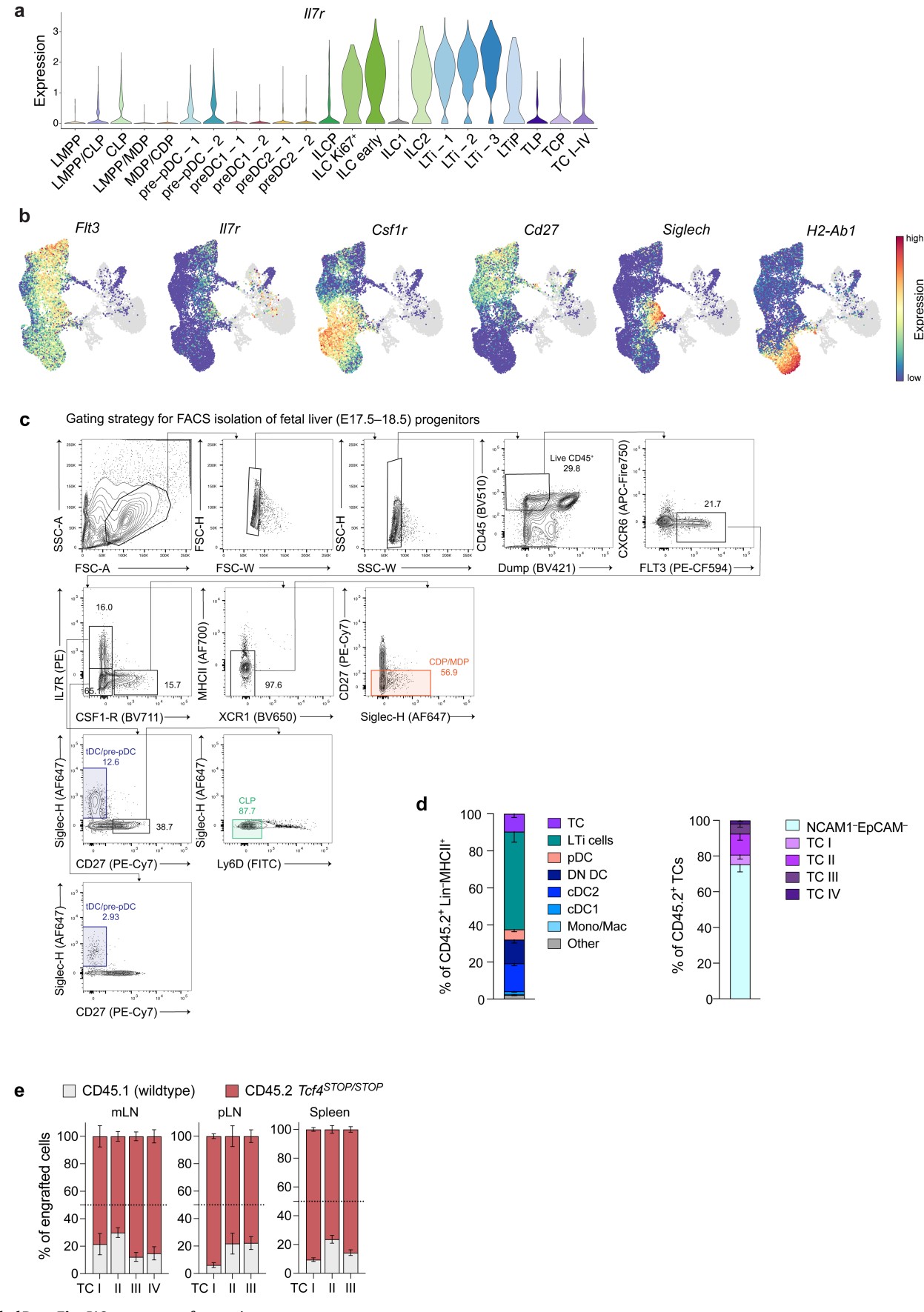

**Extended Data Fig. 5 |** See next page for caption.

**Extended Data Fig. 5 | TLP are descended from IL7R⁺SiglecH⁺ lymphoid progenitors. a**, Violin plot of *Il7r* expression across clusters defined in Fig. 1h. **b**, UMAP of FL progenitors colored by unimputed expression of indicated genes. FL cells colored in grey. **c**, FACS-isolation of Lin(CD19, NK1.1, TCRβ, TCRγδ, Ter-119, Ly-6G, CD3e, FceR1a, CD90.2, CD88, Siglec-F)⁻FLT3⁺IL7R⁺CSF1R⁻Ly6D⁻CD27⁺ CLP, Lin⁻FLT3⁺SiglecH⁺ pDC/tDC progenitors, or Lin⁻FLT3⁺IL7R⁻CSF1R⁺MHCII⁻/ᵗᵒCD11c⁻/ᵗᵒ MDP/CDP progenitors from E18.5 *Rorc^Venus* liver. **d**, Frequency of APC subsets (left) among mLN Lin(CD19, TCRβ, Siglec-Fʰⁱ)⁻MHCII⁺CD45.2⁺ cells, 6 days post transfer into CD45.1 P2 recipients (*n* = 6). Frequency of TC subsets (right) among CD45.2⁺ TCs in mLN 6 days post transfer into CD45.1 P2 recipients (*n* = 6). **e**, Competitive chimeras were generated with a 1:1 mix of CD45.2 *Tcf4^STOP/STOP*^ and CD45.1 wildtype E18.5 FL transplanted into lethally irradiated CD45.1 recipients and analyzed 4 weeks later (*n* = 5 recipients). Proportion of CD45.2 and CD45.1 cells among TC subsets in mLN, skin-draining peripheral lymph nodes (pLN) and spleen. Data in **d** pooled from five independent experiments, data in **e** representative of three independent experiments. Error bars: means ± s.e.m. One-way ANOVA (d). All *P* values are indicated on the corresponding graph.

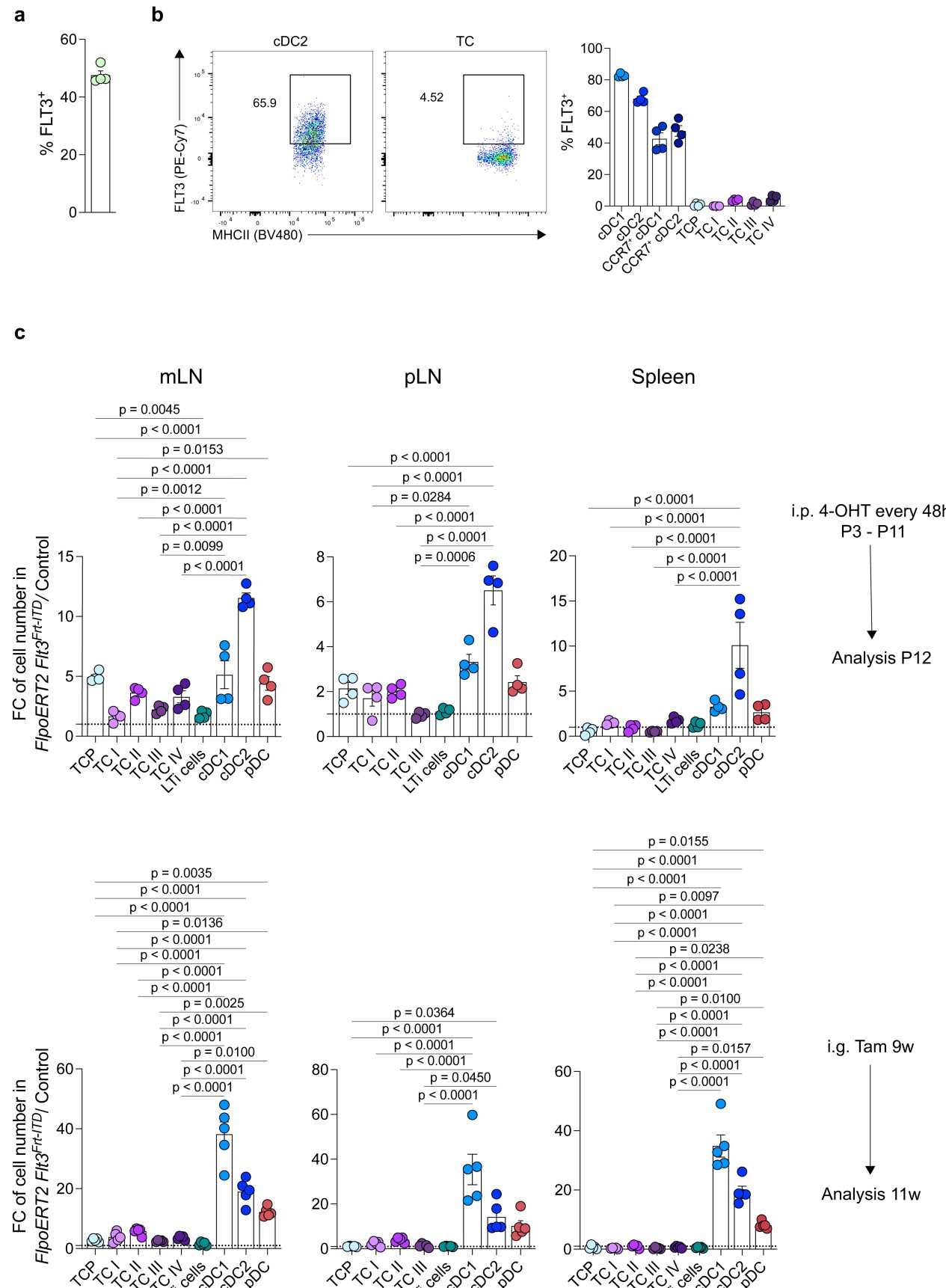

**Extended Data Fig. 6** | See next page for caption.

**Extended Data Fig. 6 | FLT3 regulation of TC differentiation. a**, Proportion of FLT3$^+$ TLP within E18.5 FL of *Rorc$^{Venus}$* mice ($n$ = 4). **b**, Flow cytometry and summary graph of FLT3 expression by indicated cell types within mLN of P14 *RORc$^{Venus}$* mice ($n$ = 4). **c**, Summary graph of fold change of cell numbers (R26:FlpoERT2 *Flt3$^{Frt-ITD}$* ($n$ = 5 adult or 4 neonatal mice) over littermate control ($n$ = 3 adult or 6 neonatal) mice for indicated cell types in mLN, pLN and spleen. Data in **b** representative of two independent experiments. Data in **c** pooled from two independent experiments (neonates) or representative of two independent experiments (adult). Error bars: means ± s.e.m. One way ANOVA (c). All *P* values are indicated on the corresponding graph.

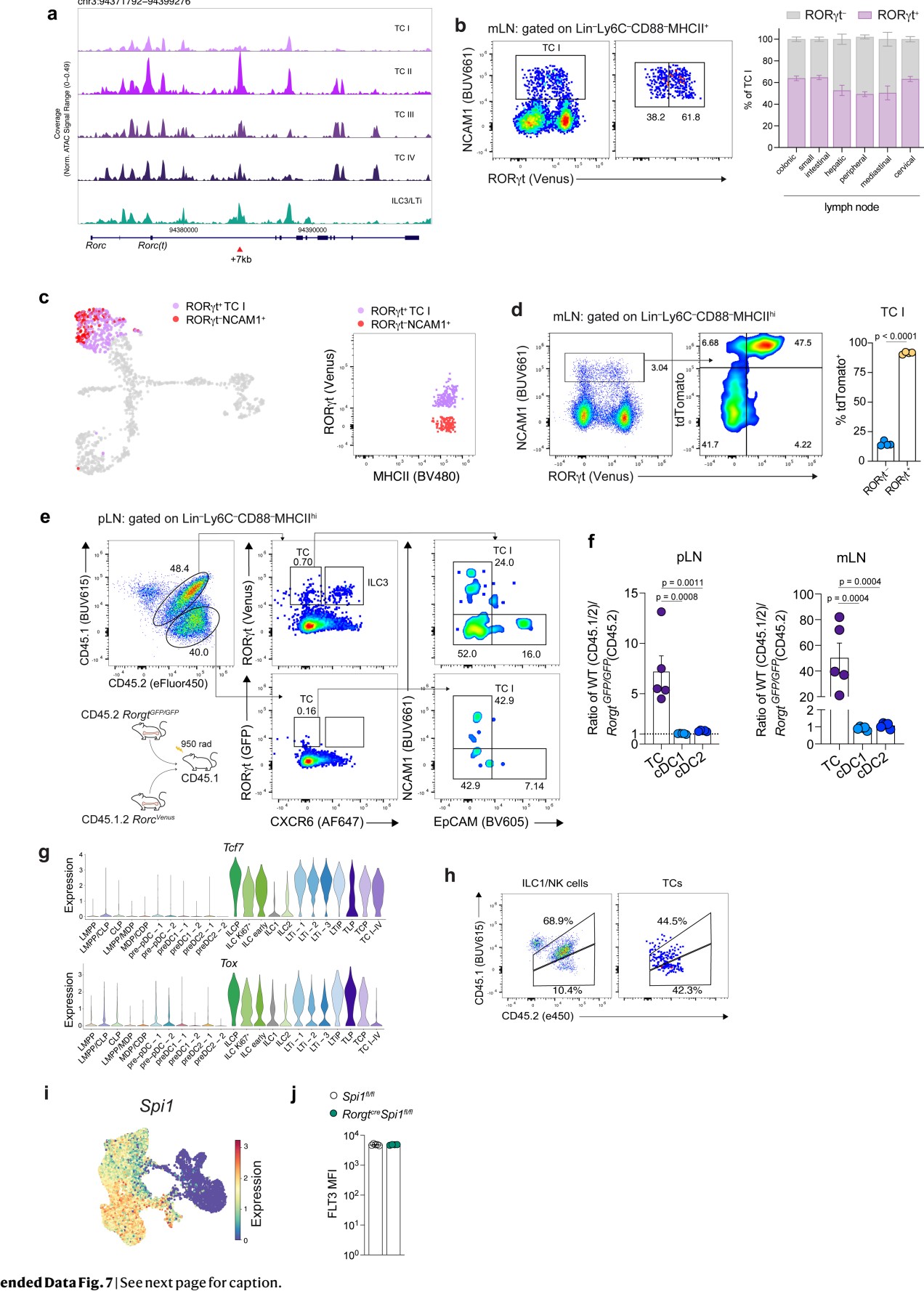

**Extended Data Fig. 7 |** See next page for caption.

**Extended Data Fig. 7 | Transcriptional regulation of TCP differentiation.**
**a**, Chromatin accessibility at the *Rorc* locus. **b**, Expression of RORγt (Venus) by Lin⁻MHCII^hi^NCAM1⁺ cells (TC I) in mLN of P14 *Rorc^Venus^Aire^GL^* mice and summary graph of proportion of RORγt(Venus)⁺ and RORγt(Venus)⁻ cells among Lin⁻MHCII^hi^NCAM1⁺ across indicated lymph nodes (*n* = 3). **c**, UMAP of RORγt⁺ TCs (as in Fig. 2a) and RORγt⁻MHCII⁺NCAM1⁺ profiled by SS3 (left) and flow plot of RORγt(Venus) expression. **d**, Expression of RORγt (Venus) and tdTomato in Lin⁻MHCII^hi^NCAM1⁺ cells from mLN of P15 *Rorc^Venus^Rorgt^cre^R26^lsl-tdTomato^* mice (*n* = 4) and summary graph of proportion of tdTomato⁺ cells (right). **e**, Mixed BM chimeras were generated with a 1:1 mix of CD45.2 *Rorgt^GFP/GFP^* and CD45.1/2 *Rorc^Venus^* (wildtype) BM transplanted into lethally irradiated CD45.1 recipients and analyzed 3 weeks later. Representative flow cytometry of Venus⁺ (wildtype) or GFP⁺ (RORγt-null) TC subsets and LTi cells in pLN. **f**, Summary graphs of ratio of CD45.1.2 wildtype cells over CD45.2 RORγt-deficient cells among indicated cell types (*n* = 5). Each symbol represents an individual mouse. **g**, Expression of *Tox* and *Tcf7* across clusters defined in Fig. 1h. **h**, Mixed BM chimeras were generated with a 1:1 mix of CD45.2⁺ *Il7r^cre^Tox^fl/fl^* and CD45.1/2 wildtype BM transplanted into lethally irradiated CD45.1 recipients and analyzed 4 weeks later. Representative flow plot of CD45.1/2 and CD45.2 cells gated on NK/ILC1 or TCs. **i**, UMAP as in Fig. 1h colored by unimputed expression of *Spi1* (PU.1). **j**, Expression of FLT3 by FL TLP in *Rorgt^cre^Spi1^fl/fl^* or littermate control mice. Data in **b**, **d**, and **e** are representative of three or more independent experiments, data in **e** and **h** are representative of two independent experiments. Error bars: means ± s.e.m. Two-tailed unpaired *t*-test (d) or one-way ANOVA (f). All *P* values are indicated on the corresponding graph.

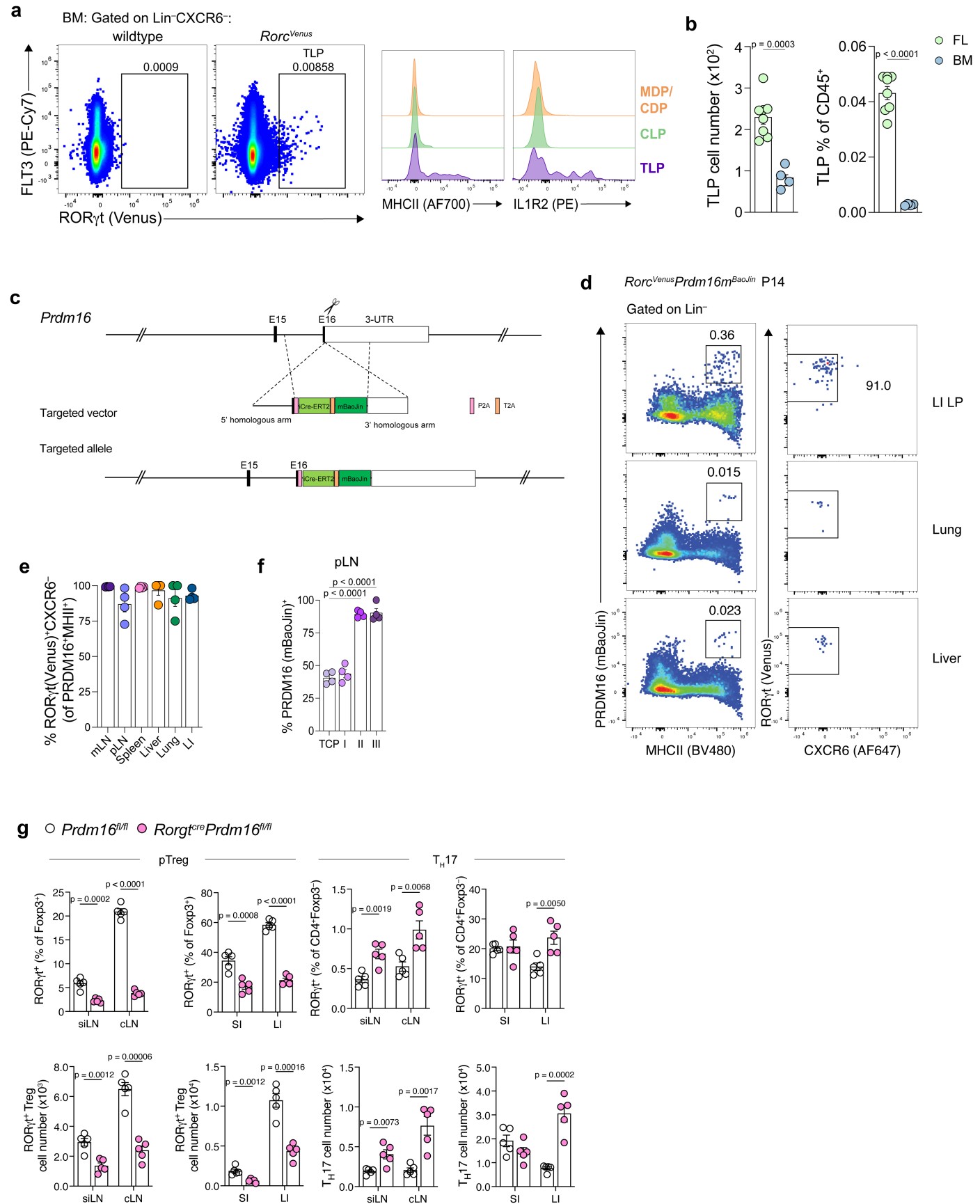

**Extended Data Fig. 8** | See next page for caption.

**Extended Data Fig. 8 | Postnatal regulation of TC differentiation.**
**a**, Representative flow plot of TLP in BM of adult (13 week) *Rorc*<sup>Venus</sup> mice and expression of TLP marker genes. Representative of $n = 3$ mice. **b**, Absolute number and frequency of TLP in E18.5 FL ($n = 8$) or adult (8-week-old) BM ($n = 4$) from *Rorc*<sup>Venus</sup> mice. **c**, Targeting strategy for the *Prmd16* locus. **d**, Representative flow cytometry of PRDM16(mBaoJin)-expressing antigen presenting cells in large intestine lamina propria (LI), lung, and liver of P14 *Rorc*<sup>Venus</sup>*Prdm16*<sup>creERT2-mBaoJin</sup> mice. Representative of $n = 4$ mice. **e**, Frequency of RORγt(Venus⁺)CXCR6⁻ TCs among all PRDM16(mBaoJin)⁺ APCs. **f**, Summary graph of frequency of PRDM16(mBaoJin)⁺ cells among TC subsets within pLN of *Rorc*<sup>Venus</sup>*Prdm16* <sup>creERT2-mBaoJin</sup> P14 mice ($n = 4$). **g**, Frequency and number of RORγt⁺Foxp3⁺ pTreg cells and CD4⁺Foxp3–CD44<sup>hi</sup>RORγt⁺ T<sub>H</sub>17 cells in colonic LN (cLN), small intestinal LN (siLN), small intestine lamina propria (SI) and LI of P28 *Rorgt*<sup>cre</sup>*Prdm16*<sup>fl/fl</sup> mice ($n = 5$ mice per group). Each symbol represents an individual mouse. Data in **d**–**g** are representative of two-three independent experiments. Each symbol represents an individual mouse. Error bars: means ± s.e.m. Two-tailed unpaired t-test (b, g) or one-way ANOVA (d). All *P* values are indicated on the corresponding graph.

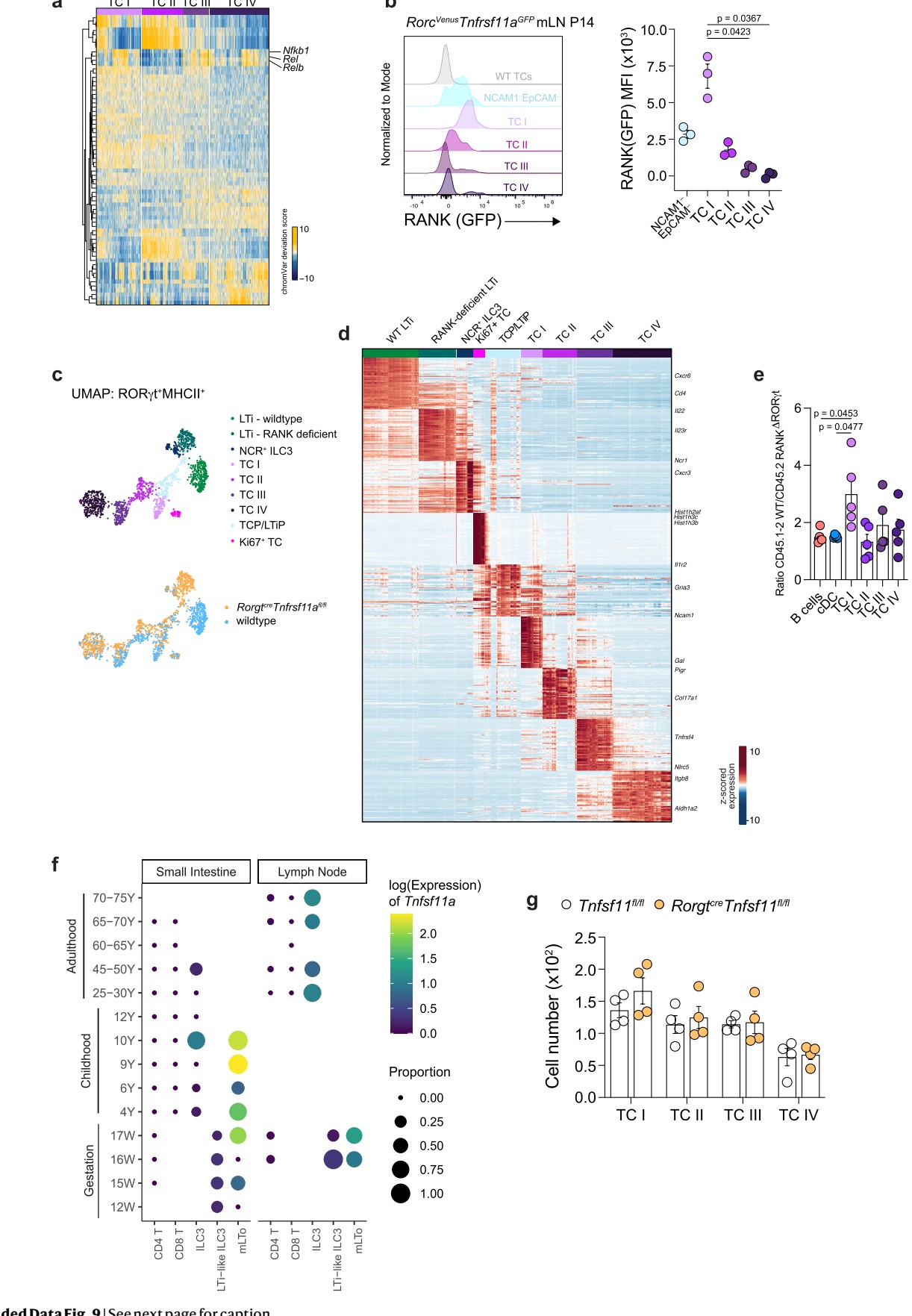

**Extended Data Fig. 9** | See next page for caption.

**Extended Data Fig. 9 | TC I differentiation is dependent on LTo-derived RANKL. a**, Heatmap reporting scaled chromVAR deviation transcription factor (TF) motif scores for top TF gene-motif pairs in TCs in scATAC-seq data from Akagbosu et al. **b**, Expression of RANK(GFP) by TC subsets within mLN of P14 $Rorc^{Venus}Tnfrsf11a^{GFP}$ ($n = 4$) mice and summary graph of median fluorescence intensity (MFI). **c**, UMAP of 1505 RORγt⁺MHCII⁺ cells from mLN of P14 $Rorc^{Venus}Rorgt^{cre}Tnfrsf11a^{fl/fl}$ ($n = 6$) or littermate $Rorc^{Venus}Tnfrsf11a^{fl/fl}$ (wildtype; $n = 4$) mice colored by cluster annotation (top) or genotype (bottom). **d**, Heatmap showing scaled, imputed expression of top 50 differentially expressed genes (one vs the rest, FC > 1.5, adj. $P < 0.01$) for each cluster. **e**, Mixed BM chimeras were generated with a 1:1 mix of CD45.2⁺ $Rorgt^{cre}Tnfrsf11a^{fl/fl}$ and CD45.1/2 wildtype BM transplanted into lethally irradiated CD45.1 recipients. Proportion of CD45.1 and CD45.2 cells among indicated TC subsets, 4 weeks post reconstitution ($n = 5$ recipients). **f**, Dot plot of $Tnfsf11$ expression by indicated cell types within lymph nodes and intestinal tissue, profiled across development (Human Gut Atlas). **g**, Summary graph of TC subset numbers within mLN of P14 $Rorgt^{cre}Tnfsf11^{fl/fl}$ mice and littermate controls ($n = 4$ per group). Each symbol represents an individual mouse. Data are representative of two (**e**), three (**g**) or four (**b**) independent experiments. Error bars: means ± s.e.m. One-way ANOVA (b, e). All $P$ values are indicated on the corresponding graph.

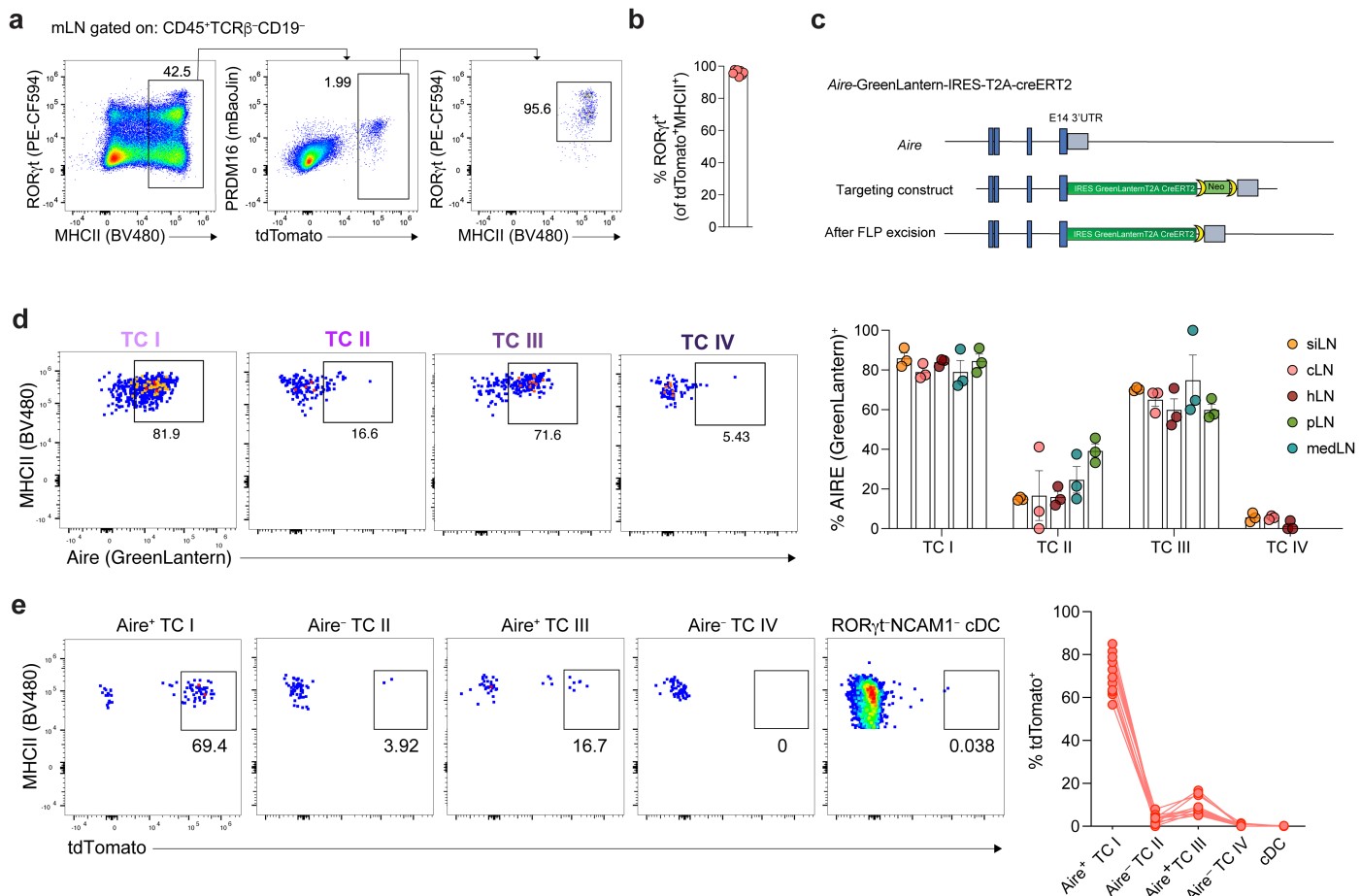

**Extended Data Fig. 10 | TC I–IV represent discrete cell subsets.**
**a**–**b**, Representative flow cytometry and summary graph of frequency of RORγt⁺ cells among CD19⁻MHCII⁺tdTomato⁺ cells from mLN of P14 *Prdm16^{creERT2-mBaoJin}* *R26^{lsl-tdTomato}* mice treated with 4-OHT from P7. **c**, Targeting strategy for the *Aire* locus. **d**, Representative flow cytometry of Aire (GreenLantern) expression among TC subsets in mesenteric lymph nodes (mLN) of *Rorc^{Venus}Aire^{GreenLantern-creERT2}* P14 (*n* = 3) mice and summary graph for frequency of Aire⁺ cells among TCs

from indicated lymph nodes (pLN, peripheral LN; hLN, hepatic LN; medLN, mediastinal LN; cLN, colonic LN; siLN, small intestinal LN). **e**, Flow cytometry and summary graph of tdTomato expression among TC subsets from mLN of P18 *Aire^{GreenLantern-creERT2}R26^{lsl-tdTomato}* mice treated with 4-OHT from P7 (*n* = 11). Data in **a** and **d** representative of three independent experiments, data in **b** and **e** pooled from two independent experiments. Each symbol represents and individual mouse. Error bars: means ± s.e.m.

# Reporting Summary

## Statistics

For all statistical analyses, confirm that the following items are present in the figure legend, table legend, main text, or Methods section.

| n/a | Confirmed | |
|---|---|---|
| ☐ | ☒ | The exact sample size (*n*) for each experimental group/condition, given as a discrete number and unit of measurement |
| ☐ | ☒ | A statement on whether measurements were taken from distinct samples or whether the same sample was measured repeatedly |
| ☐ | ☒ | The statistical test(s) used AND whether they are one- or two-sided <br> *Only common tests should be described solely by name; describe more complex techniques in the Methods section.* |
| ☒ | ☐ | A description of all covariates tested |
| ☐ | ☒ | A description of any assumptions or corrections, such as tests of normality and adjustment for multiple comparisons |
| ☐ | ☒ | A full description of the statistical parameters including central tendency (e.g. means) or other basic estimates (e.g. regression coefficient) AND variation (e.g. standard deviation) or associated estimates of uncertainty (e.g. confidence intervals) |
| ☐ | ☒ | For null hypothesis testing, the test statistic (e.g. *F*, *t*, *r*) with confidence intervals, effect sizes, degrees of freedom and *P* value noted <br> *Give P values as exact values whenever suitable.* |
| ☒ | ☐ | For Bayesian analysis, information on the choice of priors and Markov chain Monte Carlo settings |
| ☒ | ☐ | For hierarchical and complex designs, identification of the appropriate level for tests and full reporting of outcomes |
| ☒ | ☐ | Estimates of effect sizes (e.g. Cohen's *d*, Pearson's *r*), indicating how they were calculated |

*Our web collection on statistics for biologists contains articles on many of the points above.*

## Software and code

Policy information about availability of computer code

| | |
|---|---|
| Data collection | Cytek Aurora v3.3.0 and Facs Diva v8.0.1 (BD Biosciences) for flow cytometry. 10X Chromium for single cell RNA-sequencing. Details for RNA-seq are included in the methods. |
| Data analysis | Flowjo v10.10.0 for flow cytometry data. GraphPad Prism v9 and v10 for statistical analyses. <br><br> All computational methods are detailed in the Methods. No custom code was created for this study. Scripts used to analyze scRNA-seq data are available at Github (https://github.com/pty0111/TC-progenitor-2026). The following softwares were used: <br> HARP v0.1.1, Cell Ranger v8.0.0, Seurat v4.4.0, SCimilarity v0.3.0, Celltypist v1.5.3, zUMIs v2.9.7e, STAR v2.7.11a, CellRank 2 v2.0.6, ArchR v1.0.3, chromVAR v1.14.0 |

For manuscripts utilizing custom algorithms or software that are central to the research but not yet described in published literature, software must be made available to editors and reviewers. We strongly encourage code deposition in a community repository (e.g. GitHub). See the Nature Portfolio guidelines for submitting code & software for further information.

## Data

Policy information about availability of data

All manuscripts must include a data availability statement. This statement should provide the following information, where applicable:
- Accession codes, unique identifiers, or web links for publicly available datasets
- A description of any restrictions on data availability
- For clinical datasets or third party data, please ensure that the statement adheres to our policy

The mouse sequencing data is available through the Gene Expression Omnibus (accession number GSE316677)  Sequencing data for the Human Gut Atlas has been published previously and the processed data is publicly available from https://www.gutcellatlas.org. Previously published scRNA/ATAC-seq data from Cabric et al. (GSE174405) and Akagbosu et al. (GSE294005) were referenced.

## Human research participants

Policy information about studies involving human research participants and Sex and Gender in Research.

| | |
|---|---|
| Reporting on sex and gender | N/A |
| Population characteristics | N/A |
| Recruitment | N/A |
| Ethics oversight | N/A |

Note that full information on the approval of the study protocol must also be provided in the manuscript.

# Field-specific reporting

Please select the one below that is the best fit for your research. If you are not sure, read the appropriate sections before making your selection.

☒ Life sciences ☐ Behavioural & social sciences ☐ Ecological, evolutionary & environmental sciences

For a reference copy of the document with all sections, see nature.com/documents/nr-reporting-summary-flat.pdf

# Life sciences study design

All studies must disclose on these points even when the disclosure is negative.

| | |
|---|---|
| Sample size | Sample size was determined based on preliminary data, and previously published data for similar experiments (PMID: 38677292, PMID: 31358996 and 40373113). In all experiments, a minimum of three mice per group were used, Details as to the number of replicates and sample size are included in the Methods or Figure Legends. |
| Data exclusions | Samples with insufficient cell numbers were excluded from the analyses (Figure 3c; <90 CD45.2+Lin−MHCII+). |
| Replication | All experiments were repeated at least twice as successful, independent experiments. Details as to the exact number of replicates are included in the Figure Legends. |
| Randomization | All mice within a litter were used for immune-phenotyping experiments. Both male and female age-matched littermate controls were used for experiments. No treatments or interventions requiring allocation to an experimental or control group were performed. |
| Blinding | Investigators were not blinded. Mice of different genotypes were treated equally and housed together. |

# Reporting for specific materials, systems and methods

We require information from authors about some types of materials, experimental systems and methods used in many studies. Here, indicate whether each material, system or method listed is relevant to your study. If you are not sure if a list item applies to your research, read the appropriate section before selecting a response.

## Materials & experimental systems

| n/a | Involved in the study |
|---|---|
| ☐ | ☒ Antibodies |
| ☒ | ☐ Eukaryotic cell lines |
| ☒ | ☐ Palaeontology and archaeology |
| ☐ | ☒ Animals and other organisms |
| ☒ | ☐ Clinical data |
| ☒ | ☐ Dual use research of concern |

## Methods

| n/a | Involved in the study |
|---|---|
| ☒ | ☐ ChIP-seq |
| ☐ | ☒ Flow cytometry |
| ☒ | ☐ MRI-based neuroimaging |

# Antibodies

| Antibodies used | The following monoclonal fluorophore-conjugated antibodies were used in this study: |
|---|---|
| | B220 - Brilliant UltraViolet 737 - RA3-6B2 - BD Biosciences - 612838 - 1/600 |
| | c-Kit - Brilliant UltraViolet 395 - 2B8 - BioLegend - 564011 - 1/800 |
| | CCR6 - Brilliant Violet 785 - 29-2L17 - BioLegend - 129823 - 1/300 |
| | CCR6 - Alexa Fluor 647 - 140706 - BD Biosciences - 557976 - 1/200 |
| | CCR6 - PE-Fluor810 - 29-2L17 - BioLegend - 129833 - 1/100 |
| | CCR7 - Alexa Fluor 488 - 4B12 - BioLegend - 120110 - 1/100 |
| | CD11b - Brilliant UltraViolet 395 - M1/70 - BD Biosciences - 563553 - 1/1000 |
| | CD11b - Brilliant Violet 785 - M1/70 - BioLegend - 101243 - 1/400 |
| | CD11c - APC-Cy7 - N418 - BioLegend - 117323 - 1/200 |
| | CD11c - Brilliant UltraViolet 563 - N418 - BD Biosciences - 749040 - 1/400 |
| | CD19 - Brilliant UltraViolet 737 - 1D3 - BD Biosciences - 612781 - 1/800 |
| | CD19 - Biotinylated - 6D5 - BioLegend - 115504 - 1/200 |
| | CD19 - Brilliant Violet 421 - 6D5 - BioLegend - 115549 - 1/400 |
| | CD19 - FITC - 1D3 - BD Biosciences - 557398 - 1/500 |
| | CD27 - PE-Cy7 - LG.7F9 - ThermoFisher - 25-0271-80 - 1/100 |
| | CD27 - FITC - LG.7F9 - ThermoFisher - 11-0271-82 - 1/1000 |
| | CD31 - FITC - MEC13.3 - BD Biosciences - 553372 - 1/200 |
| | CD3e - Biotinylated - 145-2c11 - BioLegend - 100304 - 1/200 |
| | CD45 - Brilliant Violet 570 - 30-F11 - BioLegend - 103136 - 1/600 |
| | CD45 - Brilliant Violet 510 - 30-F11 - BioLegend - 103138 - 1/2000 |
| | CD45.1 - Brilliant UltraViolet 615 - A20 - BD Biosciences - 751467 - 1/100 |
| | CD45.2 - Violet450 - 104 - Tonbo Bioscience (Fisher Scientific) - 50-105-5147 - 1/400 |
| | CD64 - PE-Cy7 - X-54-5/7.1 - BioLegend - 139314 - 1/200 |
| | CD88 - Brilliant Violet 421 - 20/70 - BD Biosciences - 743769 - 1/400 |
| | CD88 - Biotinylated - 20/70 - BioLegend - 135811 - 1/200 |
| | CD88 - PerCP-Cy5.5 - 20/70 - BioLegend - 135813 - 1/100 |
| | CD90.2 - Biotinylated - 30-H12 - BioLegend - 105304 - 1/200 |
| | CD90.2 - Brilliant Violet 786 - 53-2.1 - BD Biosciences - 564365 - 1/800 |
| | CD90.2 - Brilliant UltraViolet 496 - 53-2.1 - BD Biosciences - 741046 - 1/400 |
| | CSF1-R - Brilliant Violet 711 - AFS98 - BioLegend - 135515 - 1/100 |
| | CXCR6 - PE - SA051D1 - BioLegend - 151104 - 1/200 |
| | CXCR6 - Alexa Fluor 647 - SA051D2 - BioLegend - 151115 - 1/200 |
| | CXCR6 - APC-Fire750 - SA051D1 - BioLegend - 151129 - 1/200 |
| | EpCAM - Brilliant Violet 605 - G8.8 - BioLegend - 118227 - 1/200 |
| | EpCAM - Brilliant Violet 650 - G8.8 - BioLegend - 118241 - 1/200 |
| | EpCAM - PE-Cy7 - G8.8 - BioLegend - 118216 - 1/1000 |
| | FceR1a - Biotinylated - MAR-1 - ThermoFisher - 13-5898-82 - 1/100 |
| | FLT3 - PE-Cy7 - A2F10.1 - BD Biosciences - 567594 - 1/600 |
| | FLT3 - PE-CF594 - A2F10.1 - BD Biosciences - 562537 - 1/200 |
| | ICAM-1 - PerCP-Cy5.5 - YN1/1.7.4 - BioLegend - 116124 - 1/200 |
| | IL1R2 - PE - 4E 2 - BD Biosciences - 554450 - 1/200 |
| | IL7R - APC - A7R34 - BioLegend - 135012 - 1/200 |
| | IL7R - Brilliant Violet 785 - A7R34 - BioLegend - 135037 - 1/200 |
| | IL7R - PE - A7R34 - Tonbo Bioscience (Fisher Scientific) - 50-1271-U100 - 1/200 |
| | IL7R - Brilliant UltraViolet 737 - SB/199 - BD Biosciences - 564399 - 1/200 |
| | KLRG1 - FITC - 2F1 - ThermoFisher - 11-5893-82 - 1/200 |
| | Ly6C - eFluor 450 - RB6-8C5 - ThermoFisher - 48-5931-80 - 1/200 |
| | Ly6C - Brilliant Violet 711 - HK1.4 - BioLegend - 128037 - 1/800 |
| | Ly6D - eFluor 450 - 49-H4 - Thermo - 48-5974-80 - 1/1600 |
| | Ly6D - FITC - 49-H4 - BioLegend - 138605 - 1/3000 |
| | Ly6G - Biotinylated - 1A8 - BioLegend - 127604 - 1/200 |
| | Ly6G - APC-Fire 810 - 1A8 - BioLegend - 127669 - 1/600 |
| | MHC Class II (I-A/I-E) - PerCP-Cy5.5 - M5/114.15.2 - Tonbo Bioscience (Fisher Scientific) - 65-5321-U100 - 1/400 |
| | MHC Class II (I-A/I-E) - Alexa Fluor 700 - M5/114.15.2 - BioLegend - 107622 - 1/1300 |
| | MHC Class II (I-A/I-E) - Brilliant Violet 480 - M5/114.15.2 - BD Biosciences - 566086 - 1/800 |
| | MHC Class II (I-A/I-E) - Brilliant UltraViolet 563 - M5/114.15.2 - BD Biosciences - 748846 - 1/1500 |
| | NCAM-1 - Brilliant UltraViolet 661 - 809220 - BD Biosciences - 750021 - 1/100 |

NCAM-1 - Brilliant Violet 480 - 809220 (RUO) - BD Biosciences - 748095 - 1/100
Neuropilin-1 - Brilliant Violet 421 - 3 E 12 - BioLegend - 145209 - 1/400
NK1.1 - eFluor 450 - PK136 - ThermoFisher - 48-5941-82 - 1/400
NK1.1 - Biotinylated - PK136 - BioLegend - 108704 - 1/200
NK1.1 - Brilliant Violet 510 - PK136 - BioLegend - 108738 - 1/300
NKp46 - Brilliant Violet 650 - 2911.4 - BioLegend - 137635 - 1/100
PD1 - Brilliant Violet 785 - 29F.1A12 - BioLegend - 135225 - 1/400
Podoplanin - PE-Cy7 - 8.1.1 - BioLegend - 127412 - 1/400
RORgt - PE-C594 - Q31-378 - BD Biosciences - 562684 - 1/400
SiglecF - Brilliant Violet 421 - E50-2440 - BD Biosciences - 562681 - 1/400
SiglecF - PE-Cy7 - S17007L - BioLegend - 155527 - 1/500
SiglecF - PerCP-Fire 806 - S17007L - BioLegend - 155535 - 1/800
SiglecH - Percp-Cy5.5 - 551 - BioLegend - 129614 - 1/200
SiglecH - Alexa Fluor 647 - 551 - BioLegend - 129608 - 1/400
SiglecH - Brilliant UltraViolet 395 - 551 - BD Biosciences - 567814 - 1/800
SIRP-a - Brilliant Violet 605 - P84 - BD Biosciences - 740390 - 1/400
Streptavidin - Brilliant UltraViolet 737 - Streptavidin - BD Biosciences - 564293 - 1/2000
Streptavidin - Brilliant Violet 421 - Streptavidin - BioLegend - 405225 - 1/4000
TCR b - APC-Cy7 - H57-597 - BioLegend - 109220 - 1/400
TCR b - Alexa Fluor 700 - H57-597 - ThermoFisher - 56-5961-82 - 1/300
TCR b - Biotinylated - H57-597 - BioLegend - 109204 - 1/200
TCR gd - Biotinylated - UC7-13D5 - ThermoFisher - 13-5811-85 - 1/200
TCR gd - Brilliant Violet 421 - GL3 - BioLegend - 118119 - 1/400
Ter-119 - Biotinylated - TER-119 - BioLegend - 116204 - 1/200
VCAM-1 - APC - 429 (MVCAM.A) - BioLegend - 105718 - 1/200
XCR1 - Brilliant Violet 650 - ZET - BioLegend - 148220 - 1/200
XCR1 - PerCP-Cy5.5 - ZET - BioLegend - 148208 - 1/100
LIVE/DEAD Fixable Zombie NIR  (Biolegend, 423105) was used at 1/2000 to exclude dead cells.
Anti-CD16/32 (Biolegend, 101320) was used at 1/100 to block binding to Fc receptors prior to cell surface staining.

Validation

All commercially available antibodies are routinely tested by the vendor. Antibody validation is provided on the suppliers 'website listed below:

B220 - Brilliant UltraViolet 737 - https://www.bdbiosciences.com/en-us/products/reagents/flow-cytometry-reagents/research-reagents/single-color-antibodies-ruo/buv737-rat-anti-mouse-cd45r-b220.612838?tab=antibody_details
c-Kit - Brilliant UltraViolet 395 - https://www.bdbiosciences.com/en-us/products/reagents/flow-cytometry-reagents/research-reagents/single-color-antibodies-ruo/buv395-rat-anti-mouse-cd117.564011?tab=product_details
CCR6 - Brilliant Violet 785 - https://sandbox.biolegend.com/en-gb/products/brilliant-violet-785-anti-mouse-cd196-ccr6-antibody-14749
CCR6 - Alexa Fluor 647 - https://www.bdbiosciences.com/en-ca/products/reagents/flow-cytometry-reagents/research-reagents/single-color-antibodies-ruo/alexa-fluor-647-rat-anti-mouse-cd196.557976?tab=product_details
CCR6 - PE-Fluor810 - https://www.biolegend.com/en-us/products/pe-fire-810-anti-mouse-cd196-ccr6-antibody-22722
CCR7 - Alexa Fluor 488 - https://www.biolegend.com/en-us/products/alexa-fluor-488-anti-mouse-cd197-ccr7-antibody-2844
CD11b - Brilliant UltraViolet 395 - https://www.bdbiosciences.com/en-us/products/reagents/flow-cytometry-reagents/research-reagents/single-color-antibodies-ruo/buv395-rat-anti-cd11b.563553?tab=product_details
CD11b - Brilliant Violet 785 - https://www.biolegend.com/en-us/products/brilliant-violet-785-anti-mouse-human-cd11b-antibody-7958
CD11c - APC-Cy7 - https://www.biolegend.com/en-us/products/apc-cyanine7-anti-mouse-cd11c-antibody-3931
CD11c - Brilliant UltraViolet 563 - https://www.bdbiosciences.com/en-us/products/reagents/flow-cytometry-reagents/research-reagents/single-color-antibodies-ruo/buv563-hamster-anti-mouse-cd11c.749040?tab=product_details
CD19 - Brilliant UltraViolet 737 - https://www.bdbiosciences.com/en-us/products/reagents/flow-cytometry-reagents/research-reagents/single-color-antibodies-ruo/buv737-rat-anti-mouse-cd19.612781?tab=product_details
CD19 - Biotinylated - https://www.biolegend.com/en-us/products/biotin-anti-mouse-cd19-antibody-1527
CD19 - Brilliant Violet 421 - https://www.biolegend.com/en-us/products/brilliant-violet-421-anti-mouse-cd19-antibody-7160
CD19 - FITC - https://www.bdbiosciences.com/en-us/products/reagents/flow-cytometry-reagents/research-reagents/single-color-antibodies-ruo/fitc-rat-anti-mouse-cd19.557398?tab=product_details
CD27 - PE-Cy7 - https://www.thermofisher.com/antibody/product/CD27-Antibody-clone-LG-7F9-Monoclonal/25-0271-82
CD27 - FITC - https://www.thermofisher.com/antibody/product/CD27-Antibody-clone-LG-7F9-Monoclonal/11-0271-82
CD31 - FITC - https://www.bdbiosciences.com/en-us/products/reagents/flow-cytometry-reagents/research-reagents/single-color-antibodies-ruo/fitc-rat-anti-mouse-cd31.553372?tab=product_details
CD3e - Biotinylated - https://www.biolegend.com/en-us/products/biotin-anti-mouse-cd3epsilon-antibody-22
CD45 - Brilliant Violet 570 - https://www.biolegend.com/en-us/products/brilliant-violet-570-anti-mouse-cd45-antibody-7452
CD45 - Brilliant Violet 510 - https://www.biolegend.com/en-us/products/brilliant-violet-510-anti-mouse-cd45-antibody-7995
CD45.1 - Brilliant UltraViolet 615 - https://www.bdbiosciences.com/en-us/products/reagents/flow-cytometry-reagents/research-reagents/single-color-antibodies-ruo/buv615-mouse-anti-mouse-cd45-1.751467?tab=product_details
CD45.2 - Violet450 - https://www.fishersci.com/shop/products/violet-450-ms-cd45-2-104-100ug/501055147?searchHijack=true&searchTerm=50-105-5147&searchType=RAPID&matchedCatNo=50-105-5147
CD64 - PE-Cy7 - https://www.biolegend.com/en-us/products/pe-cyanine7-anti-mouse-cd64-fcgammari-antibody-10062
CD88 - Brilliant Violet 421 - https://www.bdbiosciences.com/en-us/products/reagents/flow-cytometry-reagents/research-reagents/single-color-antibodies-ruo/bv421-rat-anti-mouse-cd88.743769?tab=product_details
CD88 - Biotinylated - https://www.biolegend.com/en-us/products/biotin-anti-mouse-cd88-c5ar-antibody-7785
CD88 - PerCP-Cy5.5 - https://www.biolegend.com/en-us/products/percp-cyanine5-5-anti-mouse-cd88-c5ar-antibody-16366
CD90.2 - Biotinylated - https://www.biolegend.com/en-us/products/biotin-anti-mouse-cd90-2-thy1-2-antibody-103
CD90.2 - Brilliant Violet 786 - https://www.bdbiosciences.com/en-us/products/reagents/flow-cytometry-reagents/research-reagents/single-color-antibodies-ruo/bv786-rat-anti-mouse-cd90-2.564365?tab=product_details
CD90.2 - Brilliant UltraViolet 496 - https://www.bdbiosciences.com/en-us/products/reagents/flow-cytometry-reagents/research-

reagents/single-color-antibodies-ruo/buv496-rat-anti-mouse-cd90-2.741046?tab=product_details
CSF1-R - Brilliant Violet 711 - https://www.biolegend.com/en-us/products/brilliant-violet-711-anti-mouse-cd115-csf-1r-antibody-9030
CXCR6 - PE - https://www.biolegend.com/en-us/products/pe-anti-mouse-cd186-cxcr6-antibody-12545
CXCR6 - Alexa Fluor 647 - https://www.biolegend.com/en-us/products/alexa-fluor-647-anti-mouse-cd186-cxcr6-antibody-15231
CXCR6 - APC-Fire750 - https://www.biolegend.com/en-us/products/apc-fire-750-anti-mouse-cd186-cxcr6-antibody-21764
EpCAM - Brilliant Violet 605 - https://www.biolegend.com/en-us/products/brilliant-violet-605-anti-mouse-cd326-ep-cam-antibody-9968
EpCAM - Brilliant Violet 650 - https://www.biolegend.com/en-us/products/brilliant-violet-650-anti-mouse-cd326-ep-cam-antibody-19847
EpCAM - PE-Cy7 - https://www.biolegend.com/en-us/products/pe-cyanine7-anti-mouse-cd326-ep-cam-antibody-5303
FceR1a - Biotinylated - https://www.thermofisher.com/antibody/product/FceR1-alpha-Antibody-clone-MAR-1-Monoclonal/13-5898-82
FLT3 - PE-Cy7 - https://www.bdbiosciences.com/en-us/products/reagents/flow-cytometry-reagents/research-reagents/single-color-antibodies-ruo/pe-cy7-rat-anti-mouse-cd135-flt3.567594?tab=product_details
FLT3 - PE-CF594 - https://www.bdbiosciences.com/en-us/products/reagents/flow-cytometry-reagents/research-reagents/single-color-antibodies-ruo/pe-cf594-rat-anti-mouse-cd135.562537?tab=product_details
ICAM-1 - PerCP-Cy5.5 - https://www.biolegend.com/en-us/products/percp-cyanine5-5-anti-mouse-cd54-antibody-14748
IL1R2 - PE - https://www.bdbiosciences.com/en-us/products/reagents/flow-cytometry-reagents/research-reagents/single-color-antibodies-ruo/pe-rat-anti-mouse-cd121b.554450?tab=product_details
IL7R - APC - https://www.biolegend.com/en-us/products/apc-anti-mouse-cd127-il-7ralpha-antibody-6191
IL7R - Brilliant Violet 785 - https://www.biolegend.com/en-us/products/brilliant-violet-785-anti-mouse-cd127-il-7ralpha-antibody-10803
IL7R - PE - https://www.fishersci.com/shop/products/pe-ms-cd127-a7r34-100ug/502014707?searchHijack=true&searchTerm=50-1271-U100&searchType=RAPID&matchedCatNo=50-1271-U100
IL7R - Brilliant UltraViolet 737 - https://www.bdbiosciences.com/en-us/products/reagents/flow-cytometry-reagents/research-reagents/single-color-antibodies-ruo/buv737-rat-anti-mouse-cd127.612841?tab=product_details
KLRG1 - FITC - https://www.thermofisher.com/antibody/product/KLRG1-Antibody-clone-2F1-Monoclonal/11-5893-82
Ly6C/Ly6G - eFluor 450 - https://www.thermofisher.com/antibody/product/Ly-6G-Ly-6C-Antibody-clone-RB6-8C5-Monoclonal/48-5931-80
Ly6C - Brilliant Violet 711 - https://www.biolegend.com/en-us/products/brilliant-violet-711-anti-mouse-ly-6c-antibody-8935
Ly6D - eFluor 450 - https://www.thermofisher.com/antibody/product/Ly-6D-Antibody-clone-49-H4-Monoclonal/48-5974-80
Ly6D - FITC - https://www.biolegend.com/en-us/products/fitc-anti-mouse-ly-6d-antibody-7769
Ly6G - Biotinylated - https://www.biolegend.com/en-us/products/biotin-anti-mouse-ly-6g-antibody-4772
Ly6G - APC-Fire 810 - https://www.biolegend.com/en-us/products/apc-fire-810-anti-mouse-ly-6g-antibody-21380
MHC Class II (I-A/I-E) - PerCP-Cy5.5 - https://www.fishersci.com/shop/products/percpcy5-5-ms-mhcclassii/502014823?searchHijack=true&searchTerm=65-5321-U100&searchType=RAPID&matchedCatNo=65-5321-U100
MHC Class II (I-A/I-E) - Alexa Fluor 700 - https://www.biolegend.com/en-us/products/alexa-fluor-700-anti-mouse-i-a-i-e-antibody-3413
MHC Class II (I-A/I-E) - Brilliant Violet 480 - https://www.bdbiosciences.com/en-us/products/reagents/flow-cytometry-reagents/research-reagents/single-color-antibodies-ruo/bv480-rat-anti-mouse-i-a-i-e.566086?tab=product_details
MHC Class II (I-A/I-E) - Brilliant UltraViolet 563 - https://www.bdbiosciences.com/en-us/products/reagents/flow-cytometry-reagents/research-reagents/single-color-antibodies-ruo/buv563-rat-anti-mouse-i-a-i-e.748846?tab=product_details
NCAM-1 - Brilliant UltraViolet 661 - https://www.bdbiosciences.com/en-us/products/reagents/flow-cytometry-reagents/research-reagents/single-color-antibodies-ruo/buv661-rat-anti-mouse-cd56-ncam-1.750021?tab=product_details
NCAM-1 - Brilliant Violet 480 - https://www.bdbiosciences.com/en-us/products/reagents/flow-cytometry-reagents/research-reagents/single-color-antibodies-ruo/bv480-rat-anti-mouse-cd56-ncam-1.748095?tab=product_details
Neuropilin-1 - Brilliant Violet 421 - https://www.biolegend.com/en-us/products/brilliant-violet-421-anti-mouse-cd304-neuropilin-1-antibody-8731
NK1.1 - eFluor 450 - https://www.thermofisher.com/antibody/product/NK1-1-Antibody-clone-PK136-Monoclonal/48-5941-82
NK1.1 - Biotinylated - https://www.biolegend.com/en-us/products/biotin-anti-mouse-nk-1-1-antibody-428
NK1.1 - Brilliant Violet 510 - https://www.biolegend.com/en-us/products/brilliant-violet-510-anti-mouse-nk-1-1-antibody-8615
NKp46 - Brilliant Violet 650 - https://www.biolegend.com/en-us/products/brilliant-violet-650-anti-mouse-cd335-nkp46-antibody-15729
PD1 - Brilliant Violet 785 - https://www.biolegend.com/en-us/products/brilliant-violet-785-anti-mouse-cd279-pd-1-antibody-9874
Podoplanin - PE-Cy7 - https://www.biolegend.com/en-us/products/pe-cyanine7-anti-mouse-podoplanin-antibody-6674
RORgt - PE-C594 - https://www.bdbiosciences.com/en-us/products/reagents/flow-cytometry-reagents/research-reagents/single-color-antibodies-ruo/pe-cf594-mouse-anti-mouse-ror-t.562684?tab=product_details
SiglecF - Brilliant Violet 421 - https://www.bdbiosciences.com/en-us/products/reagents/flow-cytometry-reagents/research-reagents/single-color-antibodies-ruo/bv421-rat-anti-mouse-siglec-f.562681?tab=product_details
SiglecF - PE-Cy7 - https://www.biolegend.com/en-us/products/pe-cyanine7-anti-mouse-cd170-siglec-f-antibody-20500
SiglecF - PerCP-Fire 806 - https://www.biolegend.com/en-us/products/percp-fire-806-anti-mouse-cd170-siglec-f-antibody-23384
SiglecH - Percp-Cy5.5 - https://www.biolegend.com/en-us/products/percp-cyanine5-5-anti-mouse-siglec-h-antibody-6927
SiglecH - Alexa Fluor 647 - https://www.biolegend.com/en-us/products/alexa-fluor-647-anti-mouse-siglec-h-antibody-5179
SiglecH - Brilliant UltraViolet 395 - https://www.bdbiosciences.com/en-us/products/reagents/flow-cytometry-reagents/research-reagents/single-color-antibodies-ruo/buv395-rat-anti-mouse-siglec-h.567814?tab=product_details
SIRP-a - Brilliant Violet 605 - https://www.bdbiosciences.com/en-us/products/reagents/flow-cytometry-reagents/research-reagents/single-color-antibodies-ruo/bv605-rat-anti-mouse-cd172a.740390?tab=product_details
Streptavidin - Brilliant UltraViolet 737 - https://www.bdbiosciences.com/en-us/products/reagents/flow-cytometry-reagents/research-reagents/single-color-antibodies-ruo/buv737-streptavidin.612775?tab=product_details
Streptavidin - Brilliant Violet 421 - https://www.biolegend.com/en-us/products/brilliant-violet-421-streptavidin-7297
TCR b - APC-Cy7 - https://www.biolegend.com/en-us/products/apc-cyanine7-anti-mouse-tcr-beta-chain-antibody-4137
TCR b - Alexa Fluor 700 - https://www.thermofisher.com/antibody/product/TCR-beta-Antibody-clone-H57-597-Monoclonal/56-5961-82
TCR b - Biotinylated - https://www.biolegend.com/en-us/products/biotin-anti-mouse-tcr-beta-chain-antibody-269
TCR gd - Biotinylated - https://www.thermofisher.com/antibody/product/TCR-gamma-delta-Antibody-clone-UC7-13D5-

Monoclonal/13-5811-85
TCR gd - Brilliant Violet 421 - https://www.biolegend.com/en-us/products/brilliant-violet-421-anti-mouse-tcr-gamma-delta-antibody-7249
Ter-119 - Biotinylated - https://www.biolegend.com/en-us/products/biotin-anti-mouse-ter-119-erythroid-cells-antibody-1864
VCAM-1 - APC - https://www.biolegend.com/en-us/products/apc-anti-mouse-cd106-antibody-6079
XCR1 - Brilliant Violet 650 - https://www.biolegend.com/en-us/products/brilliant-violet-650-anti-mouse-rat-xcr1-antibody-12421
XCR1 - PerCP-Cy5.5 - https://www.biolegend.com/en-us/products/percp-cyanine5-5-anti-mouse-rat-xcr1-antibody-10397
Zombie NIR  - https://www.biolegend.com/en-us/products/zombie-nir-fixable-viability-kit-8657
CD16/32  - https://www.biolegend.com/en-us/products/trustain-fcx-anti-mouse-cd16-32-antibody-5683

# Animals and other research organisms

Policy information about studies involving animals; ARRIVE guidelines recommended for reporting animal research, and Sex and Gender in Research

| Laboratory animals | AireGreenLantern-T2A-creERT2 and Prdm16-P2A-iCreERT2-T2A-mBaoJin KI mice (referred to as Prdm16creERT2-mBaojin) were generated as described in the manuscript. RorcVenus-creERT2  (referred to as RorcVenus), Il7rcre, Spi1fl/fl, Toxfl/fl, Tcf7fl/fl, Tnfrsf11acre-GFP, Tnfrsf11afl/fl, Tnfsf11fl/fl, Twist2cre, Prdm16fl/fl R26:FlpoERT2 Flt3Frt-ITD, Siglechcre and Tcf4STOP mice have been previously described. Rorgtcre, RorgtGFP, R26lsl-tdTomato(Ai14), Cx3cr1cre, C57Bl/6 (CD45.2) and CD45.1 (PtprcK302E) mice were purchased from Jackson Laboratories. Siglecfcre mice (Cat. NO. NMKI-231128) were purchased from Shanghai Model Organisms. Mice were analyzed at 2 weeks of age unless otherwise stated. |
|---|---|
| Wild animals | No wild animals were used in this study. |
| Reporting on sex | Both male and females were used in the study and we did not observe any sex-specific phenotypes. |
| Field-collected samples | No field-collected samples were used in this study. |
| Ethics oversight | Generation and treatments of mice were performed under protocol 21-05-007, approved by the Sloan Kettering Institute (SKI) Institutional Animal Care and Use Committee. |

Note that full information on the approval of the study protocol must also be provided in the manuscript.

# Flow Cytometry

## Plots

Confirm that:

☒ The axis labels state the marker and fluorochrome used (e.g. CD4-FITC).

☒ The axis scales are clearly visible. Include numbers along axes only for bottom left plot of group (a 'group' is an analysis of identical markers).

☒ All plots are contour plots with outliers or pseudocolor plots.

☒ A numerical value for number of cells or percentage (with statistics) is provided.

## Methodology

| Sample preparation | Sample preparation was performed as described in the methods section |
|---|---|
| Instrument | BD Aria or Cytek Aurora CS for cell sorting, Cytek Aurora for flow cytometry |
| Software | Flowjo v10.10.0 |
| Cell population abundance | Frequencies of cell populations are indicated on the flow plots. |
| Gating strategy | A detailed gating strategy is included in the supplementary information,  main or extended figures. |

☒ Tick this box to confirm that a figure exemplifying the gating strategy is provided in the Supplementary Information.

