## [Peer Review file · Nature]

Ontogeny and transcriptional regulation of Thetis cells

Corresponding Author: Dr Chrysothemis Brown

Version 1:

Reviewer comments:

Referee #1

(Remarks to the Author)

A. Summary of the Key Results

This manuscript describes a progenitor population (termed TCp) proposed to give rise to Thetis cells (TCs)—a class of antigen-presenting cells (APCs) with four transcriptionally defined subsets (TC I–IV) involved in promoting intestinal tolerance. The authors use lineage tracing, single-cell RNA-sequencing, and transcription factor perturbation to suggest that TCp represents a lymphoid-restricted precursor. Additional regulatory mechanisms are described, including roles for PU.1, Relb, and PRDM16 in TC subset differentiation.

B. Originality and significance

The field has recently seen a sharp rise in publications exploring TC-like cells (also referred to as ROR γ ⁺ APCs or tolerogenic dendritic cells) within the last six months. This manuscript does not substantially advance current knowledge and, in part, reuses findings that have already been reported by the same group and others. For example, the role of PRDM16 in these cells has been previously and extensively described (Fu et al., *Nature* 2025; Rodrigues et al., *Cell* 2025), and the novelty here appears limited to suggesting its role across subsets I–IV—an incremental contribution. Similarly, while the addition of PU.1 and Relb is of interest, these transcription factors are well-established regulators in myeloid and epithelial cell biology, so their involvement is not conceptually unexpected.

There also appear to be instances of data reuse or partial overlap with previous publications—for example, the Zeb2 deletion experiment (Cabric et al., *Science* 2025), which reported no effect on TC numbers, is not reconciled with the current data. Additionally, the field is grappling with inconsistent nomenclature—similar cells are described under different names by groups such as Colonna, Littman, and the present authors. Rather than clarifying this, the manuscript does not engage with this issue, making it difficult for non-specialists to interpret the significance of the findings.

Although the manuscript aims to clarify the identity and developmental origin of TCs, many of its conclusions (e.g., lymphoid fate tracing) align closely with existing literature, yet are not strongly supported by the new data presented here. Notably, the manuscript includes no functional assays (which were well explored in previous publications), focusing entirely on origin and differentiation. Given this, the work may be of interest primarily to immunologists and dendritic cell biologists, but it does not currently offer the breadth or impact expected for publication in a broad audience journal.

C. Data & methodology

While the manuscript presents an effort to define the developmental origins of Thetis cells, several methodological issues limit the strength of the conclusions. The following specific points highlight areas where additional data, clarification, or refinement would significantly improve the study:

- Bone marrow chimera experiments (Figure 1): The claim that TCs are hematopoietic in origin is difficult to evaluate based on the current design. The authors assess chimerism at just four weeks post-irradiation, which is not sufficient for full reconstitution, particularly in lymph nodes, which are highly sensitive to radiation and recover slowly. In our experience, at

least eight weeks are typically required to allow stable lymphoid organ reconstitution.

Furthermore, the data are presented without proper gating of immune cell subsets. Each population should be gated individually, and CD45.1/CD45.2 ratios should be quantified per cell type. Including well-characterized immune cells as internal benchmarks would help determine whether the observed chimerism is specific to TCs or reflective of general trends. While I do not endorse parabiosis, alternative strategies such as radiation shielding of the gut and mesenteric lymph nodes could have allowed for better preservation of tissue niches and more accurate reconstitution analysis—particularly relevant for interpreting myeloid contributions. Presenting CD45.1/CD45.2 ratios relative to other immune populations would improve interpretability and help support the conclusion that TCs arise from hematopoietic progenitors.

- FL contribution (Figure 1): The claim that fetal liver (FL) progenitors generate more TCs than bone marrow (BM) progenitors is based only on frequency data. To assess this difference meaningfully, total cell numbers should also be presented. Moreover, without comparison to other immune populations, it's unclear whether the increased frequency is specific to TCs or reflects general differences in reconstitution dynamics, such as homing or expansion potential. As presented, the conclusion that TCs are preferentially derived from FL progenitors is not fully supported by the data.
- RORγt fate mapping model (Figure 1): The RORγt fate-mapping model shows labeling in only ~10–15% of TCs, compared to ~60% in ILC3s. This raises important questions: Is RORγt expressed only transiently in a small subset of TCs, or are some TCs not derived from a RORγt⁺ progenitor at all? Without resolving TC subsets (I–IV), it's impossible to determine whether this labeling is uniform or subset-specific. Disaggregating the data to assess which subsets are labeled—and to what extent—would clarify lineage relationships and the significance of this fate-mapping model.
- Lineage relationships among TC subsets (all manuscript): The manuscript does not present direct evidence that TC I–IV share a common progenitor. Conclusions about shared ontogeny are inferred from transcriptomic similarity, which, while informative, is not equivalent to developmental origin. It would be critical to isolate each TC I–IV subset and TCp, and perform *ex vivo* differentiation assays or adoptive transfer experiments to test the potential of each subset to give rise to TC I–IV. Even if cell numbers are limiting, some functional validation is necessary.
- Progenitor gating (Figure 2): The gating strategies used to isolate CLP, MDP, CDP, and “pre-pDC/tDC” progenitors differ from previously established methods (Sulczewski 2023; Rodriguez 2023; Feng 2022), yet the manuscript does not justify these deviations. Some findings also diverge from expectations—for example, pre-pDC/tDC progenitors giving rise to cDC1s is not consistent with the literature and may reflect contamination or non-specific sorting. The absence of B cells among CLP progeny further raises questions about sorting strategies. All gating strategies, particularly for TCp, should be included in main figures, and references for marker choices should be provided if deviated from previous publications. Additionally, an explanation of why the author's results differ from previous reports would be important for readers.
- Progenitor differentiation assays (Figure 2): The conclusion that CLPs give rise to TCs should include single-cell assays. If feasible, the authors should single-cell sort CLPs and test their ability to generate TCs *in vitro*. As noted above, the authors should evaluate the potential of TCp to give rise to TC I–IV. These experiments would help clarify whether TC I–IV derive from a common precursor, and whether they emerge in a defined sequence or independently.
- Fate-mapping models and lymphoid origin: The interpretation that TCs are lymphoid-derived, based on fate-mapping with IL7R, requires more caution. Prior work (e.g., Reizis and colleagues) has shown that transient expression of lymphoid markers can be induced by certain transcription factors (e.g., E2 family) without establishing true lymphoid lineage commitment. The co-expression of CX3CR1 and IL7R in the same cells—as reported here—suggests that the cells may be myeloid-derived and expressing a lymphoid-associated marker, similar to pDCs and tDCs. Additionally, the recent work by Colonna's group using similar fate-mapping tools (hCD2-driven) arrives at comparable conclusions about the origin of TCs (also called RORγt⁺ DCs), decreasing the novelty of the findings. Unfortunately, this manuscript does not move the field forward in clarifying whether these cells are truly lymphoid-derived or acquire lymphoid-like features during development. Finally, evaluating TC I–IV separately in all the assays, including the IL7R fate-mapping model would provide valuable insight.
- Transcription factor conclusions (Figures 4–5): The authors propose that tissue-derived environmental cues regulate PRDM16 expression and TC subset fate decisions. However, this idea is speculative and not directly tested in the manuscript. If the authors wish to make this point, mechanistic evidence would be needed. Otherwise, the conclusion should be presented more cautiously as a hypothesis rather than a demonstrated result (perhaps reserved for the discussion section).
- Relationship between TC subsets (Figure 5): The developmental model presented in Figure 5j raises some inconsistencies. Earlier, the manuscript suggested that TC I and III are stable based on Aire fate mapping; later, it is proposed that TC II–IV arise from a shared progenitor. It is unclear how these conclusions coexist, and no functional data are provided to test this model. Experiments involving sorting and differentiation of TC I–IV (*in vitro* if adoptive transfer is not feasible) would help clarify whether these subsets are developmentally linked or represent distinct lineages with overlapping transcriptional profiles (especially in the case of TC I). Additionally, the relationship between TCs and RORγt⁺ DCs, as described by other groups and in previous publications by these authors, remains ambiguous. In some sections, TC II–IV are equated with RORγt⁺ DCs; in others (and previous publications by the authors), only TC IV is linked. A clearer reconciliation of terminology, gating strategies, and nomenclature between studies would increase the manuscript's clarity and broader impact.

- Inconsistent terminology and classification (all manuscript): The assertion that only CDP-derived cells should be labeled as dendritic cells is not consistent with field consensus. TCs express FLT3L dependence and are labeled in CX3CR1 models; importantly, they are professional antigen presenting cells capable of inducing de novo regulatory T cells—features typically associated with the dendritic lineage cells. Other groups, including Colonna and Littman, have used these arguments to classify these cells as dendritic cells. Morphological characteristics are insufficient as an argument, especially as pDCs can acquire full dendritic morphology and function upon activation (Reizis, Nat Rev Immunol 2023). Tracing with IL7R-Cre mice also has its problems since (as explained above) this is just a lymphoid feature that may be regulated by well-known E2 transcription factors. Overall, the manuscript does not resolve ongoing ambiguity regarding TC identity and would benefit from more nuanced and cautious language around DC lineage definitions.

- Statistics, transparency, and data analysis (all manuscript): Statistical rigor should be improved in several parts of the manuscript. For example, Figure 4I lacks statistical comparisons, and similar gaps appear elsewhere. Gating strategies for all progenitors—including TcP—should be clearly shown in main figures, as they are foundational to the experiments and necessary for reproducibility and comparison. For the transcriptomic analysis, the authors should explain how they aligned populations and which gene signatures were used to assign identities.

G. References

Key recent studies (Fu et al., Rodrigues et al., Cabric et al., Narasimhan et al., Feng et al., Sulczewski, Rodriguez, etc.) are not adequately cited. Their omission gives a misleading impression of novelty and obscures important field-wide debates. The manuscript would benefit greatly from a more honest engagement with prior work—especially where conclusions are divergent. This includes conclusions regarding nomenclature: while it is understandable that groups may adopt their own naming conventions, the lack of reconciliation now makes it increasingly difficult for non-specialists to follow the literature and understand how various reported cell types relate.

Similarly, the authors claimed that only cDCs are descended from Cx3cr1+ myeloid progenitors, but pDCs are not. However, recent data from the group of Boris Reizis (Feng et al, Immunity 2022) clearly shows that pDCs are also derived from Cx3cr1+ progenitors using in vivo barcoding. Although pDCs can be obtained in vitro from lymphoid progenitors (e.g., Tussiwand and collaborators), their in vivo contribution appears minimal at steady state (Feng et al., Immunity 2022). Similarly, there is currently no evidence that transitional DCs (tDCs) are lymphoid derived (Sulczewski, 2023; Rodriguez 2023). On the contrary, these appear to follow a Cx3cr1-dependent path. These distinctions should be acknowledged to prevent confusion.

Finally, the authors claimed that only CDP-derived cells should be called “dendritic cells” —based on only a few authors’ views and a single citation—this is not universally accepted. This definition would exclude well-established subsets such as monocyte-derived dendritic cells, plasmacytoid/transitional dendritic cells, and DC3s, and risks creating unnecessary confusion for readers outside the immediate field.

H. Clarity and context

The abstract and discussion overstate the novelty and strength of the data. In a rapidly evolving field, it is crucial to clarify how the present study fits with existing work. The inconsistent naming of similar cell types by different groups is already confusing; this manuscript adds to that confusion by failing to reconcile definitions or establish unifying criteria for identity. For instance, Dan Littman and his group recently published a manuscript on the role of PRDM16 in RORyt+ dendritic cells (Nature 2025), while the current manuscript refers to apparently similar cells as TCs. For trainees and non-specialists, these distinctions are difficult to parse, and it is our responsibility as a community to ensure that scientific findings are not obscured by inconsistent terminology. The authors are encouraged to clarify how their cell population relates to others described in the literature, and to reconcile naming, gating strategies, and definitions across groups.

Moreover, the data—including IL7R labeling—do not rule out that TCs are related to dendritic cells. These cells require FLT3L for development, are fully labeled by CX3CR1 fate-tracing, and lack features of ILC3s, as shown by others (including Littman and Colonna) and the authors. Morphology-based arguments are also insufficient; for example, pDCs can adopt dendritic morphology and activate naïve T cells upon stimulation. A particularly critical concern is the interpretation of CLP adoptive transfers. The gating strategies used are not fully aligned with widely accepted protocols, and the recovery of downstream populations diverges from prior reports—suggesting possible contamination or misidentification of precursors. If the central message of the manuscript is the lymphoid origin of TCs, it is essential that these experiments are performed with robust methodology, ideally including single-cell assays and more rigorous lineage resolution.

Finally, one of the most compelling aspects of TCs for a broad Nature audience is their ability to promote immune tolerance during early life, as elegantly demonstrated in recent work (Fu et al., Nature 2025; Rodrigues et al., Science 2025; Cabric et al., Science 2025). In contrast, this manuscript focuses narrowly on developmental origins and does not include new functional data—limiting its appeal to a more specialized readership.

Referee #2

(Remarks to the Author)

Review of Nature manuscript 2025-04-08716A "Ontogeny and transcriptional regulation of Thetis cells".

The authors studied the ontogeny of the recently identified so-called Thetis cells (TC) (I cannot find an explanation in the

literature but perhaps it refers to a cell 'between mTEC and cDC', i. e. one cell type 'married' to the another? Note added in proof; I found the explanation in the discussion). The authors used fate mapping, cell transplantation, RNA seq-derived trajectories, and conditional gene targeting with the aim to identify TC progenitor candidate cells. Overall, this is an interested topic, the experiments make (mostly) sense and are developed in good order. The data suggest that new insights in the ontogeny of TC have been gained.

Questions and concerns:

Fig. 1 legend states 'Thetis cells arise from a ROR γ t⁺ progenitor during early life' when, in fact, these cells also arise from bone marrow (Fig. 1a, b). I don't see a clear and simple comparison of the TC yield (in absolute cell numbers) after bone marrow or fetal liver transplantation, or the number of cells that are actually fate mapped. Percentages in the absence of absolute cell numbers are inconclusive. Do you find normally sized TC compartments after transplantation, or else it may be difficult to argue that there is only an hematopoietic origin?

Fig.1b: What does the background (Venus channel w/o Venus) look like?

Fig. 1f y axis: what is the unit for cell number? (this is a problem in several panels, please check them all).

Ext data Fig. 1e, f (CD45.1 donors into CD45.2 hosts): can you show, akin to panel f, positive reconstitution CD45.1 TC in the same experiment? Without such a control the data in panel f are incomplete.

Lower half of page 4: 'The classical model of hematopoiesis...."recent studies have shown that pDCs, traditionally thought to represent myeloid cells, arise from the common lymphoid progenitor (CLP)16,17, the earliest progenitor that expresses the IL-7 receptor (IL-7R)'. This is not correct. In the original Il7rcre paper by Schlenner et al (Immunity 2010), it was shown that pDC were fully labelled in Il7rCreRFP mice. However, because pDC express the IL7r, it becomes impossible to tell whether the label is derived from a progenitor stage or is acquired in the lineage.

Because this paper relies heavily on the use of Il7rcre mice to detect a lymphoid origin and to make conditional knockouts, it would be fair to cite Schlenner et al in this section for lymphoid fate tracking. On another note, Il7rcre mice were provided to labs in NY through the Rodewald lab via Andrew McKenzie, and not provided by Vinod Balachandran (as suggested in the Acknowledgements).

Fig. 2d. Are only Lin- IL7R fate-mapped and Lin-ROR γ t(Venus)⁺ cells shown? If yes, these data in disagreement with Schlenner et al who showed in their Fig. 2c high labeling in CLP but very low if any labeling in HSC and myeloid progenitors. In light of these data, the statement 'Unsupervised clustering and differential gene expression analysis revealed the full repertoire of hematopoietic progenitors spanning hematopoietic stem cells (HSCs).....highlighting the ability of IL7R⁺ progenitors to give rise to diverse immune cell types during early life' seems incorrect and indeed misleading. It is clear that IL7R Cre fate mapping labels cells downstream and not upstream from lymphoid progenitors (or else the data in this paper would be entirely inconclusive regarding a lymphoid origin of TC). This concern also applies to Fig. 2e. Overall, while the data in Fig. 2d sheds doubt on the nature of cells that were labelled prior to RNA seq, the data may still include TC progenitors. The subsequent transcriptional characterization of putative TC progenitors relies on landscapes and trajectories which, in the absence of direct fate mapping using newly identified specific marker loci (which would be more convincing) seem acceptable.

The knockout data on Zeb2D1+2+3 and Tcf7 seem clear. In contrast, why should TC be independent from Tox if the contribution of Il7rcreToxfl/fl in mixed BM chimeras is only about 1%?

The data identifying PU.1 as a key regulator of TCp differentiation into TC I-IV is really nice.

The sentence 'A recent study demonstrated a role for PRDM16 in non-ILC3 ROR γ t⁺ APC differentiation; however, did not examine individual TC subsets' lacks a reference.

The remained of the paper (Figures 4 and 5) are convincing in my view.

Overall, this is an interesting paper. Some aspects are not fully conclusive but there is a wealth of new information and plenty of leads that can be followed up in the future.

Hans-Reimer Rodewald

Referee #3

(Remarks to the Author)

I co-reviewed this manuscript with one of the reviewers who provided the listed reports.

Referee #4

(Remarks to the Author)

Iza et al. aim to identify and characterize the progenitor and developmental trajectory of Thetis cells (TCs), a newly described population of ROR γ t⁺ antigen-presenting cells (APC) (PMID: 36070798). In their earlier work, the authors showed that TCs possess transcriptional features of both medullary thymic epithelial cells (mTECs) and dendritic cells and comprise four subsets (TCI-TCIV). These cells emerge in the intestinal lymph nodes during the first two weeks after birth and promote the generation of peripheral regulatory T cells (pTregs), establishing tolerance to dietary and microbial antigens in early life.

Although the functions of these ROR γ t⁺ APCs are characterized (PMID: 40228524, 40185101, 40373113), their ontogeny remains unclear. One recent study indicates that ROR γ t⁺ dendritic cells (DCs) have lymphoid origin (PMID: 40185101), whereas another study suggests that tolerogenic DCs (tolDCs) arise from myeloid/dendritic cell progenitors (PMID: 40228524). The present study aims to clarify the progenitor-successor relationship in TC development and to identify the specific progenitors that give rise to TCs. The authors demonstrate that TCI and TCIII are labeled using a novel Aire reporter and lineage-tracing model (AireGreenLatern-creERT2R26Isl-tdTomato). Despite Aire expression, all TC subsets are of hematopoietic origin, shown in bone marrow chimeras. The authors demonstrate that although TCs lack expression of IL-7Ra, nearly all are labeled in IL-7Ra lineage-tracing mice (Il7rcreR26Isl-tdTomato), indicating that TCs originate from an IL-

7Ra⁺ progenitor population. They found transcription factors PU.1 and Prdm16, as well as the environmental cue RANKL, are required for the development of TC subsets. The authors' results support a model in which common lymphoid progenitors CLP give rise to TC I–IV cells via an intermediate progenitor that expressed Rorgt. TC differ from ILC3 in their dependence on PU.1, and the different TC subsets show differential dependence on RANK–RANKL signaling (TC I), and Prdm16 (TC II–IV).

Overall, the study combines lineage tracing, conditional knockout models, and transcriptomic analyses to understand the developmental origin and transcriptional requirement of TC subsets. While the results are of interest, some points require further clarification and experimental support to strengthen the conclusions.

Major

-Other recent work suggested that Prdm16⁺ RORγt⁺ tIDCs, which share features with TC IV, are of myeloid/dendritic cell origin (PMID: 40228524). In contrast, this and other studies (PMID: 40185101) suggest that TCs have lymphoid origin, as suggested by adoptive transfer of CLP. However, the presented experiments do not establish whether all subsets (TCI–TCIV) have a developmental origin from CLP, as TC subsets were not assessed in these experiments. The authors need to establish whether all TC subsets or only some TC subsets derive from CLP.

-The definition of TC progenitors (TCp) used may include lymphoid tissue inducer (LTi) progenitors, which also express RORγt (PMIDs: 33104170, 31882362, 32783932). In Fig. 4f, TCp is gated as Lin⁻CXCR6⁻RORγt⁺, which may include LTi progenitors in fetal liver. In Figure 4i, the gating strategy (Lin⁻CXCR6⁻RORγt⁺) for TCp in adult bone marrow may capture previously described adult bone marrow ILC3 precursors (PMID: 31128961). It is unclear if this TCp population is heterogeneous. If at all possible, the authors should assess the developmental potential of the TCp population to establish whether it can generate all TC subsets, and whether it also gives rise to LTi and ILC3. In addition, the authors could state whether PU.1 deficient mice and other genetic models used in these experiments have normal peripheral lymph nodes or whether they might have defects in LTi development or function.

Other points:

Figure 4a is missing a negative control.

The authors suggest TCs and ILCs arise from distinct developmental trajectories in part based on Il7rCre Tcf7fl/fl and Il7rCre Toxfl/fl mice, where TC numbers remain unaffected. However, to my knowledge, both types of mice still make some ILCs, and Tcf7-deficient mice possess LTi. Have the authors assessed ILC progenitors directly, by adoptive transfer?

The model presented in Figure 5d suggests TC progenitors (TCp) in the fetal liver are initially RORγt⁻ and later transition through RORγt⁺ stages. However, TCp appear to be defined by Rorgt expression, so this is unclear.

Authors could provide quantification of the number of TCp in fetal liver and adult mLN (P7) and bone marrow in Fig 4i.

The authors should perform statistical analysis for the plots where p-values are currently missing, specifically in Figures 1h, 5a, and 5c.

Version 3:

Reviewer comments:

Referee #1

(Remarks to the Author)

The authors have clearly invested substantial effort into revising the manuscript. They have added additional genetic models (including Tcf4STOP/STOP), refined progenitor gating strategies informed by scRNA-seq and showed them in Extended data, and expanded their discussion of nomenclature and prior literature. These changes improve the internal consistency of the developmental model and increase methodological transparency.

However, several of the key concerns raised in the initial review remain only partially addressed. In many instances, the responses rely on extended explanation, reinterpretation, or removal of data rather than providing direct experimental support. The main conclusions are still stated in strong terms, but the underlying evidence continues to have important gaps. The specific areas where the responses are not fully convincing are below:

1. Scope, impact, and lack of functional data

The reviewer understands the conceptual gap the authors aim to address: defining a progenitor (TCp) and developmental pathway for Thetis cells and clarifying their relationship to LTi cells and other progenitors. Nevertheless, two central issues remain. First, even with the lineage-tracing and transfer experiments, the data do not convincingly and definitively establish that these cells are lymphoid-derived; the evidence is suggestive but not conclusive. Second, by design, the revised manuscript still contains no new functional data. Given that previous high-profile studies have already established many of the functional properties and tolerogenic roles of these cells, a purely ontogeny-focused manuscript without new *in vivo* function further narrows the scope and confines the primary impact to researchers interested specifically in developmental

lineage, rather than the broader readership.

2. Novelty of transcription factor findings

In their response, the authors compare their analysis of PU.1, RelB, and other transcription factors in TC development to prior work mapping discrete regulatory regions in the *Zeb2* locus for cDC2 specification (i.e., the Ken Murphy study in *Nature*). The reviewer does not find this comparison appropriate. The *Zeb2* study provided a carefully designed, mechanistically precise dissection of specific enhancer regions that clearly reshaped understanding of cDC2 specification. By contrast, in the present manuscript, the roles of PU.1 and RelB are not resolved at the same mechanistic depth; the authors do not map specific regulatory elements or provide an equivalent level of causal detail. While the PU.1 and RelB data may be of interest within the TC ontogeny framework, the novelty and conceptual impact are not comparable to the cited *Zeb2* work.

3. Handling and removal of data

The *Zeb2* deletion data were initially presented and then removed in the revision. It remains unclear why these data were considered sufficiently novel and robust to include at first, but are now omitted. If the authors stand by the scientific novelty of those results, a more explicit rationale for their removal is warranted. Conversely, if the data were judged to be insufficiently robust or potentially misleading, that should also be clearly stated. As it stands, the decision appears strategic rather than scientific, and this does not increase confidence in the overall narrative. Similar considerations apply to other datasets that have been removed or substantially reinterpreted between versions; greater transparency about these decisions would be helpful.

4. Bone marrow chimera design and reconstitution timing

The authors maintain that 4-week bone marrow chimeras are adequate because they are focusing on innate populations, which they state reconstitute by 2 weeks. The reviewer does not agree that this argument is sufficient. There is a substantial body of literature indicating that full reconstitution of lymphoid organs, particularly with respect to migratory dendritic cells and proper lymph node architecture, typically requires ~8 weeks. Given that migration, stromal integrity, and tissue organization are central to DC priming of T cells and tolerance mechanisms (and given that TCs are proposed to be key tolerogenic APCs) the structural and temporal context of the lymph node should be carefully controlled. Using 4-week chimeras may be acceptable as one data point, but the reviewer does not consider this timing fully adequate to support strong conclusions about hematopoietic origin and relative contributions of distinct progenitors, especially when claims are framed as definitive.

5. Evidence for lymphoid origin and adoptive transfer benchmarks

Lymphoid origin of these cells has already been suggested by previous studies using lineage-tracing. The main additional element in this manuscript is the adoptive transfer of CLP and other progenitors. However, the benchmarks and outcomes of these transfers, in the authors' hands, do not clearly recapitulate key features reported by others, raising doubts about how confidently one can interpret these experiments. Moreover, the authors state that they were unable to perform single-cell differentiation assays due to technical limitations. While the technical challenges are understood, for a paper that focuses almost exclusively on ontogeny and aims for *Nature* level impact, the absence of single-cell or clonal-resolution lineage data is a significant limitation. Simply stating that it was not possible is not fully satisfactory. Alternative strategies (e.g., monolayers, stromal or bone marrow feeder layers, or modified culture systems) could have been explored more deeply. In the absence of these, the strength of the conclusion that TCs arise from lymphoid progenitors remains weaker than the manuscript implies.

Related to this, the way CLP-derived progeny is presented is still confusing (even with the included explanation and extended data). It appears, from the plots, that CLP transfers generate essentially every cell type, including substantial numbers of cDC1s and cDC2s. If this is truly the case, it raises concerns about purity, benchmarking, and specificity. If instead this is largely a matter of plotting strategy and gating (for example, focusing on Lin- MHCII⁺ cells and underrepresenting lymphocytes), then this needs to be made much clearer not in the text, but in the figure itself. As currently presented, the data are difficult to interpret and risk giving a misleading impression of both the breadth and specificity of CLP output. Similar concerns apply to the reported outputs of pre-pDC/tDC populations, where the emergence of monocytes (and cDC1s as mentioned before) is unexpected and should be explicitly benchmarked and justified.

6. Lineage bias, pDC ontogeny, and TCF4 interpretation

The manuscript now places substantial emphasis on TCF4 in balancing pDC versus TLP/TC fates and uses this to reinforce a model in which TCs share a progenitor with pDCs. However, the ontogeny of pDCs itself remains an active area of debate, with evidence from both sides of the debate: some groups arguing for myeloid origin and others favoring lymphoid contributions. The authors acknowledge this controversy, yet the figures and text (e.g. in Figures 1e and 3i) are framed in a way that effectively assumes a lymphoid identity for pDCs, which then conveniently supports the view that TCs are also lymphoid. In the reviewer's opinion, this approach is biased. Using a debated lineage assignment for pDCs as a scaffold to support a firm lymphoid classification for TCs is not scientifically robust in the absence of more definitive experimental evidence.

More broadly, the overall argument for lymphoid origin relies on a combination of IL7R/hCD2 fate mapping, scRNA-seq clustering, and adoptive transfers whose benchmarks and gating choices are not entirely convincing. IL7R expression alone does not define lymphoid origin, as the authors themselves note, and transcriptomic resemblance is inherently suggestive rather than conclusive. Given these caveats, the reviewer does not agree that the manuscript has "definitively" established lymphoid origin. A more cautious framing would be appropriate.

In summary, while the manuscript is improved and now presents a clearer and more internally consistent developmental model, many of the original concerns remain. The conclusions continue to overstate what can be definitively supported by the available data, and the overall contribution, though valuable within a specialized developmental field, does not substantially extend beyond existing knowledge generated by several recent studies.

Referee #2

(Remarks to the Author)

I have read the revised version of this manuscript and the authors' response. Overall, the authors have addressed each point that I raised in the first round of review, either by new data, new analyses, or by way of explanation. I remain positive in my support of the paper.

The one thing that really needs attention now is the fact that the authors claim to have evidence for a common progenitor for TC and LTi cells. However, they cannot prove that one progenitor has such dual fates because this would require single cell assays in which both lineages arise from a single progenitor. This could also be achieved by barcoding. In the absence of such evidence for dual fate progenitors as opposed to single fates within the progenitor population they authors should make clear in the entire paper that they describe a dual fate progenitor population. For instance, in the summary, the statement "Here, we identify a RORgt+ TC and lymphoid tissue inducer (LTi) cell progenitor (TLP) that differentiates into the immediate RORgt+ TC progenitor (TCP) and the LTi progenitor (LTiP)" suggest a bipotent progenitor which, in light of the point raised above, the authors have not shown. To be correct, the author should call it a progenitor population and this ambiguity or open question should be added to the discussion.

HR Rodewald

Referee #4

(Remarks to the Author)

The revised manuscript addresses all of my questions and criticisms. As suggestions, Extended data 7d showing that all Thetis cell populations are evident in mice reconstituted with CLP could be in the main figures and not in supplement. The CLP definition used here is (appropriately) a refined definition termed ALP published by the Weissman lab (PMID: PMID: 19833765); this key paper could be referenced. Some figures still use Latin instead of Greek letters for "gamma" in Rorgt, for e.g. Figure 5, there may be others. This could be corrected.

The work provides a wealth of experimental data from transcriptional analysis, adoptive transfer, in vitro cell culture, in vivo lineage tracing, and transcription factor deficient mice, together indicating that Thetis cells develop from lymphoid progenitors. The work is done to a high standard. The revised manuscript includes new data indicating that LTi and Thetis cells share a developmental pathway, distinct from other ILCs that also derive from lymphoid progenitors. It identifies transcription factor dependencies for many of the proposed developmental steps. The data here revise our thinking about lymphoid progenitor-derived cells to include tolerogenic APCs whose main function is to present antigens to CD4 T cells in lymph nodes. This is an important insight, and I am strongly supportive.

Avinash Bhandoola

Version 4:

Reviewer comments:

Referee #4

(Remarks to the Author)

I continue to be strongly supportive of this interesting and well-performed study.

We would like to thank the reviewers for the constructive and encouraging suggestions, which have significantly improved our paper. In the revised manuscript we now include new data that addresses Thetis cell (TC) ontogeny, particularly in terms of clarifying the lymphoid-derived nature of TCs, and examining relationships to transitional dendritic cells (tDCs)/plasmacytoid DCs (pDCs), innate lymphoid cell progenitors (ILCPs) and lymphoid tissue inducer cell progenitors (LTiPs) in more depth, using new lineage tracing approaches, genetic epistasis experiments and ex vivo lymph node cultures. This has also led to an unexpected finding: TCs and LTi cells share a common progenitor and developmental pathway.

Although LTi cells are often considered a subset of type 3 innate lymphoid cells (ILC3s), several strands of evidence support a model, first proposed by the Bendelac lab, whereby ILCs and LTi cell progenitors arise via distinct developmental pathways (PMID 24509713, 27749818, 33104170, 30926235 and Leger et al. *Immunity* 2025). While these studies have demonstrated that the LTi progenitor is developmentally distinct from the ILC progenitor, the precise developmental origin for these cells was not known. We now show that the ROR γ ⁺ progenitor identified in the fetal liver (FL) represents a common TC and LTi cell progenitor, which we call the TC and LTi cell progenitor (TLP), that gives rise either to LTiPs or TC progenitors (TCPs), with TCP differentiation occurring predominantly within lymph nodes. We show that this bifurcation in cell fate is regulated by PU1 (TCs) vs Tox/TCF7 (LTi cells). Using a novel Siglec-F fate mapper to lineage trace the progeny of the TLP we definitively show that LTi cells and all TC subsets are descended from this cell type. In addition, we identify unique markers that distinguish TCP from LTiP in the lymph nodes (IL1R2 vs PD1) and validate this with index sorting SS3, and perform ex vivo cultures of TCP using mesenteric lymph node slices to support their differentiation, establishing the TC restricted nature of the TCP.

Thus, our study delineates the developmental pathway for both TCs and LTi cells. These findings refine the classical model of hematopoiesis and will be invaluable for future studies of TC function. They will also be invaluable for the development of therapeutic strategies aimed at enhancing TC differentiation in vivo or the generation of ex vivo TCs.

Referees' comments:

Referee #1 (Remarks to the Author):

A. Summary of the Key Results

This manuscript describes a progenitor population (termed TCp) proposed to give rise to Thetis cells (TCs)—a class of antigen-presenting cells (APCs) with four transcriptionally defined subsets (TC I–IV) involved in promoting intestinal tolerance. The authors use lineage tracing, single-cell RNA-sequencing, and transcription factor perturbation to suggest that TCp represents a lymphoid-restricted precursor. Additional regulatory mechanisms are described, including roles for PU.1, Relb, and PRDM16 in TC subset differentiation.

B. Originality and significance

The field has recently seen a sharp rise in publications exploring TC-like cells (also referred to as ROR γ ⁺ APCs or tolerogenic dendritic cells) within the last six months. This manuscript does not substantially advance current knowledge and, in part, reuses findings that have already been reported by the same group and others. For example, the role of PRDM16 in these cells

has been previously and extensively described (Fu et al., Nature 2025; Rodrigues et al., Cell 2025), and the novelty here appears limited to suggesting its role across subsets I–IV—an incremental contribution. Similarly, while the addition of PU.1 and Relb is of interest, these transcription factors are well-established regulators in myeloid and epithelial cell biology, so their involvement is not conceptually unexpected.

We would like to begin by thanking the reviewer for their helpful and constructive comments. Regarding novelty, to our knowledge, no published study has identified the TC progenitor or described its developmental pathway. We believe these findings represent a substantial advance in current knowledge given disparate findings from previous studies concerning the nature of the TC lineage, with some studies suggesting myeloid ontogeny and others lymphoid ontogeny. We also believe it is vital to understand the ontogeny of these cells so as to enable research into their function and potential therapeutic opportunities. We believe our findings are of considerable interest, given the therapeutic potential of TCs and their developmentally restricted differentiation, and will pave the way to future studies examining the transcription factors that determine progressive TC commitment during hematopoiesis. We have edited the text to make the wider implications and importance of these findings clearer.

The role of PRDM16 is included here to illustrate the bifurcation in TC I vs TC II–IV cell fate, with distinct transcriptional regulators determining these differentiation pathways. In addition, we chart the temporal appearance of PRDM16 during TC development which provides important insights into the hierarchy of transcription factors that govern TC fate. In the revised version of the manuscript, we use our new *Prdm16^{cre-ERT2}* allele to address the stability of TC subsets, demonstrating that TCs are stable and do not convert to cDCs (**new Extended Data Fig. 13a–b**).

Regarding the expected nature of PU.1 and Relb regulation in TC differentiation, understanding the transcriptional regulation of TC differentiation is key to understanding their development, and we do not believe that it requires identification of a new transcription factor, not previously implicated in myeloid or epithelial cell biology, in order to be impactful and important. For example, previous studies published in Nature delineated a role for particular regions in the *Zeb2* locus for cDC2 differentiation (PMID 35732734). This study revealed critical insights into cDC1 and cDC2 specification, despite the fact that *Zeb2* was already a well-established regulator of myeloid cell differentiation.

In the revised manuscript we provide more insights into the role of PU.1, demonstrating that LTi cell and TC progenitors share a developmental pathway, with a key role for PU.1 in determining TC fate. With regards to the role of Relb, we believe it is conceptually interesting that a hematopoietic antigen-presenting cell subset exhibits overlapping transcriptional dependencies with a medullary thymic epithelial cell, emphasizing the unusual nature of the TC lineage.

In the revised manuscript we also establish a cell intrinsic role for TCF4 in suppressing differentiation of the TLP, which has not been described before (**new Fig. 3d–f and Extended Data Fig. 7e**).

There also appear to be instances of data reuse or partial overlap with previous publications—for example, the *Zeb2* deletion experiment (Cabric et al., Science 2025), which reported no effect on TC numbers, is not reconciled with the current data. Additionally, the field is grappling with inconsistent nomenclature—similar cells are described under different names by groups such as Colonna, Littman, and the present authors. Rather than clarifying this, the manuscript

does not engage with this issue, making it difficult for non-specialists to interpret the significance of the findings.

We apologize for any confusion, but there is no data reuse across this paper and others we have published.

In Cabric *et al.*, we explored a role for cDC2s in food specific pTreg differentiation using *Zeb2* enhancer mutant mice but did not specifically address TC abundance. As stated in that paper,

In Zeb2 triple enhancer mutant (Zeb2 Δ 1+2+3) mice, which lack cDC2 (28), OT-II pTreg differentiation was not affected (Fig. 4C and fig. S9A), despite an almost complete loss of cDC2s in the cLN which served as a representative gut lymph node (fig. S9B).

Since we showed that TC IV is the pTreg inducing subset, it is logical to conclude that TC IV differentiation and function must be conserved in *Zeb2 Δ 1+2+3* mice; however to exclude a role for *Zeb2* in regulation of the TC lineage, in particular differentiation via the *Zeb2*-dependent tDC progenitor, we would need to examine all TCs, rather than inferring that TC differentiation must be normal because of the normal pTreg cell numbers in these mice. We specifically did not include an analysis of TCs in *Zeb2 Δ 1+2+3* mice in Cabric *et al.* because we were including the data in this manuscript. However, we have now excluded a tDC precursor - TC progeny relationship more directly using new mouse models (*Tcf4*^{STOP/STOP} FL and *Tcf4*^{STOP/STOP} FL chimeras; **new Fig. 3d–f** and **Extended Data Fig. 7e**), therefore we have removed the *Zeb2* data.

Aside from the *Zeb2* data, we have only used previously published scRNA or ATAC-seq data to generate or test new hypotheses, or for orthogonal validation, in experiments related to the identity of the progenitors, their molecular signatures and transcriptional regulators. We believe that everything in the paper is novel and substantially advances our understanding of TC differentiation.

With regards to the nomenclature, we were trying to avoid confusion by maintaining the same nomenclature that was used in the original identification of the TC lineage and the four TC subsets (TC I-IV) in our study from 2022 (PMID 36070798). We also used the same nomenclature and subset definitions in our subsequent study by Cabric *et al.* in 2025 (PMID 40373113) where we further defined and validated our gating strategies for TC I-IV subsets. However, we note that since the original study, TC subsets have been renamed ROR γ ⁺ DC I-IV (Rodrigues *et al.* 2025), ROR γ ⁺ eTAC I-3 (Sun *et al.* 2025) or grouped together as one cell type named toIDCs (Fu *et al.* 2025). Cognizant of the confusion that this has caused in the field, we integrated all of the published scRNA-seq analyses of ROR γ ⁺ APC subsets across 8 studies (PMID 36070798, 36071167, 36071169, 40185101, 39091750, 40298935, 40373113 and 34767455) and reconciled the nomenclature in a review recently published in *Nature Immunology* (PMID 41107526). We show that all non-ILC3/LTi ROR γ ⁺ APCs map to one of four previously described TC subsets (**Reviewer Fig. 1**). We have now referenced that data in the Introduction of the revised manuscript, which is also included as. We also now discuss the naming of TCs in the Discussion.

Although the manuscript aims to clarify the identity and developmental origin of TCs, many of its conclusions (e.g., lymphoid fate tracing) align closely with existing literature, yet are not strongly

supported by the new data presented here. Notably, the manuscript includes no functional assays (which were well explored in previous publications), focusing entirely on origin and differentiation. Given this, the work may be of interest primarily to immunologists and dendritic cell biologists, but it does not currently offer the breadth or impact expected for publication in a broad audience journal.

We think our paper offers substantial advances in the understanding of the ontogeny of TCs, which will facilitate functional and therapeutic investigations in the future. Although one study has suggested lymphoid origin based on IL-7R or hCD2 lineage tracing and in vitro cultures of progenitors (PMID 40185101), other studies, most notably Fu et al. (PMID 39091750) and Sun et al. (PMID 40298935) have suggested that these cells are myeloid cells. To our knowledge, no other studies have identified the immediate TCP nor identified the developmental pathway from the common lymphoid progenitor (CLP) to the TCP, including the identification of a common Thetis cell/LTi cell progenitor (here named TLP). In the revised manuscript, we also reveal a previously unknown cell intrinsic role for TCF4 in suppressing TC differentiation (**new Fig. 3d-f** and **Extended Data Fig. 7e**). Moreover, to our knowledge, no other studies have identified the transcription factors that determine TC vs LTi cell fate (PU.1) or the signaling pathway that regulates TC I differentiation (RANKL/RANK/NF-kB). Given that TC IV is the critical pTreg inducing APC subset, and tolerogenic APCs are at the heart of immune tolerance therapeutics, there is a clear need to understand their ontogeny and the signals that promote their development. Our new data revealed developmental relationships between TCs and LTi cells and suggest relationships between pDCs and TLP which warrant future investigation, thus covering a broad swathe of hematopoiesis. As such, we believe that our findings are of interest to a broad audience. Additional functional investigation of TC subsets requires characterization of a number of new genetic models as well as functional in vivo experiments and scRNA-seq datasets, which we believe is beyond the scope of the current manuscript.

C. Data & methodology

While the manuscript presents an effort to define the developmental origins of Thetis cells, several methodological issues limit the strength of the conclusions. The following specific points highlight areas where additional data, clarification, or refinement would significantly improve the study:

- Bone marrow chimera experiments (Figure 1): The claim that TCs are hematopoietic in origin is difficult to evaluate based on the current design. The authors assess chimerism at just four weeks post-irradiation, which is not sufficient for full reconstitution, particularly in lymph nodes, which are highly sensitive to radiation and recover slowly. In our experience, at least eight weeks are typically required to allow stable lymphoid organ reconstitution.

We agree that 6-8 weeks reconstitution is typically used for stable lymphoid organ reconstitution, primarily for T cells, which take longer to reconstitute. However, here we are considering innate cell types, such as cDCs, pDCs and ILCs which have reconstituted by 2 weeks. Before selecting the timepoints for the experiments, we compared TC reconstitution at different time points and did not observe significant differences in TC or DC subset numbers at 4 vs 8 weeks (**Reviewer Fig. 2**) therefore we continued with the 4 week timepoint for analysis of TCs and other innate immune cells. I understand the concern that we could have missed a population of hematopoietic cells that had not yet reconstituted by looking too early; however,

since we identified all TC subsets at the 4 week timepoint, we do not think our conclusions are affected by this. We can include this figure in the main manuscript if needed.

Furthermore, the data are presented without proper gating of immune cell subsets. Each population should be gated individually, and CD45.1/CD45.2 ratios should be quantified per cell type. Including well-characterized immune cells as internal benchmarks would help determine whether the observed chimerism is specific to TCs or reflective of general trends. While I do not endorse parabiosis, alternative strategies such as radiation shielding of the gut and mesenteric lymph nodes could have allowed for better preservation of tissue niches and more accurate reconstitution analysis—particularly relevant for interpreting myeloid contributions. Presenting CD45.1/CD45.2 ratios relative to other immune populations would improve interpretability and help support the conclusion that TCs arise from hematopoietic progenitors.

We apologize for the confusion. We had gated each population individually and compared CD45.1/2 ratios per cell type, as suggested, using well characterized immune cells such as NK cells and monocytes as benchmarks (**Fig. 4d**, **Extended Data Fig. 9f** and also for **new Fig. 3d–f**). We have included representative flow plots in the revised manuscripts to make this clearer and provided more detailed descriptions in the figure legends (**new Extended Data Fig. 1a,d**, **new Fig. 3e,f**, **new Extended Data Fig. 9h**)

The only experiments where we gate differently are:

Extended Data Fig. 9e: Here, we have different markers (CD45.2 RORgt-GFP vs CD45.1/2 RORc-Venus) to identify RORgt null and wildtype TCs respectively. Therefore, to determine the ratios, we gate on each CD45.1 or CD45.2 RORgt⁺MHCII⁺ population separately, determine cell counts for TCs and ILCs and calculate ratios.

We agree that parabiosis is unlikely to be a successful surgery for mice <P14 when pups are still receiving milk from the mothers. We have tried irradiating young mice and have found considerable effects on growth of the mice, even with shielding of the gut and mesenteric lymph nodes. Despite this, we believe that competitive bone marrow chimerism is an appropriate way to measure the intrinsic potential of the progenitors to differentiate into TCs, given that we show in Fig. 1 that all TC subsets can be generated from bone marrow chimeras following irradiation and transfer. This method has been widely used by other studies of immune cell progenitors.

- FL contribution (Figure 1): The claim that fetal liver (FL) progenitors generate more TCs than bone marrow (BM) progenitors is based only on frequency data. To assess this difference meaningfully, total cell numbers should also be presented. Moreover, without comparison to other immune populations, it's unclear whether the increased frequency is specific to TCs or reflects general differences in reconstitution dynamics, such as homing or expansion potential. As presented, the conclusion that TCs are preferentially derived from FL progenitors is not fully supported by the data.

In the revised manuscript, we have included cell counts for TCs and other innate immune cell populations (**new Fig 1b** and **Extended Data Fig. 1f**). This analysis confirmed that the absolute number of TCs is higher in FL vs BM chimeras and this increase in cell number is specific to TCs and LT_i cells.

- RORgt fate mapping model (Figure 1): The RORgt fate-mapping model shows labeling in only ~10–15% of TCs, compared to ~60% in ILC3s. This raises important questions: Is RORgt

expressed only transiently in a small subset of TCs, or are some TCs not derived from a ROR γ ⁺ progenitor at all? Without resolving TC subsets (I–IV), it's impossible to determine whether this labeling is uniform or subset-specific. Disaggregating the data to assess which subsets are labeled—and to what extent—would clarify lineage relationships and the significance of this fate-mapping model.

We apologize for the confusion. The difference in % of fate-mapped cells relates to the tamoxifen dosing and the difference in lifespan/turnover between TCs and ILC3s. In this experiment, we gave one dose of 4-OHT rather than continuous dosing because we wanted to understand when the TC progenitor first arises through 'time-stamping'. Thus, the labeled cells reflect the cells that were derived from the ROR γ ⁺ TC progenitors that were present at the time of tamoxifen administration, and the remaining unlabeled cells reflect both cells that were descended from unlabeled ROR γ ⁺ progenitors as well as output from new ROR γ ⁺ TC progenitors that arose after tamoxifen administration. Thus, any reduction in labeling, below the 60% labeling that we typically observe 24hrs post tamoxifen administration, results from emergence of TCs from unlabeled ROR γ ⁺ progenitors that acquire ROR γ and differentiate after the time-stamping. In contrast, LT_i cells are thought to be sustained by self-renewal (PMID 26472762), thus the labeled cells are not diluted by de novo hematopoietic output to the same degree. However, we realize that accurate interpretation of this data requires a temporal assessment of TC decline and turnover which is difficult to assess given that administration of tamoxifen after the appearance of TCs leads to simultaneous labeling of both proliferative TCP and mature TC subsets. We have therefore removed this data from the revised manuscript while we evaluate the lifespan of TCs using *Prdm16^{creERT2}R26^{sl-tdtomato}* mice alongside the development of new genetic strains for selective and temporal labeling of TCPs.

Lineage relationships among TC subsets (all manuscript): The manuscript does not present direct evidence that TC I–IV share a common progenitor. Conclusions about shared ontogeny are inferred from transcriptomic similarity, which, while informative, is not equivalent to developmental origin. It would be critical to isolate each TC I-IV subset and TCP, and perform ex vivo differentiation assays or adoptive transfer experiments to test the potential of each subset to give rise to TC I-IV. Even if cell numbers are limiting, some functional validation is necessary.

We note that all TC subsets are labeled following embryonic/P1 4-OHT administration to *Rorc^{creERT2}R26^{sl-tdTomato}* mice indicating that each subset has arisen from a ROR γ ⁺ progenitor (**Reviewer Fig. 3**). The difference in proportion of labeled cells across subsets, in particular TC IV, likely reflects the different lifespans for this subsets which we are currently evaluating as noted above. While it is possible that the cells arise from distinct ROR γ ⁺ progenitors, we note that our scRNA-seq analysis only identified one ROR γ ⁺ FL cluster, even after fine subclustering (**Fig. 1j-l**). In addition, all TC subsets are affected by loss of PU1 in ROR γ ⁺ cells, suggesting that PU1 regulates differentiation of TC subsets from a common progenitor (TCP). Furthermore, every lineage tracing approach that we have tested such as IL7R-fate mapping or Siglech-fate mapping demonstrated an almost identical % of labeled cells in each TC subset (**Fig. 2g, 3h, Extended Data Fig. 2b**). In addition, TC subsets are equally affected by TCF4 deficiency (**new Extended Data Fig. 7e**). Moreover, TC II-IV share expression and dependency on PRDM16 as well as IRF8 (PMID 40373113) and Runx/Cbfb2 (PMID 40909748). Thus, it is extremely unlikely that TC II-IV arise through distinct developmental pathways from discrete progenitors. However, to definitively address this we searched for genes that distinguish the FL ROR γ ⁺ progenitor (now named TLP) and found that these cells express SiglecF gene transcript and protein (**new**

Fig. 2f and **Extended Data Fig. 5c**). Within our FL dataset, we observed the expected expression by granulocyte progenitors and a small proportion of pre-DCs, but did not identify significant SiglecF expression in candidate TC or LTi progenitor cell types (**Extended Data Fig. 5d**), indicating that *SiglecF^{cre}* mice could be used to lineage trace TLP cells. Excitingly, analysis of *SiglecF^{cre}R26^{sl-tdTomato}* mice demonstrated equivalent frequencies of labeled cells (~40-50%) in all TC subsets as well as LTi cells (**new Fig 2g**). By contrast, only 10-20% of ILC subsets were labeled. Overall these lineage tracing approaches indicate that LTi cells and TCs arise from the common ROR γ ⁺SiglecF⁺ TLP.

As per the reviewer's suggestion, we attempted to adoptively transfer TC progenitors to ascertain whether they give rise to all 4 subsets; however, these experiments are extremely technically challenging given the limited number of FL or LN TLP and TCP that we are able to sort. Thus, we were unable to recover any of the transferred cells or their progeny from recipients.

We therefore turned to ex vivo differentiation assays to assess the differentiation potential of the two TC progenitor populations defined in this study: TLP and TCP. Given that TCs are enriched in gut lymph nodes during early life, we generated lymph node slices using lymph nodes from CD45.1 2-3 week old mice, and cultured FACS-isolated CD45.2 TCP from LNs using our new gating strategy, which we validated by SS3 index sorting (**new Fig 2a-d**). Excitingly we found that these lymph node slices support differentiation of the TCP, which exclusively gives rise to TCs expressing TC II-IV marker genes including high levels of EpCAM and CD11c (**Fig. 2e**). Critically, these cells do not give rise to CXCR6⁺ LTi cells or cDCs. Of note, although we could not recover TC I cells, we note that the lymph node cells do not remain viable for even brief periods of culture and we have likely lost LTo stromal cells, which are difficult to culture in vitro and are the key cell type required for TC I differentiation. In addition, to ensure a pure population of progenitors, we cannot sort the TCP with lowest expression of ROR γ ^t due to contamination with ROR γ ^t⁻ cells. As shown in **Fig. 4a**, TC I have the lowest levels of ROR γ ^t and likely arise from the ROR γ ^t^{lo} progenitors before upregulation of ROR γ ^t and PRDM16. However, we believe that our additional data with Siglec-F lineage tracing alongside the ex vivo culture data provide strong evidence that the TCP is a common progenitor that can give rise to all four TC subsets.

With regards to culture of terminally differentiated TC I-IV subsets, we have not identified culture conditions that support the survival or expansion of these cells, similar to e.g. terminally differentiated epithelial cells or dendritic cells. We note that Aire fate-mapping experiments in **Extended Data Fig. 13** indicated that TC subsets were not plastic. Therefore, even if we could "force" a TC subset to transdifferentiate in vitro, this would likely not represent the physiological pathway for TC differentiation.

Progenitor gating (Figure 2): The gating strategies used to isolate CLP, MDP, CDP, and "pre-pDC/tDC" progenitors differ from previously established methods (Sulczewski 2023; Rodriguez 2023; Feng 2022), yet the manuscript does not justify these deviations. Some findings also diverge from expectations—for example, pre-pDC/tDC progenitors giving rise to cDC1s is not consistent with the literature and may reflect contamination or non-specific sorting. The absence of B cells among CLP progeny further raises questions about sorting strategies. All gating strategies, particularly for TCp, should be included in main figures, and references for marker

choices should be provided if deviated from previous publications. Additionally, an explanation of why the author's results differ from previous reports would be important for readers.

We had provided an explanation for marker choices in the text as well as **Fig. 1i** and **Extended Data Fig. 7b**, but we apologize that this was not clear. The rationale for designing new gating strategies was to ensure that our choice of markers reflected accurate expression of these markers by FL progenitors, which has not been established in the three cited studies that instead characterized bone marrow or splenic progenitors. Moreover, the markers/gating strategies are not the same across the cited studies, highlighting the lack of established methods for the above progenitors (see **Reviewer Table 1**). Since our gating strategy was informed by the scRNA-seq analysis of FL progenitors, we believe that this is the 'gold standard' as it ensures pure populations of FL progenitors. Of note, all of the cited studies (Sulczewski et al., Rodrigues et al. and Feng et al.) developed their own gating strategy for pre-pDC/tDC isolation based on either unbiased 255 LEGENDplex flow cytometry or scRNA-seq and cluster gene expression analysis, similar to our approach.

Nonetheless, we do not believe that the gating strategies differ significantly (see **Reviewer Table 1**) and, if anything, our gating strategy would result in purer populations of progenitors. Lui et al (2022) identified CLP in the BM as $\text{Lin}^- \text{CD135}^+ \text{CD127}^+ \text{CD117}^{\text{int}} \text{Sca-1}^{\text{int}}$ cells. Similarly, from FL, we sorted CLP as $\text{Lin}^- \text{CD135}^+ \text{CD127}^+ \text{CD27}^+ \text{CD115}^- \text{SiglecH}^- \text{Ly6D}^-$ to ensure we excluded any CD115^+ myeloid or $\text{SiglecH}^+ \text{Ly6D}^+$ pDC/tDC progenitors. We did not include CD117 in our gating strategy as *Kit* expression levels were not sufficiently discriminatory between the tested progenitors in our scRNA-seq analysis (**Reviewer Fig. 4**). Lui et al (2022) identified MDP/CDP as $\text{Lin}^- \text{SiglecH}^- \text{CD135}^+ \text{CD117}^{\text{hi/int}} \text{CD115}^+ \text{MHCII}^- \text{CD11c}^-$ cells. Similarly, from FL, we sorted MDP/CDP as $\text{Lin}^- \text{SiglecH}^- \text{CD135}^+ \text{CD115}^+ \text{MHCII}^{-/\text{lo}} \text{CD11c}^{-/\text{lo}} \text{CD27}^- \text{SiglecH}^- \text{Ly6D}^-$ to ensure we excluded any CD27^+ lymphoid or $\text{SiglecH}^+ \text{Ly6D}^+$ pDC progenitors. Given the different gating strategies for tDC/pre-pDCs across Sulczewski 2023, Rodrigues 2023, and Feng 2022, we first wanted to develop a gating strategy for isolation of the pre-pDC cluster in our dataset. Projection of the BM tDC gene signature onto our progenitor UMAP suggested that these cells could be identified in the FL as $\text{Lin}^- \text{CD135}^+ \text{CD115}^- \text{CD27}^- \text{SiglecH}^+$. While Sulczewski 2023, Rodrigues 2023, and Feng 2022 have suggested that these tDCs can be further subclustered based on CX3CR1 and CD11c expression, dissection of these subsets further was unnecessary given that bulk transfer of tDCs did not give rise to TCs. Of note, aside from the adoptive transfer of MDP, the study by Rodrigues et al. characterized splenic populations rather than bone marrow progenitors. We note that their definition of splenic tDCs ($\text{FLT3}^+ \text{CSF1R}^- \text{IL7R}^+ \text{SiglecH}^+ \text{Ly6D}^+$) overlaps with our gating with the exception that we included $\text{SigH}^+ \text{Ly6D}^-$ cells as our scRNA-seq demonstrated that most of the cells in the pre-pDC/tDC clusters did not express Ly6D (**Reviewer Fig. 4**). This is also in line with Rodrigues 2023 which showed that upregulation of Ly6D accompanies progressive pDC lineage commitment.

Our inclusion of CD27 for stringent CLP isolation is in line with published studies demonstrating expression of CD27 by CLP (Serwold et al. Blood 2011). In addition, we exclude Siglec-H and Ly6D^+ cells because $\text{FLT3}^+ \text{CSF1R}^- \text{IL7R}^+ \text{CD27}^+$ cells are also present in the pre-pDC/tDC cluster. Thus, our gating strategy results in a purer population of CLPs.

With regard to the ability of pre-pDCs/tDCs to give rise to cDC1s, while we agree that tDC to cDC2 is likely the physiological pathway of differentiation, our adoptive transfer findings are not inconsistent with the literature as the study by Sulczewski et al. (Fig. 3m) similarly showed that

immature pDC and pro-pDCs (equivalent to our pre-pDC/tDC cluster) gave rise to cDC1s and CD11b⁻ DCs 4 days post adoptive transfer. Of note, the main purpose of the pre-pDC/tDC transfers was to determine if tDCs could give rise to TCs. We have used orthogonal approaches to address this question, including Zeb2 mutant mice in the first submission, and in the revised manuscript we used TCF4 null mice which lack tDCs.

With regards to B cells, we apologize for the confusion - as noted on the flow plot legend (**Fig. 3b**), CD19⁺ cells were excluded from the analysis because we are focusing on visualization of abundance of rarer cell types. We have been clearer about this in our explanation of Lin⁻ cells in the figure legends.

As suggested, we have moved gating strategies for the TLP and TCP to main figures (**new Fig. 1p, 2c,d**) and have made it clearer in the text that the selection of markers for progenitor sorts were based on our scRNA-seq analyses with accompanying figures.

Progenitor differentiation assays (Figure 2): The conclusion that CLPs give rise to TCs should include single-cell assays. If feasible, the authors should single-cell sort CLPs and test their ability to generate TCs in vitro. As noted above, the authors should evaluate the potential of TCp to give rise to TC I-IV. These experiments would help clarify whether TC I-IV derive from a common precursor, and whether they emerge in a defined sequence or independently.

We spent a considerable amount of time trying to establish culture conditions that support CLP→TC differentiation and were able to generate TCs from CLP using conditions typically used to promote differentiation of lymphoid cells such as ILC or LT_i cells (IL7, stem cell factor (SCF), FLT3) (**Reviewer Fig. 5**). Of note, we could not generate TCs from MDP/CDP or CLP using typical culture conditions for DC differentiation (FLT3/GM-CSF). This is in line with published findings from PMID 40185101, examining in vitro differentiation of progenitors and their ability to generate ROR γ ⁺ APCs. However, even under optimized conditions, the number of cells generated is very low, and we were not able to robustly detect TCs once the input dropped below 100 cells. This is in line with reported findings from other studies such as PMID 29925996, which noted that lymphoid cell progenitors have diminished survival or proliferation capacity in vitro, relative to CDP. Thus, when we attempted to scale down to single CLP differentiation assays, we could only recover one or two cells and could not make any robust conclusions as to the identity of these cells, nor could we assess the ability of a single cell to generate TC I-IV. However, we have addressed the potential of the LN TCP, which we believe is the immediate progenitor with TC restricted potential, using bulk cultures, confirming that this progenitor can give rise to CD11c⁺EpCAM⁺ TCs, representing II-IV (**Fig. 2e**), as noted above. We also turned to genetic lineage tracing approaches to assess precursor-progeny relationships for TLP and TCP using a new Siglec-F lineage tracing mouse model in which cells that are descended from the ROR γ ⁺SiglecF⁺ TLP are labeled with tdTomato⁺, as outlined above. This new data strongly suggests that TC I-IV have a common precursor. Overall, we believe that our bulk CLP cultures, alongside adoptive transfer experiments of CLP, and exclusion of precursor-progeny relationships with other IL7R⁺ progenitors such as tDC, strongly support a model where TCs arise from the CLP.

While we cannot determine whether TC subsets arise in parallel or in sequence using our lymph node cultures, due to the limited number of cells generated and inability to perform sequential timepoint analysis, our new fate mapping assays with *Prdm16*^{creERT2} (**new Fig. 13a**) and

previous results with *Aire^{creERT2}* alleles (**Fig. 13e**), revealed that *Aire*⁺ TCs cannot give rise to *Aire*⁻ TCs, thereby excluding e.g. a pathway from TC I or TC III to either TC II or TC IV, and conversely, PRDM16⁺ TC II-IV cannot give rise to TC I, suggesting limited plasticity between TC subsets.

- Fate-mapping models and lymphoid origin: The interpretation that TCs are lymphoid-derived, based on fate-mapping with IL7R, requires more caution. Prior work (e.g., Reizis and colleagues) has shown that transient expression of lymphoid markers can be induced by certain transcription factors (e.g., E2 family) without establishing true lymphoid lineage commitment. The co-expression of CX3CR1 and IL7R in the same cells—as reported here—suggests that the cells maybe myeloid-derived and expressing a lymphoid-associated marker, similar to pDCs and tDCs. Additionally, the recent work by Colonna's group using similar fate-mapping tools (hCD2-driven) arrives at comparable conclusions about the origin of TCs (also called RORγt+ DCs), decreasing the novelty of the findings. Unfortunately, this manuscript does not move the field forward in clarifying whether these cells are truly lymphoid-derived or acquire lymphoid-like features during development. Finally, evaluating TC I–IV separately in all the assays, including the IL7R fate-mapping model would provide valuable insight.

We agree that IL7R does not denote derivation from the CLP as a proportion of tDC/pre-pDCs also express IL7R, as noted by the reviewer. Our scRNA-seq analysis of FL progenitors identified IL7R expression across the CLP, CLP/lymphoid primed progenitors and a cluster of SiglecH⁺ progenitors encompassing pre-pDCs and potentially tDCs (referred to as pre-pDC/tDC in the manuscript) (**new Extended Data Fig. 7a**). In our revised manuscript we have been explicit about the potential interpretation of IL7R fate-mapping when we first introduce the mouse model, and we have been clearer in the text that the IL7R lineage tracing only infers descendancy from an IL7R progenitor (not necessarily CLP), up until the point that we have addressed the TC potential of CLP using adoptive transfers. In the revised manuscript, we have definitively addressed the precursor-progeny relationship of pre-pDC/tDC and TCP using TCF4-deficient (*Tcf4^{STOP/STOP}*) mice which lack tDCs and pre-pDCs. Analysis of *Tcf4*-deficient FL or competitive *Tcf4^{STOP/STOP}*/wildtype FL chimeras revealed that loss of TCF4 does not impair TLP, TCP or TC I-IV differentiation (**new Fig. 3d–f** and **Extended Data Fig. 7e**). Unexpectedly, these experiments revealed a significant increase in FL TLP numbers as well as peripheral TCs, suggesting that TCF4 may determine the balance between pDC vs TLP cell fate in early Siglec-H⁺ progenitors. Additional experiments were performed to address this including Siglec-H fate mapping which demonstrated labeling within TCs and LTi cells but not other ILC subsets (**new Fig. 3h**). Thus, having excluded pre-pDC/tDCs as a source of TCs, the remaining candidates are CLP or other IL7R lymphoid primed progenitors. While we cannot exclude the possibility that our CLP transfers contain a rare, as yet uncharacterized myeloid progenitor with lymphoid features, we believe that the balance of evidence, in particular our adoptive transfer of CLP which generates TCs, and conversely a lack of TCs with MDP/CDP transfers, supports our model of lymphoid-derived TCs.

In addition, using new mouse models for lineage tracing the RORγt⁺ FL TLP cluster, we show that TCs and LTi cells are descended from a common RORγt⁺SiglecF⁺ progenitor. Since previous studies have already established that LTi cells are derived from the CLP (PMID 21909092, 20929731, 26832410, 11359812), we believe that by extension, TCs must be lymphoid derived cells.

While other studies have suggested lymphoid origin based on hCD2 lineage tracing, as noted by the reviewer, this may reflect expression of hCD2 by cells expressing lymphoid associated genes, rather than true lymphoid origin. We believe that our additional experiments have definitively established that TCs are lymphoid derived. Moreover, we believe additional novelty arises from identification of the TC progenitor and the transcription factors required for TCP differentiation, as well as establishing the differentiation pathway for LT_i cells. In addition, we use in vivo adoptive transfer assay to show that the TCP does not arise via a CLP-tDC-TC or CLP-MDP/CDP-TC pathway, which had not been addressed by the study by Colonna's group and is necessary for determining whether TCs share a developmental pathway with cDCs or have a distinct developmental pathway. In contrast, we show that TCs arise through a lymphoid restricted pathway of development.

In the revised manuscript, we have included analysis of the individual TC I-IV subsets in the IL7R fate mapping model, demonstrating that all TC subsets are fate mapped (**new Extended Data Fig. 2b**).

- Transcription factor conclusions (Figures 4–5): The authors propose that tissue-derived environmental cues regulate PRDM16 expression and TC subset fate decisions. However, this idea is speculative and not directly tested in the manuscript. If the authors wish to make this point, mechanistic evidence would be needed. Otherwise, the conclusion should be presented more cautiously as a hypothesis rather than a demonstrated result (perhaps reserved for the discussion section).

We agree, we have revised the language and only speculate on environmental cues that regulate the window for TC differentiation in the discussion, without specific reference to PRDM16. However, we note that we were only able to successfully culture TCP when we moved to culture systems with ex vivo lymph node slices, indicating that lymph node cues are required to support TC differentiation.

- Relationship between TC subsets (Figure 5): The developmental model presented in Figure 5j raises some inconsistencies. Earlier, the manuscript suggested that TC I and III are stable based on Aire fate mapping; later, it is proposed that TC II–IV arise from a shared progenitor. It is unclear how these conclusions coexist, and no functional data are provided to test this model. Experiments involving sorting and differentiation of TC I–IV (in vitro if adoptive transfer is not feasible) would help clarify whether these subsets are developmentally linked or represent distinct lineages with overlapping transcriptional profiles (especially in the case of TC I). Additionally, the relationship between TCs and ROR γ ⁺ DCs, as described by other groups and in previous publications by these authors, remains ambiguous. In some sections, TC II–IV are equated with ROR γ ⁺ DCs; in others (and previous publications by the authors), only TC IV is linked. A clearer reconciliation of terminology, gating strategies, and nomenclature between studies would increase the manuscript's clarity and broader impact.

We apologize for the confusion. We do not believe that a shared progenitor and stable subsets are incompatible.

Our model is one where the TCP differentiates into 4 subsets (TC, I, II, III, IV) and each subset represents a terminally differentiated cell that cannot interconvert between subsets. The Aire fate mapping data is not inconsistent with this model as it shows that TC I and TC III cannot become TC II-IV. In addition, since the TCP does not express Aire (**Reviewer Fig. 6**), it does not conflict with our model where TCP gives rise to both Aire negative and Aire positive subsets.

As noted above we cannot sort individual subsets as the terminally differentiated cells do not survive in culture *ex vivo*.

As noted above, we have resolved disparate nomenclature across published studies for a review published in *Nature Immunology* (PMID 41107526) and have included this figure as **Reviewer Fig. 1**.

- **Inconsistent terminology and classification (all manuscript):** The assertion that only CDP-derived cells should be labeled as dendritic cells is not consistent with field consensus. TCs express FLT3L dependence and are labeled in CX3CR1 models; importantly, they are professional antigen presenting cells capable of inducing *de novo* regulatory T cells—features typically associated with the dendritic lineage cells. Other groups, including Colonna and Littman, have used these arguments to classify these cells as dendritic cells. Morphological characteristics are insufficient as an argument, especially as pDCs can acquire full dendritic morphology and function upon activation (Reizis, *Nat Rev Immunol* 2023). Tracing with IL7R-Cre mice also has its problems since (as explained above) this is just a lymphoid feature that may be regulated by well-known E2 transcription factors. Overall, the manuscript does not resolve ongoing ambiguity regarding TC identity and would benefit from more nuanced and cautious language around DC lineage definitions.

In 2014, guidelines for DC nomenclature were proposed and these became widely accepted in the field (Guilliams *et al. Nature Reviews Immunology*). These guidelines state that dendritic cells are derived from the CDP. A more recent commentary in *Nature Reviews Immunology* (2022) from Florent Ginhoux, Miriam Merad and Martin Guilliams further confirmed that current consensus defines dendritic cells according to ontogenetic relationships. To our knowledge, there has been no revised definition of dendritic cells based on transcriptional programs, FLT3L dependence or CX3CR1 fate mapping. Of note, FLT3L dependence cannot be used to ascribe DC identity because the CLP expresses FLT3, and is impacted by FLT3L deficiency (PMID 30596405 and 12387740), along with other established lymphoid lineages such as ILCs (PMID 26851220). The authors that described the FLT3L dependence of TCs (PMID 39993193) did not claim myeloid lineage based on this feature. We note that T cells and ILCs were CX3CR1 fate mapped to a similar degree as TCs (**Extended Data Fig. 2c**); thus, CX3CR1 is not exclusively expressed by myeloid progenitors. We agree that morphology can change and thus we agree with the guidelines that ontogeny is the best way to define DCs.

While we note that other groups have tried to define TCs as DCs based on expression of DC-associated genes, the published peer review comments for PMID 39091750, did not agree with the authors' DC definition based on these criteria, and instead support an ontogenetic definition of DCs, as below

R1: The current work does not show direct evidence for the APC belonging to the DC lineage at all. Certainly they share some features of conventional DC and the new, epigenetic studies are interesting. However many of the phenotypic properties are not specific to DC (eg CD11c, MHCII, CSF1R) and their expression by the new APC was heterogeneous. To identify the APC as bona fide DC, it will be necessary to show eg that they are derived from a pre-cDC

R1: as I noted in the previous reviews, their identification as DC remains much less definitive than the manuscript makes out; it is based mostly on assumptions from overlapping/inconsistent

markers. Too many phrases such as "by process of elimination", "closely resembling" etc are not sufficiently precise for unravelling such a complicated issue

R2: Furthermore, I agree with Reviewer 1 that the terminology "tolerogenic DC" may not be warranted based on current evidence, and more generally there are a number of speculative statements and interpretations I feel should be avoided, including several new statements speculating about lineage relationships

Similarly, although the study from the Colonna lab renamed TC I-IV as ROR γ t dendritic cells I-IV, despite showing that these cells are hCD2-lineage traced and derived from lymphoid progenitors with ex vivo culture assays, indicating lymphoid origin, no justification was provided for the revised DC nomenclature.

With regard to the FLT3L dependency, this required further investigation since our own data with FLT3 activation (**Extended Data Fig. 8c**) and the Colonna lab's study with FLT3L treatment showed only modest effects on TC numbers and FLT3L did not appear to efficiently support TC differentiation in vitro (40185101). A separate study by the Gardner lab showed no impact of FLT3L or FLT3L blockade on TC abundance (PMID 40298935). To examine the role of FLT3 more closely we repeated experiments with the FLT3-ITD mice, analyzing a cohort of mice that received tamoxifen in adulthood, and extended our previous analysis of neonatal FLT3-ITD activation to include analysis of TC abundance in pLN and spleen. This analysis revealed that despite dramatic increases in cDC and pDC numbers across all lymphoid tissues examined, changes in TCs were modest and variable. While we observed modest (2-4 fold increases) increases in TC numbers in the mLN and pLN, we did not observe changes in the spleen (**new Extended Data Fig. 8c**). This suggests that the role of FLT3 in regulation of TCs is i) minor and potentially indirect and ii) distinct from FLT3 regulation of cDCs,

- Statistics, transparency, and data analysis (all manuscript): Statistical rigor should be improved in several parts of the manuscript. For example, Figure 4I lacks statistical comparisons, and similar gaps appear elsewhere. Gating strategies for all progenitors—including TCp—should be clearly shown in main figures, as they are foundational to the experiments and necessary for reproducibility and comparison. For the transcriptomic analysis, the authors should explain how they aligned populations and which gene signatures were used to assign identities.

We apologize for this omission and have added statistical comparisons for all of the graphs. We also provide further details in the methods describing alignment of particular populations and, as noted above, we have moved the gating strategies for the TC progenitor into the main figures. For all other progenitor gatings, we are constrained for space in the main figures but have included these in supplementary figures.

G. References

Key recent studies (Fu et al., Rodrigues et al., Cabric et al., Narasimhan et al., Feng et al., Sulczewski, Rodriguez, etc.) are not adequately cited. Their omission gives a misleading impression of novelty and obscures important field-wide debates. The manuscript would benefit greatly from a more honest engagement with prior work—especially where conclusions are divergent. This includes conclusions regarding nomenclature: while it is understandable that groups may adopt their own naming conventions, the lack of reconciliation now makes it

increasingly difficult for non-specialists to follow the literature and understand how various reported cell types relate.

We apologize for the confusion and oversight – at the time of submission, Cabric *et al.* had not been published, and Fu *et al.* had just been published online as an advance publication. Nevertheless, we had included Fu *et al.*, Narasimhan *et al.*, Sulczewski *et al.* and have added Feng *et al.* and all of the recently published studies in the revised manuscript. We now discuss nomenclature in the introduction and Discussion. As noted above, we performed an integrated analysis of all published datasets and resolved disparate nomenclature for a review that is now online at *Nature Immunology* (PMID 41107526). We have also referenced the review in the revised manuscript.

Similarly, the authors claimed that only cDCs are descended from Cx3cr1+ myeloid progenitors, but pDCs are not. However, recent data from the group of Boris Reizis (Feng *et al.*, *Immunity* 2022) clearly shows that pDCs are also derived from Cx3cr1+ progenitors using in vivo barcoding. Although pDCs can be obtained in vitro from lymphoid progenitors (e.g., Tussiwand and collaborators), their in vivo contribution appears minimal at steady state (Feng *et al.*, *Immunity* 2022). Similarly, there is currently no evidence that transitional DCs (tDCs) are lymphoid derived (Sulczewski, 2023; Rodriguez 2023). On the contrary, these appear to follow a Cx3cr1-dependent path. These distinctions should be acknowledged to prevent confusion.

We apologize for the confusion. We did not mean to infer that pDCs are not derived from CX3CR1 progenitors - we did not include pDCs in the CXCR1 fate mapping graph because, as demonstrated by Feng *et al.*, the pre-pDC progenitor expresses CX3CR1, therefore CX3CR1 fate mapping is not useful for assessing descendancy from an upstream CX3CR1+ myeloid progenitor, which was the point we were trying to address. With regards to pDC ontogeny, we note that this has not been unequivocally resolved with some studies suggesting lymphoid origin (PMID 31213723, 29925996, 38821051), and others myeloid origin (PMID 38227649). Our study was not aimed at addressing pDC ontogeny as it is beyond the scope of the manuscript to revisit this debate. In the revised manuscript, we have provided more context to the current view of pDC ontogeny, acknowledging two potential pathways for development, with reference to the above studies.

Finally, the authors claimed that only CDP-derived cells should be called “dendritic cells” — based on only a few authors’ views and a single citation—this is not universally accepted. This definition would exclude well-established subsets such as monocyte-derived dendritic cells, plasmacytoid/transitional dendritic cells, and DC3s, and risks creating unnecessary confusion for readers outside the immediate field.

We respectfully disagree and note that the manuscript we cited was a review article that proposed a unified nomenclature for dendritic cells, monocytes and macrophages based on ontogeny, which has become the consensus in the field. The review was co-authored by leaders in dendritic cell biology including Martin Guillems, Florent Ginhoux, Elodie Segura, Roxane Tussiwand, among others, and has been cited over 2000 times. Since some studies have provided evidence of lymphoid ontogeny for pDCs, there have been many debates, both in the literature and at DC conferences, surrounding the DC nomenclature for pDCs, and the prevailing view is that dendritic cells are defined by ontogeny. We note that some groups have called for changes in pDC nomenclature, including renaming them plasmacytoid innate lymphoid cells. Clearly, this reflects ongoing controversy surrounding their ontogeny (myeloid vs

lymphoid) and we have acknowledged the different viewpoints in the revised manuscript. While we have revised the language to provide context to the pDC ontogeny studies, we do not feel that we can change the ontological definition of a dendritic cell. We note, that when DC3s were first named, they were thought to be a subset of cDC2s, and it was only later that they were shown to be derived from monocytic precursors and thus represent a subset of monocytes that transcriptionally resemble cDCs. Although tDCs can give rise to cDC2s, in particular cDC2As, this does not appear to be the dominant pathway for cDC2A differentiation given that >95% of cDC2A are lineage traced with Clec9a-cre (Nguyen et al. *Nat Immunol* in press DOI: <https://doi.org/10.21203/rs.3.rs-4784425/v2>). In the revised manuscript we have toned down the discussion related to the morphology of TCs and emphasized only that TCs have a developmental pathway that is distinct from cDCs. Moreover, we now show that TCs and LTi cells share an immediate common progenitor, and show that TCs and LTi cells are CLP-derived and do not arise via an intermediate tDC or CDP/MDP progenitor. It is beyond the scope of this manuscript to settle the debate over DC definitions, but note that the field can continue to evaluate the “dendritic cell” identity of TCs, based on the evidence provided.

H. Clarity and context

The abstract and discussion overstate the novelty and strength of the data. In a rapidly evolving field, it is crucial to clarify how the present study fits with existing work. The inconsistent naming of similar cell types by different groups is already confusing; this manuscript adds to that confusion by failing to reconcile definitions or establish unifying criteria for identity. For instance, Dan Littman and his group recently published a manuscript on the role of PRDM16 in ROR γ t⁺ dendritic cells (*Nature* 2025), while the current manuscript refers to apparently similar cells as TCs. For trainees and non-specialists, these distinctions are difficult to parse, and it is our responsibility as a community to ensure that scientific findings are not obscured by inconsistent terminology. The authors are encouraged to clarify how their cell population relates to others described in the literature, and to reconcile naming, gating strategies, and definitions across groups.

We apologize – we did not intend to create further confusion, rather the opposite. We feel that all future studies published after the full description of the TC lineage in 2022 should have provided reference to the original datasets and avoided additional names for the cells, though we acknowledge this is a difficulty with the generation of so many datasets. We completely agree that this is causing considerable confusion, particularly for those who do not work on the cells. We therefore dedicated a considerable amount of the paper to validating our gating strategies, including a new SS3 index sorting analysis (**new Fig 2a–d**) and the use of new Aire-reporter mice to resolve the confusion over previously defined ROR γ t⁺ extra thymic Aire expressing cells (**Extended Data Fig. 13c,d**). We use the same SS3-validated flow gating strategy for TC subset identification throughout the study, allowing others to reproduce our findings. This gating strategy can now be adopted by the field to allow cross comparison in future studies. In addition, as noted above, we have performed an integrated scRNA-seq analysis that incorporates the original datasets from 2022 and recently published studies, which is part of a review recently published in *Nature Immunology* and now referenced in the revised manuscript (PMID 41107526).

Moreover, the data—including IL7R labeling—do not rule out that TCs are related to dendritic cells. These cells require FLT3L for development, are fully labeled by CX3CR1 fate-tracing, and lack features of ILC3s, as shown by others (including Littman and Colonna) and the authors.

Morphology-based arguments are also insufficient; for example, pDCs can adopt dendritic morphology and activate naïve T cells upon stimulation. A particularly critical concern is the interpretation of CLP adoptive transfers. The gating strategies used are not fully aligned with widely accepted protocols, and the recovery of downstream populations diverges from prior reports—suggesting possible contamination or misidentification of precursors. If the central message of the manuscript is the lymphoid origin of TCs, it is essential that these experiments are performed with robust methodology, ideally including single-cell assays and more rigorous lineage resolution.

We believe that our data in the revised manuscript provide strong evidence that TCs have a distinct developmental pathway from myeloid derived dendritic cells as well as demonstrating that TCs are not descended from tDCs, the non-canonical pathway for DC differentiation. Thus, it is unclear why TCs would be considered dendritic cells as opposed to a new APC lineage, as proposed in our original study identifying and characterizing TCs. This study included lineage tracing approaches addressing descendancy from the $ROR\alpha^+$ ILCP or $CLEC9A^+$ CDP/preDC (PMID 36070798). As noted above, their dependence on FLT3L is nuanced and inconsistent across published datasets and thus requires further investigation, which we performed in the revised manuscript, outlined above. In addition, we do not believe that FLT3L dependence is evidence of dendritic cell identity given that CLP express FLT3, are dependent on FLT3 signaling, and other lymphoid lineages including ILCs and LT_i cells are dependent on FLT3L.

In addition, TCs, unlike cDCs and pDCs, are not fully labeled by CX3CR1 fate-mapping, rather they are labeled to a similar degree as T cells and ILCs.

With regards to the gating strategy for CLP, we note that our gating strategy is very similar to other gating strategies, if anything it is more stringent due to the inclusion of CD27 and therefore less likely to contain contaminating myeloid/DC progenitors. As noted above, we do not feel that our findings diverge from published studies as others have shown that tDCs can give rise to cDC1s following adoptive transfer. While this may represent an impure population of tDCs, we addressed a precursor-progeny relationship between tDCs and TCs with additional genetic mouse models. We also note that no other studies have established flow cytometry gatings for FL progenitors. Our conclusion that TCs are not related to dendritic cells does not solely rely on CLP transfers, as MDP/CDP transfers were unable to give rise to TCs. We have also excluded a potential contribution from tDCs using TCF4-null mice and these experiments are included in the revised manuscript, as outlined above. As suggested by the reviewer we attempted to perform single cell CLP cultures. However, although we were able to optimize conditions for generation of TCs from bulk CLP (**Reviewer Fig. 5**), we recovered very few cells, in line with their relative abundance in vivo, and reducing the input number of cells resulted in too few cells for robust analysis of precursor-progeny relationships. This is in line with a study by Rodrigues et al. PMID 29925996 which demonstrated that lymphoid progenitors have reduced proliferative capacity ex vivo or survival relative to myeloid CDP. We therefore turned to genetic mouse models to lineage trace putative progenitors, including *Siglech^{cre}R26^{Isl-tdTomato}* for the earliest TC/LT_i primed progenitors and *Siglect^{cre}R26^{Isl-tdTomato}* for the common TLP. These data established that TCs and LT_i cells are descended from a common progenitor (the TLP), further suggesting TCs are lymphoid-derived. Despite exhaustive investigation, we have no evidence to support a myeloid pathway for TC development based on lineage tracing, genetic mouse models and adoptive transfer experiments.

Finally, one of the most compelling aspects of TCs for a broad Nature audience is their ability to

promote immune tolerance during early life, as elegantly demonstrated in recent work (Fu et al., Nature 2025; Rodrigues et al., Science 2025; Cabric et al., Science 2025). In contrast, this manuscript focuses narrowly on developmental origins and does not include new functional data—limiting its appeal to a more specialized readership.

We feel that it would be beyond the scope of the manuscript to address both ontogeny and function of individual subsets. We believe that understanding the ontogeny of TCs is essential to resolve inconsistencies in the field and to enable future research into their function, as well as their potential for therapeutic development to target food allergy, autoimmunity and transplantation. Moreover, as the reviewer noted, a compelling aspect of TCs is their developmental wave in early life which we believe underlies the window of opportunity for immune tolerance. Understanding the ontogeny of TCs and the cues that drive their regulation, is key to unraveling this phenomenon.

Referee #2 (Remarks to the Author):

Review of Nature manuscript 2025-04-08716A "Ontogeny and transcriptional regulation of Thetis cells".

The authors studied the ontogeny of the recently identified so-called Thetis cells (TC) (I cannot find an explanation in the literature but perhaps it refers to a cell 'between mTEC and cDC', i. e. one cell type 'married' to the another? Note added in proof; I found the explanation in the discussion). The authors used fate mapping, cell transplantation, RNA seq-derived trajectories, and conditional gene targeting with the aim to identify TC progenitor candidate cells. Overall, this is an interested topic, the experiments make (mostly) sense and are developed in good order. The data suggest that new insights in the ontogeny of TC have been gained.

We thank the reviewer for their positive and constructive comments.

Questions and concerns:

Fig. 1 legend states 'Thetis cells arise from a ROR γ t⁺ progenitor during early life' when, in fact, these cells also arise from bone marrow (Fig. 1a, b). I don't see a clear and simple comparison of the TC yield (in absolute cell numbers) after bone marrow or fetal liver transplantation, or the number of cells that are actually fate mapped. Percentages in the absence of absolute cell numbers are inconclusive. Do you find normally sized TC compartments after transplantation, or else it may be difficult to argue that there is only an hematopoietic origin?

We apologize for the confusion, this legend was intended to highlight the predominant early-life pathway for TC development, but we agree that it may be interpreted as a complete absence of TC progenitors in adult BM. We have revised the legend. In addition, in the revised manuscript, we have included cell counts for TCs and other innate immune cell populations (**Fig. 1b** and **Extended Data Fig. 1f**) in FL vs BM chimeras. This analysis confirmed that the absolute number of TCs is higher in chimeras reconstituted with FL vs BM, and this increase in cell number is specific to TCs as well as LTi cells, which are also known to develop predominantly from FL progenitors. The size of the TC compartment is only modestly reduced after transplantation, potentially reflecting the change in the size of LNs in irradiated recipients. However, we believe that our new chimeras that address host and donor TCs in the same mice (**new Extended Fig. 1a–e**) establish that all TCs have a hematopoietic origin.

Fig.1b: What does the background (Venus channel w/o Venus) look like?

When we gate on host CD45.1 Lin-MHCII⁺ cells, there is no signal in the Venus channel. However, the previous chimera analyses with RORc-Venus mice have been replaced with an analysis of CD45.1/CD45.2 chimeras, with intracellular staining of RORγt in both host and donor TCs, as suggested by the reviewer in the comment below.

Fig. 1f y axis: what is the unit for cell number'? (this is a problem in several panels, please check them all).

We apologize for the confusion – we have added the unit to all graphs.

Ext data Fig. 1e, f (CD45.1 donors into CD45.2 hosts): can you show, akin to panel f, positive reconstitution CD45.1 TC in the same experiment? Without such a control the data are panel f are incomplete.

As noted above, we have replaced these experiments with CD45.2 C57Bl/6 donors into CD45.1 hosts and analyzed host and donor derived TCs by intracellular staining (**new Extended Data Fig. 1a,b,d,e**). We performed parallel analyses of CD45.1 into CD45.2 mice and were able to reconstitute TCs (**new Extended Data Fig. 1c**)

Lower half of page 4: 'The classical model of hematopoiesis...."recent studies have shown that pDCs, traditionally thought to represent myeloid cells, arise from the common lymphoid progenitor (CLP)^{16,17}, the earliest progenitor that expresses the IL-7 receptor (IL-7R)¹'. This is not correct. In the original Il7rcre paper by Schlenner et al (Immunity 2010), it was shown that pDC were fully labelled in Il7rCreRFP mice. However, because pDC express the IL7r, it becomes impossible to tell whether the label is derived from a progenitor stage or is acquired in the lineage.

Because this paper relies heavily on the use of Il7rcre mice to detect a lymphoid origin and to make conditional knockouts, it would be fair to cite Schlenner et al in this section for lymphoid fate tracking. On another note, Il7rcre mice were provided to labs in NY through the Rodewald lab via Andrew McKenzie, and not provided by Vinod Balachandran (as suggested in the Acknowledgements).

We wholeheartedly agree and apologize for the confusion. In the revised manuscript, we have provided greater clarity on the interpretation of IL-7R fate mapping, citing Schlenner *et al.*, and have been explicit about the potential IL7R⁺ progenitors and progeny that are fate mapped. As noted in our response to Reviewer 1, our scRNA-seq analysis of FL progenitors identified two IL7R expressing cell types – CLP/lymphoid primed progenitors and pre-pDCs/tDCs (likely pre-pDCs as discussed above). Our conclusion that TCs are lymphoid derived is based on our CLP transfers, which were the only progenitors to give rise to TCs, as well as new genetic epistasis experiments with TCF4-null mice that showed that loss of pre-pDCs/tDCs—the other candidate IL7R⁺ progenitors—does not impact TC differentiation. We also note that we now show that LTi cells and TCs share a developmental pathway using a new genetic mouse model to label TLP progeny. Since previous studies have already established that LTi cells are lymphoid derived cells (PMID 21909092, 20929731, 26832410, 11359812) we do not believe the lymphoid pathway for LTi/TC differentiation is contentious. We have amended the text and revised our interpretation of the IL-7R fate mapping, providing a clear rationale for additional studies

addressing the TC potential of candidate IL-7R⁺ progenitors identified in **Fig. 1h** and **new Extended Data Fig. 7a**.

We sincerely apologize for the misattribution of the mouse strain. We have separately written to explain how this arose and have credited the mice to the Rodewald lab.

Fig. 2d. Are only Lin⁻ IL7R fate-mapped and Lin⁻RORgt(Venus)⁺ cells shown? If yes, these data in disagreement with Schlenner et al who showed in their Fig. 2c high labeling in CLP but very low if any labeling in HSC and myeloid progenitors. In light of these data, the statement 'Unsupervised clustering and differential gene expression analysis revealed the full repertoire of hematopoietic progenitors spanning hematopoietic stem cells (HSCs).....highlighting the ability of IL7R⁺ progenitors to give rise to diverse immune cell types during early life' seems incorrect and indeed misleading. It is clear that IL7R Cre fate mapping labels cells downstream and not upstream from lymphoid progenitors (or else the data in this paper would be entirely inconclusive regarding a lymphoid origin of TC). This concern also applies to Fig. 2e. Overall, while the data in Fig. 2d sheds doubt on the nature of cells that were labelled prior to RNA seq, the data may still include TC progenitors. The subsequent transcriptional characterization of putative TC progenitors relies on landscapes and trajectories which, in the absence of direct fate mapping using newly identified specific marker loci (which would be more convincing) seem acceptable.

The cells in Fig. 2d were sorted as Lin⁻tdTomato⁺ (IL-7R fate-mapped) and Lin⁻RORgt(Venus)⁺ cells. We analyzed *Il7r* transcript expression across all clusters and noted detectable *Il7r* expression in the mixed LMPP/CLP cluster, as well as CLP and pre-pDC clusters. We have included this analysis in the revised manuscript (**new Extended Data Fig. 3d and 7a**). We did not observe IL7R expression in the previously annotated HSC cluster, prompting us to explore the identity of this cluster in more detail. In addition to expression of canonical genes, we had annotated these clusters according to their expression of gene modules defined in previously published scRNA-seq datasets of bone marrow hematopoiesis, including PMID 32203468. This revealed enrichment of an "HSC-3" cluster signature in our previously annotated HSC cells (**Reviewer Fig. 7**). However, on closer inspection we realized that this gene signature was also enriched in our LMPP/MDP cluster suggesting that the HSC-3 cluster may in fact represent a myeloid progenitor. We therefore turned to an alternative scRNA-seq analysis of bone marrow progenitors (PMID 38821051) which included HSCs, LMPP and MDPs. Our analysis confirmed a lack of enrichment of HSC signature genes, and instead demonstrated enrichment of the LMPP signature. The CLP signature defined in both PMID 32203468 and 38821051 was enriched in our CLP cluster as well as a subset of the LMPP/CLP cluster (**Reviewer Fig. 7**). We have therefore updated the HSC annotation to LMPP, reflecting the earliest lymphoid and myeloid primed progenitors within our dataset (**Fig. 1h**). All of the other cluster annotations remain the same. We apologize for the confusion and thank the reviewer for prompting us to review this more critically.

We have also confirmed IL7R lineage tracing by flow analysis of tdTomato expression in the indicated cell types shown in Fig. 2d and 2e (neutrophils, monocytes, macrophages, eosinophils, dendritic cells). This confirmed that FL IL7R progenitors can give rise to a broad array of immune cells (**new Extended Data. Fig 3e**).

The knockout data on Zeb2D1+2+3 and Tcf7 seem clear. In contrast, why should TC be

independent from Tox if the contribution of *Il7rcrToxfl/fl* in mixed BM chimeras is only about 1%?

We apologize for the confusion. The axis represents the ratio of wildtype vs IL7R deficient cells for each cell type shown. For TCs, the ratio is ~1 indicating that TCs can develop in the absence of Tox, in contrast to e.g. T cells or ILCs. In the revised manuscript, we have included representative flow plots showing the relative chimerism (new **Extended Data Fig. 9h**).

The data identifying PU.1 as a key regulator of TCp differentiation into TC I-IV is really nice.

The sentence 'A recent study demonstrated a role for PRDM16 in non-ILC3 RORγt+ APC differentiation; however, did not examine individual TC subsets' lacks a reference.

We apologize for the oversight; we have added this reference (Fu et al. Nature 2025).

The remained of the paper (Figures 4 and 5) are convincing in my view.

Overall, this is an interesting paper. Some aspects are not fully conclusive but there is a wealth of new information and plenty of leads that can be followed up in the future.

Hans-Reimer Rodewald

Referee #3 (Remarks to the Author):

I co-reviewed this manuscript with one of the reviewers who provided the listed reports.

Referee #4 (Remarks to the Author):

Iza et al. aim to identify and characterize the progenitor and developmental trajectory of Thetis cells (TCs), a newly described population of RORγt⁺ antigen-presenting cells (APC) (PMID: 36070798). In their earlier work, the authors showed that TCs possess transcriptional features of both medullary thymic epithelial cells (mTECs) and dendritic cells and comprise four subsets (TCI–TCIV). These cells emerge in the intestinal lymph nodes during the first two weeks after birth and promote the generation of peripheral regulatory T cells (pTregs), establishing tolerance to dietary and microbial antigens in early life.

Although the functions of these RORγt⁺ APCs are characterized (PMID: 40228524, 40185101, 40373113), their ontogeny remains unclear. One recent study indicates that RORγt⁺ dendritic cells (DCs) have lymphoid origin (PMID: 40185101), whereas another study suggests that tolerogenic DCs (tolDCs) arise from myeloid/dendritic cell progenitors (PMID: 40228524). The present study aims to clarify the progenitor–successor relationship in TC development and to identify the specific progenitors that give rise to TCs. The authors demonstrate that TCI and TCIII are labeled using a novel Aire reporter and lineage-tracing model (AireGreenLatern-creERT2R26Isl-tdTomato). Despite Aire expression, all TC subsets are of hematopoietic origin, shown in bone marrow chimeras. The authors demonstrate that although TCs lack expression of IL-7Ra, nearly all are labeled in IL-7Ra lineage-tracing mice (*Il7rcrR26Isl-tdTomato*), indicating that TCs originate from an IL-7Ra⁺ progenitor population. They found transcription factors PU.1 and Prdm16, as well as the environmental cue RANKL, are required for the

development of TC subsets. The authors' results support a model in which common lymphoid progenitors CLP give rise to TC I–IV cells via an intermediate progenitor that expressed Ror γ t. TC differ from ILC3 in their dependence on PU.1, and the different TC subsets show differential dependence on RANK–RANKL signaling (TC I), and Prdm16 (TC II–IV).

Overall, the study combines lineage tracing, conditional knockout models, and transcriptomic analyses to understand the developmental origin and transcriptional requirement of TC subsets. While the results are of interest, some points require further clarification and experimental support to strengthen the conclusions.

We thank the reviewer for their positive and constructive comments.

Major

-Other recent work suggested that Prdm16⁺ ROR γ t⁺ toIDCs, which share features with TC IV, are of myeloid/dendritic cell origin (PMID: 40228524). In contrast, this and other studies (PMID: 40185101) suggest that TCs have lymphoid origin, as suggested by adoptive transfer of CLP. However, the presented experiments do not establish whether all subsets (TC I–TC IV) have a developmental origin from CLP, as TC subsets were not assessed in these experiments. The authors need to establish whether all TC subsets or only some TC subsets derive from CLP.

In new data added to the paper, we show that all TC subsets are fate-mapped by IL7R in both young and adult mice – we have included this data as **Extended Data Fig. 2b**. For the adoptive transfer experiments, we recover all TC subsets as well as the TCP following CLP transfer (**new Extended Data Fig. 7d**). We have also performed ex vivo cultures of the immediate TCP using mLN slices from congenic 2-3 week old mice to support TCP differentiation. Excitingly we found that the TCP exclusively gives rise to TCs expressing TC II-IV marker genes including EpCAM and CD11c (**new Fig. 2e**). Critically, these cells do not give rise to CXCR6⁺ LTi cells or cDCs. Of note, although we could not recover TC I cells, we note that the lymph node cells do not remain viable for even brief periods of culture and we have likely lost LTo stromal cells, which are difficult to culture in vitro, and are the key cell type required for TC I differentiation. In addition, to ensure a pure population of progenitors, we cannot sort the TCP with lowest expression of ROR γ t due to contamination with ROR γ t⁻ cells. As shown in Fig 4a, TC I have the lowest levels of ROR γ t and likely arise from the ROR γ t^{lo} progenitors before upregulation of ROR γ t and PRDM16.

In addition to these ex vivo cultures, we generated a new *Siglec^fcre R26^{Isl-tdTomato}* mouse model for lineage tracing the FL ROR γ t⁺ progenitor, which expressed Siglec-F, and showed that an almost identical proportion of LTi cells and TC I-IV subsets are labeled (~40%), confirming that TCs and LTi cells have a common progenitor (**new Fig. 2f and g**). Thus, we believe that these experiments provide strong evidence that TC I-IV belong to one lineage of lymphoid-derived cells rather than e.g. TC IV representing a dendritic cell and TC I-III representing cells of lymphoid origin.

We also identified a cell intrinsic role for TCF4 in suppressing TLP differentiation and showed that all TC subsets are equally affected by TCF4 deficiency using competitive bone marrow chimeras (**new Fig. 3f and Extended Data Fig. 7e**). We hypothesized that TCs arise from a

lymphoid derived Siglec-H progenitor and demonstrated almost identical proportion of Siglec-H fate mapped cells across TC I-IV subsets (new **Fig. 3h**).

Together, we believe these data provide strong evidence that TC I-IV share a developmental pathway with a common immediate progenitor.

-The definition of TC progenitors (TCp) used may include lymphoid tissue inducer (LTi) progenitors, which also express ROR γ t (PMIDs: 33104170, 31882362, 32783932). In Fig. 4f, TCp is gated as Lin⁻CXCR6⁻ROR γ t⁺, which may include LTi progenitors in fetal liver. In Figure 4i, the gating strategy (Lin⁻CXCR6⁻ROR γ t⁺) for TCp in adult bone marrow may capture previously described adult bone marrow ILC3 precursors (PMID: 31128961). It is unclear if this TCp population is heterogeneous. If at all possible, the authors should assess the developmental potential of the TCp population to establish whether it can generate all TC subsets, and whether it also gives rise to LTi and ILC3. In addition, the authors could state whether PU.1 deficient mice and other genetic models used in these experiments have normal peripheral lymph nodes or whether they might have defects in LTi development or function.

We thank the reviewer for this incredibly thoughtful and fruitful perspective. We had previously thought that we had effectively excluded FL ROR γ t⁺ LTi and LTiP cells through exclusion of CXCR6 and CD90 expressing cells (**Fig. 1p, Extended Data Fig. 4d**), markers that were previously shown to be expressed by LTiP cells in the cited publications (PMID 32783932 and 33104170) as well as PMID 26779601 and 21909092. However, reflecting on our scRNA-seq data, we realized that CXCR6⁺CD90⁺ROR γ t⁺ FL cells were not present within our annotated LTiP or ROR γ t FL cluster (now termed TLP), and were instead clustering with more differentiated/mature LTi cells (**Fig. 1h and Extended Data Fig. 4d**). In contrast to these ROR γ t⁺CXCR6⁺CD90⁺ FL LTi cells, our cellRank 2 pseudotime analysis and fine subclustering of the ROR γ t⁺ FL cluster had identified 3 clusters which we had annotated as FL TCp, LN TCp and LN LTIP. Of note, all of these populations were CXCR6⁻. Intriguingly, our CellRank2 pseudotime analysis (**new Fig 1n**) indicated that the FL ROR γ t⁺ population we had labeled as the TCP was in fact the common progenitor to both TCP and LTIP with this bifurcation in cell fate and differentiation to restricted TC progenitors occurring predominantly in the LN. To test this, we have adopted a number of approaches:

First, we generated a new SS3 dataset to examine the LN TCP and LTIP populations. Our index sorting analysis confirmed that both of these cell types are CXCR6⁻, unlike differentiated LTi cells, and sit within our ROR γ t⁺MHCII⁺CXCR6⁻NCAM1⁻EpCAM⁻ flow cytometry gate (**new Fig. 2a-c**). Differential gene expression analysis revealed that LTIP and TCP could be distinguished by mutually exclusive expression of IL1R2 and PD1 (**Fig. 2a-b and d**) allowing us to establish a gating strategy for identification and FACS-isolation of these cells. Ex vivo cultures of the LN TCP using lymph node slices to support the differentiation of these cells revealed that LN TCP exclusively gave rise to TC subsets but not LTi cells, as detailed above.

Additionally, we sought to identify TLP genes that would allow us to lineage trace the progeny of these cells. Excitingly our differential gene expression analysis of TLP, LTIP and TCP identified expression of Siglec-F by the FL TLP and LN TCP. As outlined above, analysis of *Siglec^{fore}R26^{sl-tdTomato}* fate mapper revealed tdTomato expression by ~40-50% of all TC subsets and LTi cells (new **Fig. 2g**).

Thus, a key question is what are the TFs that determine TC vs LTi fate. As noted in our previous experiments, the LTiP transcriptionally resembles the ILCP, and is similarly dependent on Tox and TCF1. In contrast, PU1 appears to be the TF that is required for TC development. These findings are in line with a previous study from the Bendelac lab which identified increased accessibility for both TCF7 and PU1 motifs in FL ROR γ ⁺ progenitors which at the time were thought to represent LTiP (PMID 33104170).

In line with the role of PU1 in regulating TCP but not LTIP development, *Rorgt^{cre}Spi1^{fl/fl}* mice have normal numbers of LTi cells and have normal peripheral lymph nodes. Thus, PU1 appears to represent the bifurcation point for TCP and LTiP differentiation. *Il7r^{cre}Tcf7^{fl/fl}* mice have smaller lymph nodes than their littermate controls, yet despite this, the number of TCs is preserved.

Finally, providing further evidence of a common pathway for LTIP and TCP development, distinct from ILCP development, we found that TCF4 suppresses TLP differentiation in a cell intrinsic manner. Our computational analysis predicted that the TLP arose from early Siglec-H⁺ CLP-derived progenitors, prior to their upregulation of TCF4 (**Fig. 3g**). Since pDCs also arise from Siglec-H expressing progenitors, we speculate that the TLP emerges from the same precursors with TCF4 determining the bifurcation between pre-pDC and TLP cell fate. In support of this, we found that ~20-30% of LTi cells and TCs were fate mapped in mLN of *Siglech^{cre}R26^{Isl-tdTomato}* mice, in contrast to <5% of ILC1, 2 and 3 subsets, indicating that LTi and TCP share a developmental pathway through progenitors that may have low/variable expression of Siglec-H.

Other points:

Figure 4a is missing a negative control.

We have replaced this figure with an analysis of ROR γ T protein which is more functionally relevant than RORc (Venus) levels and have included a non-ROR γ T expressing cell type (CLP) as a negative control. The results remain the same – lowest levels of ROR γ T expression in the FL TLP and LN TC I, with a spectrum of ROR γ T levels in LN TCP and increased levels in TC II-IV.

The authors suggest TCs and ILCs arise from distinct developmental trajectories in part based on *Il7rCre Tcf7fl/fl* and *Il7rCre Toxfl/fl* mice, where TC numbers remain unaffected. However, to my knowledge, both types of mice still make some ILCs, and *Tcf7*-deficient mice possess LTi. Have the authors assessed ILC progenitors directly, by adoptive transfer?

We agree that *Il7r^{cre}Tcf7^{fl/fl}* mice have some ILC3s, as shown in **Fig. 4c**, although they exhibit an almost complete loss of other ILC subsets. However, despite a ~75% reduction in ILC3s, we still do not observe any impairment in TC numbers. In contrast, the *Tox* chimeras have virtually complete absence of ILCs and LTi cells, in keeping with the lack of lymph nodes in steady state *Il7r^{cre}Tox^{fl/fl}* mice. Unfortunately, we have not successfully established adoptive transfers of ILC progenitors. This is in part due to the need to use lymphoreplete mice that support TC differentiation, whereas ILCP transfers are typically performed with immunodeficient mice, which lack lymph nodes. We note that we previously assessed the precursor progeny relationship between ILCP and TCs using RORa-fate mapping mice (Akagbosu et al. PMID 36070798) and

showed that TCs are not descended from the $ROR_{\alpha+}$ ILCP. Our analysis of FL progenitors confirmed that the ILCP (but not TLP) expresses ROR_{α} (**reviewer Fig. 8**), thus we feel that this lineage tracing approach has already excluded a precursor-progeny relationship between ILCP and TCs.

The model presented in Figure 5d suggests TC progenitors (TCp) in the fetal liver are initially $ROR_{\gamma t}$ and later transition through $ROR_{\gamma t}^+$ stages. However, TCp appear to be defined by $Rorgt$ expression, so this is unclear.

We apologize for the confusion, this schema was meant to show that the TLP in FL are $ROR_{\gamma t}$ 'low', although we note that there are some cells in the FL TLP cluster which do not express *Rorc* (**Reviewer Fig. 9**). However, this could be due to dropout with 10X single-cell sequencing. We have updated the schema to reflect the expression of $ROR_{\gamma t}$ at different stages of development.

Authors could provide quantification of the number of TCp in fetal liver and adult mLN (P7) and bone marrow in Fig 4i.

We have included the absolute numbers of TLP in FL and adult BM (**Extended Data Fig. 10b**), demonstrating reduced numbers of TLP in bone marrow.

The authors should perform statistical analysis for the plots where p-values are currently missing, specifically in Figures 1h, 5a, and 5c.

We apologize for the oversight, we have added statistics for these and other plots.

Reviewer Table 1

CLP										
	Lin	FLT3	CSF1R	IL7R	CD117	Sca1	CD27	SiglecH	Ly6D	
Liu Nature 2022		+	-	+	int	int				
This study		+	-	+			+	-	-	
MDP/CDP										
	Lin	FLT3	CSF1R	IL7R	CD117	CD27	SiglecH	Ly6D	MHCI	CD11c
Feng et al. proDC (CDP)		+	+	-	lo/neg		lo/-	-	-	+
Feng et al. Myp-DC (MDP)		+	+	-	int		-		-	-
Rodriguez 2023		+	+	-						
Liu Nature 2022		+	+		int/hi (CDP/MDP)		-		-	-
This study		+	+	-		-	lo/-		-/lo	**
Feng et al. Fig3E and text										
** all MHCII-/lo cells are CD11c-										
pro-pDC/prepDC/tDC										
	FLT3	CSF1R	IL7R	CD117	SiglecH	Ly6D	MHCI	CD11c	CX3CR1	
Sulczewski 2023	+	-	+ / -	int-lo	+	+	-	+	+	
Rodriguez 2023 ** SPLEEN	+	-	+		+	+				
Feng 2022 (pro-pDC)	+	-	+	-/lo	+	+	-	lo	+	
This study	+	-	+/-		+			+		
Notes: Rodrigues et al. 2023 - tDCs are both CD11clo and CD11chi and correspond to pro-pDC										

Reviewer Figures

1. Reconciling the spectrum of ROR γ ⁺ APCs. **a**, Uniform manifold approximation and projection (UMAP) visualization of integrated single-cell transcriptomes of ROR γ ⁺ APCs from published studies (PMID 36070798, 36071167, 36071169, 40185101, 39091750, 40298935, 40373113 and 34767455). Cells colored by their original cell-type annotation from available metadata or expression of published signature genes in reanalysis of extracted ROR γ ⁺MHC class II⁺ single-cell transcriptomes. **b**, Expression of Aire delineating TC I and TC III subsets. **c**, Reconciling nomenclature across individual studies. These integrated data indicate the presence of five ROR γ ⁺ APCs: LTi cells, and four subsets of TCs (TC I–IV). JC, Janus cell; TolDC, tolerizing dendritic cell; R-eTAC, ROR γ ⁺ extra-thymic Aire-expressing cell.
2. Absolute number of cells in mLN of CD45.1 recipient mice, 4 or 8 weeks post-irradiation and transfer of CD45.2 bone marrow cells.
3. Proportion of tdTomato⁺ cells among TC subsets in mLN of P10 *Rorc*^{Venus-creERT2}*R26*^{sl-tdTomato} mice that received one dose of 4-OH tamoxifen at E18.5.
4. Expression of *Kit* and *Ly6d* in fetal liver progenitors as in Fig. 1h.
5. 10² sort-purified fetal liver CLP (as in Extended Data. Fig. 7c) from E18.5 *Rorc*^{Venus} mice were cultured for 14d in cRPMI containing IL-7, SCF and Flt3L (each 20ng/ml). Where indicated, GM-CSF (20 ng/ml) was added to cultures from day 8. Representative flow plot and summary graph of 10 technical replicate wells.
6. Expression of Aire (GreenLantern) by IL1R2⁺ TCP (left) and TLP (right) in mLN or FL of *Rorc*^{Venus}*Aire*^{GreenLantern} mice.
7. Expression of published hematopoietic progenitor gene signatures in FL progenitor cells in Fig. 1h.
8. Expression of *Rora* in ILC, LTi and TC progenitors. Clusters defined in Fig. 1h and I.
9. Dot plot showing expression of *Rorc* in TCP and TLP clusters defined in Fig. 1h and I.

Referees' comments:

Referee #1 (Remarks to the Author):

The authors have clearly invested substantial effort into revising the manuscript. They have added additional genetic models (including Tcf4STOP/STOP), refined progenitor gating strategies informed by scRNA-seq and showed them in Extended data, and expanded their discussion of nomenclature and prior literature. These changes improve the internal consistency of the developmental model and increase methodological transparency.

We thank the reviewer for their positive comments.

However, several of the key concerns raised in the initial review remain only partially addressed. In many instances, the responses rely on extended explanation, reinterpretation, or removal of data rather than providing direct experimental support. The main conclusions are still stated in strong terms, but the underlying evidence continues to have important gaps. The specific areas where the responses are not fully convincing are below:

1. Scope, impact, and lack of functional data

The reviewer understands the conceptual gap the authors aim to address: defining a progenitor (TCp) and developmental pathway for Thetis cells and clarifying their relationship to LTi cells and other progenitors. Nevertheless, two central issues remain. First, even with the lineage-tracing and transfer experiments, the data do not convincingly and definitively establish that these cells are lymphoid-derived; the evidence is suggestive but not conclusive. Second, by design, the revised manuscript still contains no new functional data. Given that previous high-profile studies have already established many of the functional properties and tolerogenic roles of these cells, a purely ontogeny-focused manuscript without new in vivo function further narrows the scope and confines the primary impact to researchers interested specifically in developmental lineage, rather than the broader readership.

We believe that the central novelty of our study is not the lymphoid ontogeny of TCs per se but rather the identification of the progenitor, the delineation of its relationship to the LTi lineage, and the discovery of the transcriptional regulators required for both the broad TC lineage and specific TC subsets. These conceptual advances provide critical insights into both TC and LTi cell development. Importantly, reviewers 2 and 3 - both experts in hematopoiesis - agreed that the manuscript is sufficiently novel, mechanistically meaningful, and impactful for publication in Nature, even without new functional assays.

2. Novelty of transcription factor findings

In their response, the authors compare their analysis of PU.1, RelB, and other transcription factors in TC development to prior work mapping discrete regulatory regions in the Zeb2 locus for cDC2 specification (i.e., the Ken Murphy study in Nature). The reviewer does not find this comparison appropriate. The Zeb2 study provided a carefully designed, mechanistically precise dissection of specific enhancer regions that clearly reshaped understanding of cDC2 specification. By contrast, in the present manuscript, the roles of PU.1 and RelB are not resolved at the same mechanistic depth; the authors do not map specific regulatory elements or provide an equivalent level of causal detail. While the PU.1 and RelB data may be of interest within the TC ontogeny framework, the novelty and conceptual impact are not comparable to the cited Zeb2 work.

We apologize for the confusion. Our response in relation to Zeb2 was not intended to equate the mechanistic level of the two studies, but rather to highlight that delineating the lineage of TCs does not require the identification of new transcription factors, not previously implicated in myeloid or epithelial cell biology, to be impactful. Our findings that PU.1 and RelB regulate TC development and subset heterogeneity provide a foundational framework for future fine-scale studies - similar to how initial observations of e.g. IRF8 dependency of cDC1 differentiation preceded enhancer dissection in cDC1s. Thus, our results represent important first steps in defining the regulatory architecture of the TC lineage.

3. Handling and removal of data

The Zeb2 deletion data were initially presented and then removed in the revision. It remains unclear why these data

were considered sufficiently novel and robust to include at first, but are now omitted. If the authors stand by the scientific novelty of those results, a more explicit rationale for their removal is warranted. Conversely, if the data were judged to be insufficiently robust or potentially misleading, that should also be clearly stated. As it stands, the decision appears strategic rather than scientific, and this does not increase confidence in the overall narrative. Similar considerations apply to other datasets that have been removed or substantially reinterpreted between versions; greater transparency about these decisions would be helpful.

We apologize for the confusion regarding the handling of the Zeb2 data. The removal was not due to lack of robustness or novelty. Rather, in the first round of review, Reviewer 1 expressed concern that our prior work (Cabric et al.) had already suggested that Zeb2 was dispensable for TC differentiation and therefore questioned the originality and significance of including these data (Reviewer 1, Section B). Because our intention in presenting the Zeb2 experiments was to clarify the relationship between TCs and tDCs, and because the new TCF4-null data addressed this question more directly and definitively, we felt the Zeb2 dataset was redundant. We stated this explicitly in our prior response. That said, if the reviewer now feels that inclusion of the Zeb2 data would strengthen transparency and completeness, we would be very happy to restore these experiments to the manuscript, as they provide orthogonal evidence for a tDC-independent pathway of TC differentiation.

The reviewer also notes that “other datasets” were removed or substantially reinterpreted. We may have misunderstood which specific datasets are being referenced, as aside from the Zeb2 data, we made only two substantive changes:

1. Bone-marrow chimera experiments (Supplemental Figure 1)

These were replaced in response to Reviewer 2's request that we perform analysis of TCs from reciprocal CD45.1/CD45.2 chimeras rather than the Aire-creERT2-GL chimera strategy, to allow a more direct and quantitative comparison of TC reconstitution from each donor source. We described this change explicitly in our response to Reviewer 2.

2. Rorc-creERT2 fate-mapping experiments (previously in Figure 1).

As the revision progressed, the manuscript became substantially longer due to additional datasets requested during peer review. To maintain clarity and place the identification of the TC progenitor at the forefront of the paper, we moved most of old Figure 1 to supplemental and removed the RORc-creERT2 fate mapping panels that primarily illustrated the kinetics of TC development rather than advancing the central conceptual message. The removal was also in part in response to Reviewer 1's comments which indicated some confusion in interpreting these experiments, largely due to the inherent limitations of “time-stamping” models in which both progenitors and mature TCs can be labeled, making it difficult to distinguish incomplete Cre efficiency from post-labeling turnover. Because this limitation cannot be resolved with the Rorc-creERT2 model, we did not feel we could adequately address the reviewer's concerns with the existing data. We are currently awaiting litters from Prdm16-creERT2 fate-mapping mice to more rigorously examine TC lifespan. However, if the reviewer would prefer that the original Rorc-creERT2 data be retained for completeness, we would be happy to include them in the Supplementary Information. We note that we had provided a justification for removal of these panels in our prior response.

More broadly, we performed additional analyses of scRNA-seq experiments during the revision which yielded new insights into TC-LTi relationships. These changes were made in direct response to reviewer feedback, in particular Reviewer 3, with the goal of strengthening the manuscript and improving clarity - precisely the purpose of peer review. Our rationale for each modification was detailed in our point-by-point responses, which would be publicly available as part of the transparent peer review file if the manuscript is published.

We hope this clarifies the reasoning behind the changes and reassures the reviewer that our decisions were motivated by scientific rigor, clarity, and responsiveness to the review process rather than strategic considerations.

4. Bone marrow chimera design and reconstitution timing

The authors maintain that 4-week bone marrow chimeras are adequate because they are focusing on innate populations, which they state reconstitute by 2 weeks. The reviewer does not agree that this argument is sufficient. There is a substantial body of literature indicating that full reconstitution of lymphoid organs, particularly with respect

to migratory dendritic cells and proper lymph node architecture, typically requires ~8 weeks. Given that migration, stromal integrity, and tissue organization are central to DC priming of T cells and tolerance mechanisms (and given that TCs are proposed to be key tolerogenic APCs) the structural and temporal context of the lymph node should be carefully controlled. Using 4-week chimeras may be acceptable as one data point, but the reviewer does not consider this timing fully adequate to support strong conclusions about hematopoietic origin and relative contributions of distinct progenitors, especially when claims are framed as definitive.

We respectfully disagree. As noted in our previous reviewer response, we directly compared the numbers of DC and TC subsets at 4 weeks vs 8 weeks post reconstitution and did not identify differences in cell numbers. We provided this data in the previous rebuttal and will include this in the supplementary data. We note that our findings are in line with previously published studies that demonstrated full reconstitution of innate immune cell populations by 2-4 weeks post irradiation (PMID 15570252, 21177980 and 16797423). The purpose of these chimeras was not to assess lymph node architecture, T cell priming, or tolerance induction, but to determine whether specific innate cell types reconstitute from defined progenitors. The concern raised by the reviewer appears to be that it would take longer than 4 weeks for our cell types of interest to appear and therefore we would miss key populations that would lead to erroneous conclusions in relation to their hematopoietic origin. However, in all of our panels we show that we identify all of the relevant innate cell types at 4 weeks.

5. Evidence for lymphoid origin and adoptive transfer benchmarks

Lymphoid origin of these cells has already been suggested by previous studies using lineage-tracing. The main additional element in this manuscript is the adoptive transfer of CLP and other progenitors. However, the benchmarks and outcomes of these transfers, in the authors' hands, do not cleanly recapitulate key features reported by others, raising doubts about how confidently one can interpret these experiments. Moreover, the authors state that they were unable to perform single-cell differentiation assays due to technical limitations. While the technical challenges are understood, for a paper that focuses almost exclusively on ontogeny and aims for Nature level impact, the absence of single-cell or clonal-resolution lineage data is a significant limitation. Simply stating that it was not possible is not fully satisfactory. Alternative strategies (e.g., monolayers, stromal or bone marrow feeder layers, or modified culture systems) could have been explored more deeply. In the absence of these, the strength of the conclusion that TCs arise from lymphoid progenitors remains weaker than the manuscript implies.

We apologize for the confusion. We should have provided more context to our previous single cell culture experiments. In addition to single cell cultures with media/cytokines/growth factors, we performed the experiments suggested by the reviewer (culture with stromal cells (OP9 +/- Notch ligands) and culture with bone marrow feeder cells) but we could not identify conditions which supported TC differentiation from single cell progenitors. We do not believe that single cell cultures are required for evidence of lymphoid origin, or benchmarking of adoptive transfers as bulk cultures can provide the same information. To this end, we included data demonstrating evidence of TC generation from bulk CLP and we can include this data in the revised manuscript alongside bulk CDP/MDP and pre-pDC/tDC progenitors. However, we note that this data would not be novel, as the previous study from the Colonna lab already included bulk in vitro cultures and showed that across different progenitors i) only CLP could give rise to TCs and ii) TCs were not generated under culture conditions that support dendritic cell differentiation but rather required a combination of FLT3, SCF and IL7. Besides in vitro cultures, we provided data from orthogonal approaches which support the proposed lymphoid pathway for TC development. In addition, we note that the previous request for single cell cultures was to confirm that all TC subsets arose from a common progenitor. Since this has been established by multiple lineage tracing approaches, we are unclear about the value of single cell CLP cultures.

Related to this, the way CLP-derived progeny is presented is still confusing (even with the included explanation and extended data). It appears, from the plots, that CLP transfers generate essentially every cell type, including substantial numbers of cDC1s and cDC2s. If this is truly the case, it raises concerns about purity, benchmarking, and specificity. If instead this is largely a matter of plotting strategy and gating (for example, focusing on Lin- MHCII⁺ cells and underrepresenting lymphocytes), then this needs to be made much clearer not in the text, but in the figure itself. As currently presented, the data are difficult to interpret and risk giving a misleading impression of both the breadth and specificity of CLP output. Similar concerns apply to the reported outputs of pre-pDC/tDC populations, where the emergence of monocytes (and cDC1s as mentioned before) is unexpected and should be explicitly benchmarked and

justified.

Previous studies have already shown that the FL CLP generates myeloid cells including cDC1s and cDC2s (PMID 11359812 and 33469015), thus our findings are in line with published data. Moreover, our scRNA-seq and flow cytometry analysis of IL7R fate-mapped cells further confirmed that IL7R⁺ progenitors can give rise to a broad array of cell lineages that includes cDC1, cDC2 and monocytes. We also note that our conclusions do not rely on the results of the adoptive transfer experiments. We have used genetic lineage tracing, genetic epistasis and adoptive transfer experiments, and in vitro bulk cultures to establish the developmental pathway for TCs. We explicitly stated in the manuscript that the numbers of cells recovered from pre-pDC/tDC transfers were low, therefore it is difficult to draw strong conclusions from the recovered cell types where e.g. 1% may represent one cell. We therefore used TCF4 deficient mice to address the relationship between tDCs and TCs, and showed that TCs were not derived from TCF4-dependent progenitors.

6. Lineage bias, pDC ontogeny, and TCF4 interpretation

The manuscript now places substantial emphasis on TCF4 in balancing pDC versus TLP/TC fates and uses this to reinforce a model in which TCs share a progenitor with pDCs. However, the ontogeny of pDCs itself remains an active area of debate, with evidence from both sides of the debate: some groups arguing for myeloid origin and others favoring lymphoid contributions. The authors acknowledge this controversy, yet the figures and text (e.g. in Figures 1e and 3i) are framed in a way that effectively assumes a lymphoid identity for pDCs, which then conveniently supports the view that TCs are also lymphoid. In the reviewer's opinion, this approach is biased. Using a debated lineage assignment for pDCs as a scaffold to support a firm lymphoid classification for TCs is not scientifically robust in the absence of more definitive experimental evidence.

More broadly, the overall argument for lymphoid origin relies on a combination of IL7R/hCD2 fate mapping, scRNA-seq clustering, and adoptive transfers whose benchmarks and gating choices are not entirely convincing. IL7R expression alone does not define lymphoid origin, as the authors themselves note, and transcriptomic resemblance is inherently suggestive rather than conclusive. Given these caveats, the reviewer does not agree that the manuscript has "definitively" established lymphoid origin. A more cautious framing would be appropriate.

We apologize for the confusion. We show in both the figures and state in the text that pDCs can arise from both lymphoid and myeloid pathways of development in line with studies from the Colonna, Tussiwand, Murphy and Ginhoux labs. We explicitly state in the text that the lymphoid pathway for pDCs is debated. Our conclusions on lymphoid ontogeny for TCs are unrelated to the ontogeny of pDCs and we are not using the pDC lineage assignment as a scaffold to support a lymphoid classification for TCs. Rather the lymphoid ontogeny for TCs relates to i) the IL7R lineage tracing alongside exclusion of a contribution from other IL7R progenitors such as tDCs/pre-pDCs using TCF4 null mice, ii) the CLP transfers (with a CLP gating strategy that is even more stringent than other gating strategies due to inclusion of CD27), iii) the identification of a common LTi/TC progenitor and the multitude of evidence from previously published studies establishing that LTi cells are lymphoid derived cells.

Moreover, we have already provided a cautious framing in the manuscript. We are happy to revise the text further to clarify that the model does not preclude the existence of a parallel myeloid pathway for pDC development.

In summary, while the manuscript is improved and now presents a clearer and more internally consistent developmental model, many of the original concerns remain. The conclusions continue to overstate what can be definitively supported by the available data, and the overall contribution, though valuable within a specialized developmental field, does not substantially extend beyond existing knowledge generated by several recent studies.

Referee #2 (Remarks to the Author):

I have read the revised version of this manuscript and the authors' response. Overall, the authors have addressed

each point that I raised in the first round of review, either by new data, new analyses, or by way of explanation. I remain positive in my support of the paper.

The one thing that really needs attention now is the fact that the authors claim to have evidence for a common progenitor for TC and LTi cells. However, they cannot prove that one progenitor has such dual fates because this would require single cell assays in which both lineages arise from a single progenitor. This could also be achieved by barcoding. In the absence of such evidence for dual fate progenitors as opposed to single fates within the progenitor population they authors should make clear in the entire paper that they describe a dual fate progenitor population. For instance, in the summary, the statement "Here, we identify a ROR γ ⁺ TC and lymphoid tissue inducer (LTi) cell progenitor (TLP) that differentiates into the immediate ROR γ ⁺ TC progenitor (TCP) and the LTi progenitor (LTiP)" suggest a bipotent progenitor which, in light of the point raised above, the authors have not shown. To be correct, the author should call it a progenitor population and this ambiguity or open question should be added to the discussion.

We thank the reviewer for their positive feedback. We will revise the abstract accordingly and add this open question to the discussion.

Referee #4 (Remarks to the Author):

The revised manuscript addresses all of my questions and criticisms. As suggestions, Extended data 7d showing that all Thetis cell populations are evident in mice reconstituted with CLP could be in the main figures and not in supplement. The CLP definition used here is (appropriately) a refined definition termed ALP published by the Weissman lab (PMID: PMID: 19833765); this key paper could be referenced. Some figures still use Latin instead of Greek letters for "gamma" in ROR γ , for e.g. Figure 5, there may be others. This could be corrected.

The work provides a wealth of experimental data from transcriptional analysis, adoptive transfer, in vitro cell culture, in vivo lineage tracing, and transcription factor deficient mice, together indicating that Thetis cells develop from lymphoid progenitors. The work is done to a high standard. The revised manuscript includes new data indicating that LTi and Thetis cells share a developmental pathway, distinct from other ILCs that also derive from lymphoid progenitors. It identifies transcription factor dependencies for many of the proposed developmental steps. The data here revise our thinking about lymphoid progenitor-derived cells to include tolerogenic APCs whose main function is to present antigens to CD4 T cells in lymph nodes. This is an important insight, and I am strongly supportive.

We thank the reviewer for their positive feedback. We will cite the paper from the Weissman lab and update the formatting for ROR \$\gamma\$ throughout the manuscript.

Referee #2 (Remarks to the Author):

I have read the revised version of this manuscript and the authors' response. Overall, the authors have addressed each point that I raised in the first round of review, either by new data, new analyses, or by way of explanation. I remain positive in my support of the paper.

The one thing that really needs attention now is the fact that the authors claim to have evidence for a common progenitor for TC and LTi cells. However, they cannot prove that one progenitor has such dual fates because this would require single cell assays in which both lineages arise from a single progenitor. This could also be achieved by barcoding. In the absence of such evidence for dual fate progenitors as opposed to single fates within the progenitor population they authors should make clear in the entire paper that they describe a dual fate progenitor population. For instance, in the summary, the statement "Here, we identify a ROR γ ⁺ TC and lymphoid tissue inducer (LTi) cell progenitor (TLP) that differentiates into the immediate ROR γ ⁺ TC progenitor (TCP) and the LTi progenitor (LTiP)" suggest a bipotent progenitor which, in light of the point raised above, the authors have not shown. To be correct, the author should call it a progenitor population and this ambiguity or open question should be added to the discussion.

We thank the reviewer for their positive feedback. We have revised the abstract and main text accordingly and acknowledged this open question in the discussion.

Referee #4 (Remarks to the Author):

The revised manuscript addresses all of my questions and criticisms. As suggestions, Extended data 7d showing that all Thetis cell populations are evident in mice reconstituted with CLP could be in the main figures and not in supplement. The CLP definition used here is (appropriately) a refined definition termed ALP published by the Weissman lab (PMID: PMID: 19833765); this key paper could be referenced. Some figures still use Latin instead of Greek letters for "gamma" in ROR γ , for e.g. Figure 5, there may be others. This could be corrected.

The work provides a wealth of experimental data from transcriptional analysis, adoptive transfer, in vitro cell culture, in vivo lineage tracing, and transcription factor deficient mice, together indicating that Thetis cells develop from lymphoid progenitors. The work is done to a high standard. The revised manuscript includes new data indicating that LTi and Thetis cells share a developmental pathway, distinct from other ILCs that also derive from lymphoid progenitors. It identifies transcription factor dependencies for many of the proposed developmental steps. The data here revise our thinking about lymphoid progenitor-derived cells to include tolerogenic APCs whose main function is to present antigens to CD4 T cells in lymph nodes. This is an important insight, and I am strongly supportive.

We thank the reviewer for their positive feedback. We have cited the paper from the Weissman lab and updated the formatting for ROR γ throughout the figures and legends.